# Gaussian Pre-Activations in Neural Networks: Myth or Reality?

## Abstract

The study of feature propagation at initialization in neural networks lies at the root of numerous initialization designs. An assumption very commonly made in the field states that the pre-activations are Gaussian. Although this convenient *Gaussian hypothesis* can be justified when the number of neurons per layer tends to infinity, it is challenged by both theoretical and experimental works for finite-width neural networks. Our major contribution is to construct a family of pairs of activation functions and initialization distributions that ensure that the pre-activations remain Gaussian throughout the network's depth, even in narrow neural networks. In the process, we discover a set of constraints that a neural network should fulfill to ensure Gaussian pre-activations. Additionally, we provide a critical review of the claims of the Edge of Chaos line of works and build an non-asymptotic Edge of Chaos analysis. We also propose a unified view on pre-activations propagation, encompassing the framework of several well-known initialization procedures. Finally, our work provides a principled framework for answering the much-debated question: is it desirable to initialize the training of a neural network whose pre-activations are ensured to be Gaussian?

## Notations and vocabulary

Bold letters $\mathbf{Z}$, $\mathbf{W}$, $\mathbf{B}$, $\mathbf{x}$... represent tensors of order larger or equal to 1. For a tensor $\mathbf{W} \in \mathbb{R}^{n \times p}$, we denote by $W_{ij} \in \mathbb{R}$ its component at the intersection of the $i$-th row and $j$-th column, $\mathbf{W}_{i\cdot} \in \mathbb{R}^{1 \times p}$ its $i$-th row and $\mathbf{W}_{\cdot j} \in \mathbb{R}^n$ its $j$-th column. Upper-case letters $\mathbf{W}$, $X$, $Y$, $Z$, $G$... represent random variables. For a random variable $Z$, the function $f_Z$ represents its density, $F_Z$ its Cumulative Distribution Function (CDF), $S_Z$ its survival function, and $\psi_Z$ its characteristic function. The *depth* of a neural network is its number of layers. The *width* of one layer is its number of neurons or convolutional units. The *infinite-width limit* of a neural network is the limiting case where the width of each layer tends to infinity.

# 1 Introduction

Let us take a neural network with $L$ layers, in which every layer $l \in [1, L]$ performs the following operation:

$$\mathbf{X}^{l+1} := \phi(\mathbf{Z}^{l+1}), \tag{1}$$

$$\text{with:} \quad \mathbf{Z}^{l+1} := \frac{1}{\sqrt{n_l}} \mathbf{W}^l \mathbf{X}^l + \mathbf{B}^l, \tag{2}$$

where $\mathbf{X}^{l+1} \in \mathbb{R}^{n_{l+1}}$ is its *activation*, $\mathbf{Z}^{l+1} \in \mathbb{R}^{n_{l+1}}$ its *pre-activation*, $\phi$ is the coordinate-wise *activation function*, $\mathbf{W}^l \in \mathbb{R}^{n_{l+1} \times n_l}$ is the *weight matrix* of the layer, $\mathbf{B}^l \in \mathbb{R}^{n_{l+1}}$ its *vector of biases*, and $\mathbf{X}^l \in \mathbb{R}^{n_l}$ its *input* (also the preceding layer activation). This paper focuses on the distribution of the pre-activations $\mathbf{Z}^l$ as $l$ grows, for a fixed input $\mathbf{x}$, and weights $\mathbf{W}^l$ and biases $\mathbf{B}^l$ randomly sampled from known distributions.

Recurring questions arise in both Bayesian deep learning and in parameter initialization procedures: How to choose the distribution of the parameters $\mathbf{W}^l, \mathbf{B}^l$ and according to which criteria, and how should the distribution of the pre-activations $\mathbf{Z}^l$ look like? Answering these questions is fundamental to finding efficient ways of initializing neural networks, that is, appropriate distributions for $\mathbf{W}^l$ and $\mathbf{B}^l$ at initialization. In Bayesian deep learning, this question is related to the search for a suitable prior, which is still a topic of intense research (Wenzel et al., 2020; Fortuin et al., 2022).

**Initialization strategies.** A whole line of works in the field of initialization strategies for neural networks is based on the preservation of statistical characteristics of the pre-activations when they propagate into a network. In short, the input of the neural network is assumed to be fixed, while all the parameters are considered as randomly drawn, according to a candidate initialization distribution. Then, by using heuristics, some statistical characteristics of the pre-activations are deemed desirable. Finally, these statistical characteristics are propagated to the initialization distribution, which indicates how to choose it.

For instance, one of the first results of this kind, proposed by Glorot & Bengio (2010), is based on the preservation of the variance of both the pre-activations and the backpropagated gradients across the layers of the neural network. The resulting constraint on the initialization distribution of the weights $\mathbf{W}^l$ is about its variance: $\mathrm{Var}(W_{ij}^l/\sqrt{n_l}) = 2/(n_l + n_{l+1})$.[1] Then, He et al. (2015) have refined this idea by taking into account the nonlinear deformation of the pre-activations by the activation function. They also showed that the inverted arithmetical average $2/(n_l + n_{l+1})$, resulting from a compromise between the preservation of variance during both propagation and backpropagation, can be changed into $1/n_l$ or $1/n_{l+1}$ with negligible damage to the neural network.[2] Notably, with $\phi(x) = \mathrm{ReLU}(x) := \max(0, x)$, they obtain $\mathrm{Var}(W_{ij}^l) = 2/n_l$, where the factor 2 is meant to compensate for the loss of information due to the fact that ReLU is zero on $\mathbb{R}^-$.

---

[1] In the original paper, the considered weight matrix is $\tilde{\mathbf{W}}^l := \mathbf{W}^l/\sqrt{n_l}$, so $\mathrm{Var}(\tilde{W}_{ij}^l) = 2/(n_l + n_{l+1})$.

[2] More generally, any choice of the form $1/(n_l^\alpha \, n_{l+1}^{1-\alpha})$ with $\alpha \in [0, 1]$ is valid, as long as the same $\alpha$ is used for all layers.

After these studies, Poole et al. (2016) and Schoenholz et al. (2017) focused on the correlation between the pre-activations of two data points $\mathbf{x}_a$ and $\mathbf{x}_b$. By preserving this correlation when propagating the inputs into the neural network, information about the geometry of the input space is meant to be preserved. So, training the weights is meaningful at initialization, regardless of their positions in the network. This heuristic is finer than the preceding ones since attention is paid to the correlation between pre-activations and their variance (a joint criterion is used instead of a marginal one). The range of valid initialization distributions is changed accordingly, with a relation between $\sigma_w^2 = \text{Var}(W_{ij}^l)$ and $\sigma_b^2 = \text{Var}(B_i^l)$ that should be ensured. This specific relationship is referred to as the *Edge of Chaos* (EOC).

Finally, it is worth mentioning the work of Hayou et al. (2019), in which the usual claims about the EOC initialization are tested with several choices of activation functions. Notably, the authors have run a large series of experiments in order to check whether the intuition behind the EOC initialization leads to better performance after training. Also, at the opposite of finding an initialization distribution for the parameters, Klambauer et al. (2017) focused on tuning the parameters of the activation function (leading to the SELU activation function). As in the preceding techniques, they aim for variance-preserving layers.

In the following, the term "Edge of Chaos" is used in two different manners: "EOC framework", "EOC formalism", or "EOC theory" refer to a setup where input data points are deterministic and weights and biases are random, whereas in the context of initialization of weights and biases, "EOC" alone refers to a specific set of pairs $(\sigma_w^2, \sigma_b^2)$ matching a given theoretical condition (Point 2, see Section 2.1).

**Bayesian prior and initialization distribution.** There exists a close relationship between the initialization distribution in deterministic neural networks and the prior distribution in Bayesian neural networks. For instance, let us use variational inference to approximate the Bayesian posterior of the parameters of a neural network (Graves, 2011). In this case, the Bayesian posterior is approximated sequentially by performing a gradient descent over the so-called "variational parameters" (Hoffman et al., 2013). This technique requires to backpropagate the gradient of the loss through the network, as when training deterministic networks. Therefore, as with the initialization distribution, the prior distribution must be constructed in such a way that the input and the gradient of the loss propagate and backpropagate correctly (see Sec. 2.2, Ollivier, 2018).

**Gaussian hypothesis for the pre-activations.** Within a context of random weights and biases, we call the *Gaussian hypothesis* the assumption that all the pre-activations $Z_i^l$ are Gaussian random variables, at any layer $l$ and for any neuron $i$. This hypothesis is common in the theoretical analysis of the properties of neural networks at initialization. Specifically, this is a fundamental assumption when studying the "Neural Tangent Kernels" (NTK) (Jacot et al., 2018) or Edge of Chaos (Poole et al., 2016). In a nutshell, the NTK is an operator describing the optimization trajectory of an *infinitely wide* neural network (NN), which is believed to help understand the optimization of ordinary NNs. On one side, this Gaussian hypothesis can be justified in the case of "infinitely wide" NNs (i.e., when the widths $n_l$ of

the layers tend to infinity), by application of the Central Limit Theorem (Matthews et al., 2018). On the other side, this Gaussian hypothesis is apparently necessary to get the results of the EOC and NTK lines of work. However, it remains debated for both theoretical and practical reasons.

First, from a strictly theoretical point of view, it has been shown that, for finite-width NNs (finite $n_l$), the distribution of $Z_i^l$ has heavier tails as $l$ increases, that is, as information flows from the input to the output (Vladimirova et al., 2019; 2021). Second, a series of experiments tend to show that pushing the distribution of the pre-activations towards a Gaussian (e.g., through a specific Bayesian prior) leads to worse performances than pushing it towards distributions with heavier tails, e.g., Laplace distribution (Fortuin et al., 2022).

Besides, the condition under which the Gaussian hypothesis remains valid is an important source of confusion. As an example, Sitzmann et al. (2020) state that: "for a uniform input in $[-1, 1]$, the [pre-]activations throughout a SIREN[3] are standard normal distributed [...], irrespective of the depth of the network, if the weights are distributed uniformly in the interval $[-c, c]$ with $c = \sqrt{6}/n_l$ in each layer $[l]$." (Theorem 1.8, Appendix 1.3). Though this formal statement seems to hold for all layers and whatever their widths, it is only an asymptotic result, since it uses the Central Limit Theorem in its proof. Consequently, this theorem is not usable in practical SIRENs, since it does not provide any speed of convergence of the distribution of the pre-activations to a Gaussian, as each $n_l$ tends to infinity.[4]

**Contributions.**   Our goals are twofold: first, we aim to reproduce and test the results of the papers in the EOC line of works; second, we aim to move beyond the Gaussian hypothesis in finite neural networks. Accordingly, we have obtained the following results:

- we experimentally demonstrate that the Gaussian hypothesis is mostly invalid in multilayer perceptrons with finite width;

- we show that, contrary to a claim of Poole et al. (2016) and usual practical results in the EOC framework, the variance of the pre-activations does not always have at most one nonzero attraction point; we provide an example of an activation function for which the number of such attraction points is infinite;

- we deduce a set of constraints that the activation function and the initialization distribution of weights and biases must fulfill to guarantee Gaussian pre-activations at initialization (including with finite-width layers);

- we propose a new family of activation functions and initialization distributions designed to achieve this goal (Gaussian pre-activations at initialization);

---

[3]Sinusoidal representation network.

[4]Hopefully, according to Matthews et al. (Th. 4, 2018), the pre-activations tend to become Gaussian irrespective of the growth rates of each $n_l$, so Theorem 1.8 of Sitzmann et al. (2020) is *asymptotically true for all layers and all growth rates of each $n_l$*. But this result still does not provide any convergence speed.

- we demonstrate empirically that the distribution of the pre-activations always tends to drift away from the standard Gaussian distribution during propagation; however, this drift is much greater when using tanh and ReLU activation functions than ours.

Additionally, we train, evaluate and compare neural networks built according to our family of activation functions and initialization distributions, and usual ones (tanh or ReLU, Gaussian EOC initialization).

**Summary of the paper.** As a preliminary, we make in Section 2 a critical review of several results about pre-activations propagation in a neural network: the discussion, additional experiments, and the criticism we are proposing, particularly in the EOC line of works, are the foundations of our contributions. In Section 3, we propose a new family of activation functions, along with a family of initialization distributions. They are defined so as to ensure that the pre-activations distribution propagates without deformation across the layers, including with networks that are far from the "infinite-width limit". More specifically, we ensure that the pre-activations remain Gaussian at any layer, and we provide a set of constraints that the activation function and the initialization distribution of the parameters should match to attain this goal. Finally, we propose in Section 4 a series of simulations in order to check whether our propositions meet the requirement of maintaining Gaussian pre-activations across neural networks. We also show the performance of trained neural networks in different setups, including standard ones and the one we are proposing.

## 2 Propagating pre-activations

In this section, we propose a critical review of several aspects of the Edge of Chaos framework. We recall the fundamental ideas of the EOC in Section 2.1. In Section 2.2, we perform some experiments at the initialization of a multilayer perceptron, in which we propagate data points sampled from CIFAR-10. These results illustrate a limitation of the EOC framework when using neural networks with a small number of neurons per layer. Then, we build in Section 2.3 an activation function such that the variance of the propagated pre-activations admits an infinite number of stable fixed points, which is a counterexample to a claim of Poole et al. (2016). Finally, we propose in Section 2.4 a unified representation of several initialization procedures.

### 2.1 Propagation of the correlation between data points

**"Edge of Chaos" (EOC) framework.** In the EOC line of work, the inputs of the neural network are supposed to be fixed, while the weights and biases are random. In order to study the propagation of the distribution of the pre-activations $\mathbf{Z}^l$, Poole et al. (2016) and

Schoenholz et al. (2017) propose to study two quantities:

$$v_a^l := \mathbb{E}[(Z_{j;a}^l)^2], \tag{3}$$

$$c_{ab}^l := \frac{1}{\sqrt{v_a^l v_b^l}} \mathbb{E}[Z_{j;a}^l Z_{j;b}^l], \tag{4}$$

where $Z_{j;a}^l$ is the $j$-th coordinate of the vector $\mathbf{Z}_a^l$ of pre-activations before layer $l$, when the input of the neural network is a data point $\mathbf{x}_a$. The expectation is computed over the full set of the parameters, i.e., weights and biases. So, we can interpret $v_a^l$ as the variance of the pre-activations of a data point $\mathbf{x}_a$, and $c_{ab}^l$ as the correlation between the pre-activations of two data points $\mathbf{x}_a$ and $\mathbf{x}_b$, over random initializations of the parameters, distributed independently in the following way:

$$W_{ij}^l \sim \mathrm{P}_w(\sigma_w) \quad \text{with} \quad \mathbb{E}[W_{ij}^l] = 0 \quad \text{and} \quad \mathrm{Var}(W_{ij}^l) = \sigma_w^2, \tag{5}$$

$$B_i^l \sim \mathrm{P}_b(\sigma_b) \quad \text{with} \quad \mathbb{E}[B_i^l] = 0 \quad \text{and} \quad \mathrm{Var}(B_i^l) = \sigma_b^2. \tag{6}$$

**Remark 1.** *Since the parameters are sampled independently with zero-mean, then:*

$$\mathbb{E}[Z_{j_1;a}^l Z_{j_2;a}^l] = v_a^l \delta_{j_1 j_2},$$

$$\frac{1}{\sqrt{v_a^l v_b^l}} \mathbb{E}[Z_{j_1;a}^l Z_{j_2;b}^l] = c_{ab}^l \delta_{j_1 j_2}.$$

*This is why Definitions (3) and (4) do not depend on $j$, and the crossed terms in $j_1$ and $j_2$ are not worth studying (they are zero).*

**Remark 2.** *The distributions of $W_{ij}^l$ and $B_i^l$ considered by Poole et al. (2016) and Schoenholz et al. (2017) are normal with zero-mean, that is:*

$$\mathrm{P}_w(\sigma_w) = \mathcal{N}(0, \sigma_w^2) \quad \text{and} \quad \mathrm{P}_b(\sigma_b) = \mathcal{N}(0, \sigma_b^2).$$

*We loosen this assumption in (5) and (6), where we assume that these random variables are zero-mean with a variance we can control. Their theoretical claim remain valid under this broader assumption.*

**Theoretical analysis.** Given two fixed inputs $\mathbf{x}_a$ and $\mathbf{x}_b$, the goal of the EOC theory is to study the propagation of the correlation $c_{ab}^l$ of $Z_{j;a}^l$ and $Z_{j;b}^l$. Poole et al. (2016) and Schoenholz et al. (2017) were the first to:

1. build recurrence equations for $(c_{ab}^l)_l$ of the form: $c_{ab}^{l+1} = f(c_{ab}^l)$;

2. describe the dynamics of $(c_{ab}^l)_l$;

3. provide a procedure to compute the variance of the weights' and biases' distributions such that $(c_{ab}^l)_l$ tends to 1 with a sub-exponential rate (instead of an exponential rate).

**Point 1: recurrence equations.** Point 1 is achieved by using the Gaussian hypothesis for the pre-activations. That is, the distribution of the pre-activations $\mathbf{Z}^l$ is assumed to be Gaussian, whatever the layer $l$ and its width $n_l$. The recurrence equations define a variance map $\mathcal{V}$ and a correlation map $\mathcal{C}$ as follows:

$$v_a^{l+1} = \mathcal{V}(v_a^l | \sigma_w, \sigma_b) := \sigma_w^2 \int \phi \left( \sqrt{v_a^l} z \right)^2 \mathcal{D}z + \sigma_b^2 \tag{7}$$

$$c_{ab}^{l+1} = \mathcal{C}(c_{ab}^l, v_a^l, v_b^l | \sigma_w, \sigma_b) := \frac{1}{\sqrt{v_a^{l+1} v_b^{l+1}}} \left[ \sigma_w^2 \int \phi \left( \sqrt{v_a^l} z_1 \right) \phi \left( \sqrt{v_b^l} z_2' \right) \mathcal{D}z_1 \mathcal{D}z_2 + \sigma_b^2 \right], \tag{8}$$

$$\text{where } z_2' := c_{ab}^l z_1 + \sqrt{1 - (c_{ab}^l)^2} z_2, \quad \text{and} \quad \mathcal{D}z := \frac{1}{\sqrt{2\pi}} \exp\left( -\frac{z^2}{2} \right) \mathrm{d}z. \tag{9}$$

These equations are approximations of the true information propagation dynamics, which involves necessarily the number of neurons $n_l$ per layer. Actually, passing a Gaussian vector $\mathbf{Z}^l$ through a layer with random Gaussian weights and biases produces a pre-activation $\mathbf{Z}^{l+1}$ with a distribution which is difficult to describe. On one side, as the dimension $n_l$ of the input $\mathbf{Z}^l$ tends to infinity, the Central Limit Theorem (CLT) applies, and the output $\mathbf{Z}^{l+1}$ tends to become Gaussian.[5] The assumption that the components of $\mathbf{Z}^{l+1}$ are Gaussian is referred to as the *Gaussian hypothesis*. On the other side, with finite $n_l$, the tail of the distribution of $\mathbf{Z}^{l+1}$ has been proven to become heavier than the Gaussian one, both theoretically (Vladimirova et al., 2019; 2021) and experimentally (see Section 2.2).

Finally, by assuming that $(v_a^l)_l$ and $(v_b^l)_l$ have already converged to the same limit $v^* \neq 0$ as $l \to \infty$, it is possible to rewrite Equation (8) in a nicer way:

$$c_{ab}^{l+1} = \mathcal{C}_*(c_{ab}^l) := \mathcal{C}(c_{ab}^l, v^*, v^* | \sigma_w, \sigma_b). \tag{10}$$

Now that the dynamics of $(c_{ab}^l)_l$ is written in the form $c_{ab}^{l+1} = \mathcal{C}_*(c_{ab}^l)$, it becomes sufficient to plot $\mathcal{C}_*$ to study its convergence. The two hypotheses made, i.e., the Gaussian one and the instant convergence of $(v_a^l)_l$ and $(v_b^l)_l$ to a unique nonzero limit $v^*$, are fundamental to obtaining the simple equation of evolution (10).

**Point 2: dynamics of the correlation through the layers.** Then, Point 2 can be achieved. Now that the trajectory of $(c_{ab}^l)_l$ is determined only by the function $\mathcal{C}_*$, it becomes easy to find numerically its limit $c^*$ and its rate of convergence. Specifically, we can distinguish three possible cases:

- *chaotic phase*: $\lim_{l \to \infty} c_{ab}^l = c^* < 1$. The correlation between $Z_{j;a}^l$ and $Z_{j;b}^l$ tends to a constant that is strictly less than 1. So, even if $Z_{j;a}^1$ and $Z_{j;b}^1$ are highly correlated (which means that $\mathbf{x}_a$ and $\mathbf{x}_b$ are close to each other), they tend to decorrelate when going deeper in the network;

---

[5]Even if the coordinates $Z_j^l$ are dependent, the CLT is still valid, as proven by Matthews et al. (2018) by using properties of exchangeable random variables (De Finetti, 1937).

- *ordered phase*: $\lim_{l \to \infty} c_{ab}^l = c^* = 1$ with $\mathcal{C}_*'(1) < 1$ (the prime here denotes the derivative of a function). The correlation between $Z_{j;a}^l$ and $Z_{j;b}^l$ tends to 1 with an exponential rate, including when $Z_{j;a}^1$ and $Z_{j;b}^1$ are almost fully decorrelated;

- *edge of chaos*: $\lim_{l \to \infty} c_{ab}^l = c^* = 1$ with $\mathcal{C}_*'(1) = 1$. The correlation between $Z_{j;a}^l$ and $Z_{j;b}^l$ tends to 1 with a sub-exponential rate, including when $Z_{j;a}^1$ and $Z_{j;b}^1$ are almost fully decorrelated.

**Point 3: best choices for the initialization distribution.** Poole et al. (2016) and Schoenholz et al. (2017) claim that pairs $(\sigma_w^2, \sigma_b^2)$ which lead either to the chaotic phase or the ordered phase should be avoided. In both cases, we expect that information contained in the propagated data (or the backpropagated gradients) would vanish at an exponential rate. So, we want to find pairs $(\sigma_w^2, \sigma_b^2)$ lying "at the edge of chaos", that is, making the sequence $(c_{ab}^l)_l$ converge to 1 at a sub-exponential rate. The *Edge of Chaos initializations* are the initialization distributions of the weights and the biases such that the pair of variances $(\sigma_w^2, \sigma_b^2)$ lies at the Edge of Chaos.

**Remark 3.** *Even in the favorable edge of chaos configuration, the sequence of correlations $(c_{ab}^l)_l$ tends to 1, whatever the data points $\mathbf{x}_a$ and $\mathbf{x}_b$. So a loss of information at initialization seems unavoidable in very deep networks.*

*If one wants to create an initialization procedure with a smaller information loss, it becomes reasonable to consider data-dependent initialization schemes (i.e., a warm-up phase before training). Such an initialization strategy has been sketched by Mao et al. (2021), who make use of the "Information Bottleneck" formalism (Tishby, 1999; Shwartz-Ziv & Tishby, 2017; Saxe et al., 2019).*

**Strengths and weaknesses of the Edge of Chaos framework.** One key feature of the Edge of Chaos framework is the simplicity of the recurrence equation (10): it involves only the correlation $c_{ab}^l$ as a variable, and all other parameters (such as $v_a^l$ and $v_b^l$) are assumed to be fixed once and for all. Notably, in Equation (8), the computation of $c_{ab}^{l+1}$ involves the distribution of the pre-activations outputted by layer $l$, which is assumed to be $\mathcal{D} = \mathcal{N}(0, 1)$. In other words, the distribution $\mathcal{D}^l$ of the pre-activations $\mathbf{Z}^l$ is assumed to be constant and equal to $\mathcal{D}$. However, in neural networks with finite widths, $\mathcal{D}^l$ is not constant and evolves according to a propagation equation:

$$\mathcal{D}^{l+1} \text{ is the distribution of } \mathbf{Z}^{l+1} = \frac{1}{\sqrt{n_l}} \mathbf{W}^l \phi(\mathbf{Z}^l) + \mathbf{B}^l, \text{ where } \mathbf{Z}^l \sim \mathcal{D}^l.$$

Such an equation involves a sum of products of random variables, which is usually difficult to keep track of.[6] Though Noci et al. (2021) have proposed an analytical procedure to compute the $\mathcal{D}^l$ explicitly, it works only for networks with ReLU or linear activation functions, and involves Meijer G-functions (Erdélyi et al., 1953), which are difficult to handle numerically.

---

[6]For instance, a product of two Gaussian random variables is not Gaussian.

## 2.2 Results on realistic datasets

As far as we know, there does not exist any experimental result about the propagation of the correlations with a non-synthetic dataset and a finite-width neural network. We propose to visualize in Figure 1 the propagation of correlation $c_{ab}^l$ with dataset CIFAR-10 (results on MNIST are reported in Appendix H.1), in the case of the multilayer perceptron with various numbers of neurons per layer $n_l$ (i.e., widths). Then, we show in Figure 2 the distance between the standardized distribution of the pre-activations and the standard Gaussian $\mathcal{N}(0, 1)$.

**Propagation of the correlations.** First, we have sampled randomly 10 data points in each of the 10 classes of the CIFAR-10 dataset, that is 100 in total for each dataset. Then, for each tested neural network (NN) architecture, we repeated $n_{\text{init}} = 1000$ times the following operation: (i) sample the parameters according to the EOC;[7] (ii) propagate the 100 data points in the NN. Thereafter, for each pair $(\mathbf{x}_a, \mathbf{x}_b)$ of the selected 100 data points, we have computed the empirical correlation $c_{ab}^l$ between the obtained pre-activations, averaged over the $n_{\text{init}}$ samples. Finally, we have averaged the results over the classes: the matrix $C_{pq}^l$ plotted in Figure 1 shows the mean of the correlation $c_{ab}^l$ for data points $\mathbf{x}_a$ and $\mathbf{x}_b$ belonging respectively to classes $p$ and $q$ in $\{0, \cdots, 9\}$.[8] Only the experiments with CIFAR-10 are reported in Figure 1; the results on MNIST, which are similar, are reported in Figure 14 in Appendix H.1.

In accordance with the theory of the EOC, we observe in Figure 1 that the average correlation between pre-activations tends to 1, except in the case $\phi = \text{ReLU}$ and $n_l = 10$. In this case, it is not even clear that the sequences of correlations $(C_{pq}^l)_l$ converge at all, since some inter-class correlations are lower at $l = 30$ than $l = 10$, while we expected them to grow until 1. There is also a difference between activation functions tanh and ReLU: the convergence to 1 seems to be much quicker with $\phi = \text{ReLU}$ than with $\phi = \tanh$, when $n_l = 100$.

We observe that the rate of convergence of $(C_{pq}^l)_l$ towards 1 not only varies with the NN width $n_l$ but also varies in different directions depending on the activation function. When $n_l$ grows from 10 to 100, the convergence of $(C_{pq}^l)_l$ to 1 slows down with $\phi = \tanh$, while it accelerates with $\phi = \text{ReLU}$. Since this striking inconsistency with the EOC theory is related to $n_l$, it is due to the "infinite-width" approximation, precisely made to eliminate the dependency on $n_l$ in the recurrence Equations (7) and (8), and consequently simplify them.

**Remark 4.** *According to the framework of the EOC, the inputs $\mathbf{x}_a$ and $\mathbf{x}_b$ are assumed to be fixed. So, it is improper to define a correlation $c_{ab}^0$ between $\mathbf{x}_a$ and $\mathbf{x}_b$. However, when considering the correlation $c_{ab}^1$ of the pre-activations right after the first layer, the empirical*

---

[7]For the tanh activation function, a study of the EOC can be found in Poole et al. (2016). For ReLU, the EOC study is more subtle and can be found in Hayou et al. (2019).

[8]Correlations $c_{ab}^l$ with $a = b$ have been excluded from the computation to show the intra-class correlation between *different* samples.

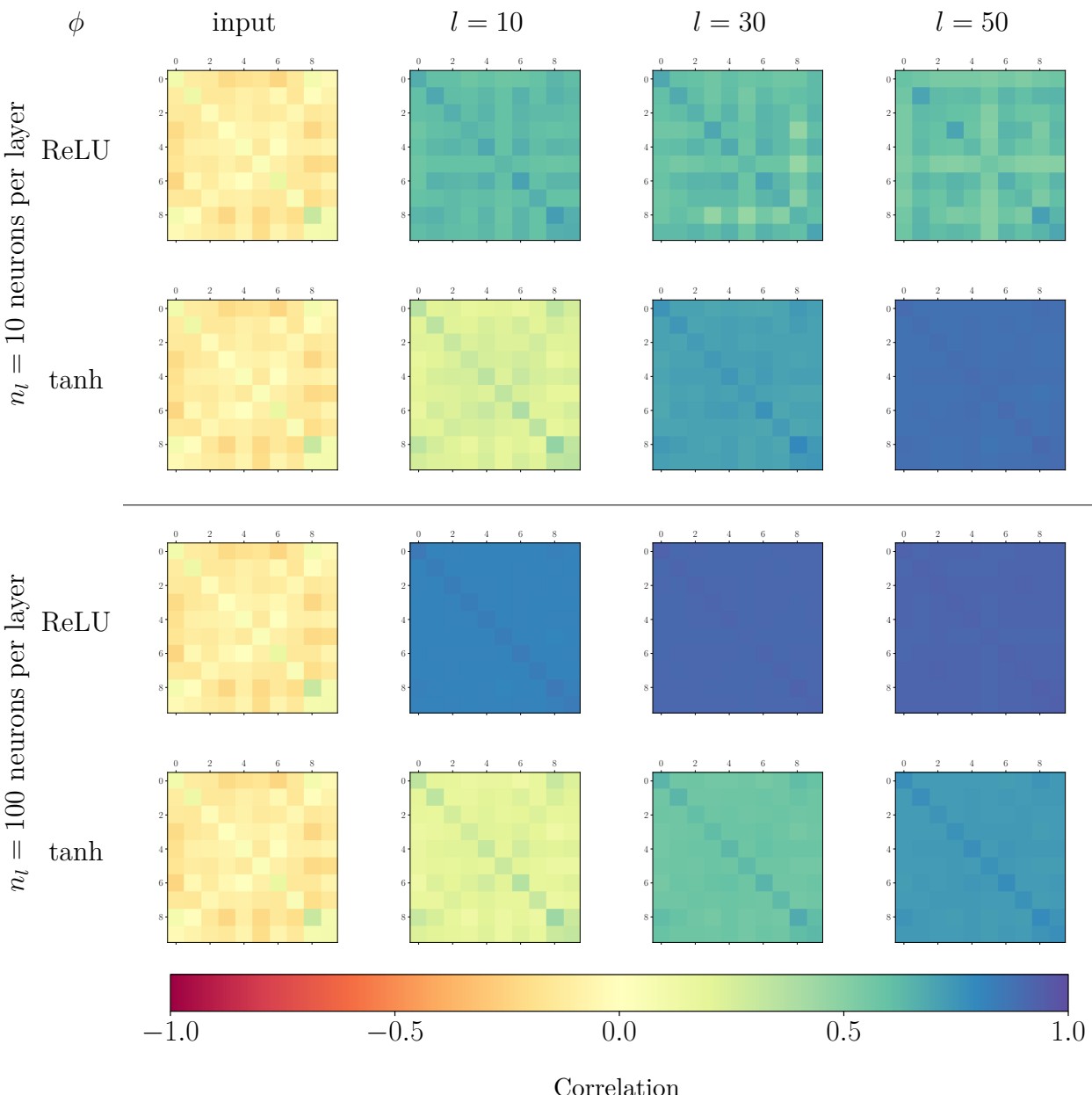

Figure 1 – Propagation of correlations $c_{ab}^l$ in a multilayer perceptron with activation function $\phi \in \{\tanh, \mathrm{ReLU}\}$ and inputs sampled from the CIFAR-10 dataset. The NN is initialized at the EOC. Each plot displays a $10 \times 10$ matrix $C_{pq}^l$ whose entries are the average correlation between the pre-activations propagated by samples from classes $p, q \in \{0, \cdots, 9\}$, at the input and right after layers $l \in \{10, 30, 50\}$. See Fig. 14, App. H.1, for results on MNIST.

*correlation between inputs* $\mathbf{x}_a$ *and* $\mathbf{x}_b$ *appears naturally:*

$$c_{ab}^1 = \sigma_w^2 \cdot \frac{\mathbf{x}_a^T \mathbf{x}_b}{n} + \sigma_b^2,$$

*where* $\hat{c}_{ab}^0 := \mathbf{x}_a^T \mathbf{x}_b / n$ *plays the role of an empirical correlation between* $\mathbf{x}_a$ *and* $\mathbf{x}_b$, *assuming that the empirical mean and variance of both* $\mathbf{x}_a$ *and* $\mathbf{x}_b$ *are respectively* 0 *and* 1.[9]

**Propagation of the distances to the Gaussian distribution.** We test in Figure 2 the Gaussian hypothesis in a multilayer perceptron of $L = 100$ layers, with a constant width $n_l \in \{10, 100, 1000\}$, and an activation function $\phi \in \{\mathrm{ReLU}, \tanh\}$. We propagate a single point sampled from the CIFAR-10 dataset, and we compute the empirical distribution of the pre-activations by drawing 10000 samples of the parameters of the neural network.

In Figure 2, we plot the Kolmogorov–Smirnov statistic of the standardized pre-activations for each layer, and we compare it to a threshold corresponding to a $p$-value of 0.05. Thus, according to Figure 2, the Gaussian hypothesis is rejected with a $p$-value of 0.05 for the ReLU activation function in all considered setups ($n_l \in \{10, 100, 1000\}$), and it is rejected too for the tanh activation function in the narrow network setup ($n_l = 10$).

We provide all the details about the Kolmogorov–Smirnov test in Appendix G, and additional details and experiments in the multilayer perceptron setup in Section 4.2.

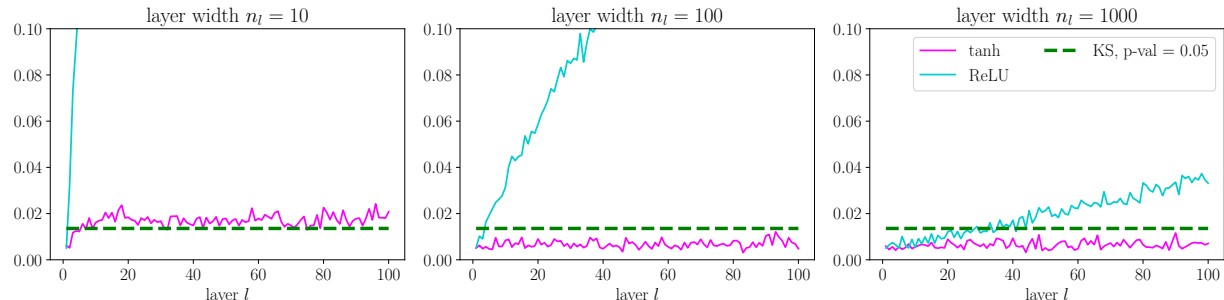

Figure 2 – Evolution of the $\mathcal{L}^\infty$-distance of the standardized empirical CDF of the pre-activations and the standardized Gaussian $\mathcal{N}(0, 1)$, for various layer widths and activation functions. The empirical CDF of $\mathcal{D}^l$ has been computed with 10000 samples. The green dotted line corresponds to the threshold given by the Kolmogorov–Smirnov test with a $p$-value of 0.05: any point above it corresponds to a distribution for which the Gaussian hypothesis should be rejected with a $p$-value of 0.05.

---

[9]Usually, this assumption does not hold exactly: it is common to normalize the entire dataset in such a way that the whole set of the features of all training data points has empirical mean 0 and variance 1, but not each data point individually. See also Definitions 5 and 6.

## 2.3 Convergence of the sequence of variances $(v_a^l)_l$

**Multiple stable limits.** As reminded in Point [1], Section 2.1, it is a key assumption of the EOC formalism to assume that, whatever the starting point $v_a^0$, the sequence $(v_a^l)_l$ converges to the same limit $v^*$ as $l \to \infty$. As far as we know, no configuration where the map $\mathcal{V}$ has two nonzero stable points or more has been encountered in past works. Moreover, it is believed that such configurations do not exist, as stated for example by Poole et al. (2016): "for monotonic nonlinearities $[\phi]$, this length map $[\mathcal{V}]$ is a monotonically increasing, concave function whose intersections with the unity line determine its fixed points."

In this subsection, we build a *monotonic* activation function for which the $\mathcal{V}$ map is not concave and admits an *infinite* number of stable fixed points.

**Definition 1** (Activation function $\varphi_{\delta,\omega}$). *For $\delta \in [0,1]$ and $\omega > 0$ two real numbers, define:*

$$\varphi_{\delta,\omega}(x) := x \exp\left( \frac{\delta}{\omega} \sin(\omega \ln |x|) \right).$$

It is easy to prove that for all $\delta \in [0,1]$ and $\omega > 0$:

1. $\varphi_{\delta,\omega}(0) = 0$, by continuity;

2. $\varphi_{\delta,\omega}$ is odd;

3. $\varphi_{\delta,\omega}$ is strictly increasing;

4. the map[10] $v \mapsto \int \varphi_{\delta,\omega}(\sqrt{v}z)^2 \, \mathcal{D}z$ is $\mathcal{C}^1$ and strictly increasing.

**Definition 2** (Stable fixed points). *Let $(u_n)_n$ be a sequence defined by recurrence:*

$$u_{n+1} = f(u_n) \quad with \quad u_0 \in \mathbb{R}.$$

*For any starting point $a$, we denote by $(u_n(a))_n$ the sequence defined as above, with $u_0 = a$.*

*We say that $u^*$ is a* stable fixed point *of $f$ if $f(u^*) = u^*$ and if there exists an open ball $\mathcal{B}(u^*, \epsilon)$ centered in $u^*$ of radius $\epsilon > 0$ such that:*

$$\forall a \in \mathcal{B}(u^*, \epsilon), \quad u_n(a) \xrightarrow[n \to \infty]{} u^*.$$

**Proposition 1.** *If $f(u^*) = u^*$ and $f$ is $\mathcal{C}^1$ in a neighborhood of $u^*$ with $f'(u^*) \in (-1, 1)$, then $u^*$ is a stable fixed point of $f$.*

**Proposition 2.** *For any $\delta \in (0,1]$ and $\omega > 0$, let us use the activation function $\phi = \varphi_{\delta,\omega}$ of Definition 1. We consider the sequence $(v^l)_l$ defined by:*

$$\forall l \geq 0, \quad v^{l+1} = \sigma_w^2 \int \varphi_{\delta,\omega}\left(\sqrt{v^l}z\right)^2 \mathcal{D}z + \sigma_b^2, \quad with \quad v^0 \in \mathbb{R}_*^+. \tag{11}$$

*Then there exist $\sigma_w > 0$, $\sigma_b \geq 0$, and a strictly increasing sequence of stable fixed points $(v_k^*)_{k \in \mathbb{Z}}$ of the recurrence equation (11).*

---

[10]Notation $\mathcal{D}z$ is defined in Eqn. (9).

The proof can be found in Appendix A.1.

**Remark 5.** *In short, Proposition 2 ensures that there exists an infinite number of possible (nonzero) limits for the sequence $(v_a^l)_l$, depending on $v_a^0$. Consequently, the proposed activation functions $\varphi_{\delta,\omega}$ are counterexamples to the claim of Poole et al. (2016).*

**Plots.** Figure 3 shows the shape of several activation functions $\varphi_{\delta,\omega}$ for various $\omega$, along with their $\mathcal{V}$ maps. We have chosen $\delta = 0.99 < 1$ to ensure that $\varphi_{\delta,\omega}$ is a strictly increasing function.[11] We have chosen $\sigma_b^2 = 0$ and $\sigma_w^2 = \sigma_\omega^2$, computed as indicated in Appendix A.2.

In Figure 3a, the proposed activation functions exhibit reasonable properties: they are non-linear, differentiable at each point (excluding 0), and remain dominated by a linear function. However, we expect that as $\omega$ grows, $\varphi_{\delta,\omega}$ should become closer and closer to the identity function,[12] which is not desirable for the activation function of a NN.

In Figure 3b, it is clear that the function $\varphi_{\delta,\omega}$ with $\omega = 6$ is a counterexample to the claim of Poole et al. (2016): two nonzero stable points appear. So, in that case, depending on the square norm $v_a^0$ of the input, the variance $v_a^l$ of the pre-activations may converge to different values. For instance, for $\omega = 6$ and an input with square norm $v_a^0$ around the unstable point at $v \approx 2.3$, it may converge either to $v^* \approx 0.8$ or $v^* \approx 6.5$.

Also, we observe that for $\omega = 6$, the variance map $\mathcal{V}$ tends to be closer to the identity function than for smaller $\omega$. Thus, we expect the sequence $(v_a^l)_l$ to converge at a slower rate with $\omega = 6$ than with $\omega = 2$.

In Figure 3d, all the configurations lie in the chaotic phase. Since all the correlation maps $\mathcal{C}$ are below the identity function, the sequence of correlations $(c_{ab}^l)_l$ always tends to 0. However, the plots are close to the identity, so $(c_{ab}^l)_l$ varies very slowly, and we expect that the correlation between data points propagates into the NN with little deformation. Despite being not perfect and lying in the chaotic phase, this configuration roughly preserves the input correlations between data points (without performing a pre-training phase), which is a desirable property at initialization: information propagates with little deformation to the output, and the error can be backpropagated to the first layers.

## 2.4 Maintaining a property of the pre-activations during propagation

To conclude this section and introduce the next one, we propose a common representation of the various methods used to build initialization distributions for the weights $(\mathbf{W}^l)_l$ and biases $(\mathbf{B}^l)_l$.

Several initialization methods (Glorot & Bengio, 2010; He et al., 2015; Poole et al., 2016; Schoenholz et al., 2017) are based on the same principle: initialization should be done in such a way that some characteristic $\kappa^l$ of the distribution of $\mathbf{Z}^l$ is preserved during propagation (e.g., $\kappa^l = \mathrm{Var}(\mathbf{Z}^l)$). Intuitively, any change between $\kappa^l$ and $\kappa^{l+1}$ reflects a loss of information

---

[11] We have chosen $\delta$ close to 1 to obtain a function $\varphi_{\delta,\omega}$ that is strongly nonlinear, but a bit lower than 1 to ensure that $\varphi'_{\delta,\omega}$ remains strictly positive, in order to prevent the training process from being stuck.

[12] As $\omega \to \infty$, $\varphi_{\delta,\omega}$ converges pointwise to the identity function.

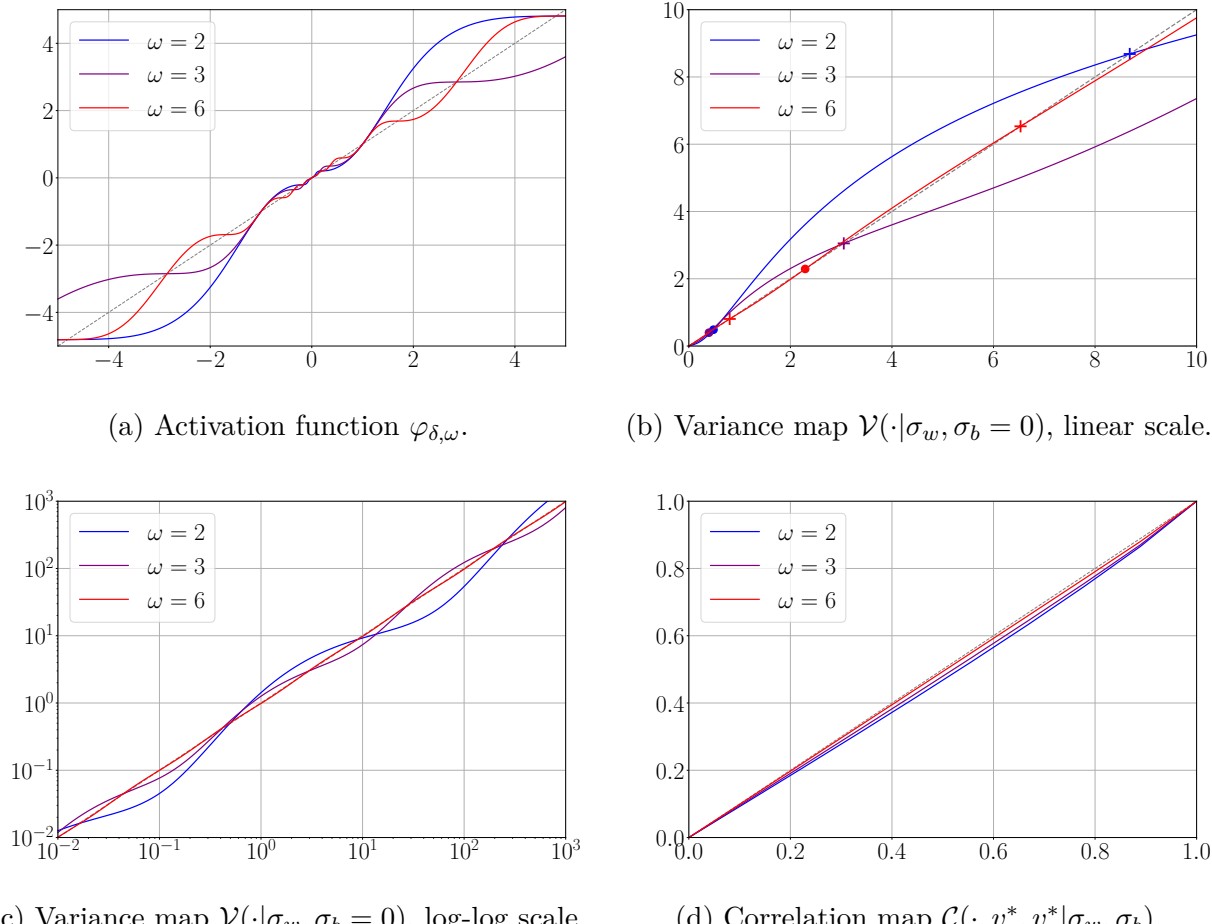

(a) Activation function $\varphi_{\delta,\omega}$.

(b) Variance map $\mathcal{V}(\cdot|\sigma_w, \sigma_b = 0)$, linear scale.

(c) Variance map $\mathcal{V}(\cdot|\sigma_w, \sigma_b = 0)$, log-log scale.

(d) Correlation map $\mathcal{C}(\cdot, v^*, v^*|\sigma_w, \sigma_b)$.

Figure 3 – Properties of the proposed counterexamples $\varphi_{\delta,\omega}$ represented in Fig. 3a for $\omega \in \{2, 3, 6\}$ and $\delta = 0.99$.

In Fig. 3b, stable points are marked by crosses (+), and unstable points by bullets (•), when they are away from 0: two stable points appear for $\omega = 6$ (in red). As established in Proposition 2, $\sigma_w$ is tuned for every $\omega$ in such a way that $\mathcal{V}$ crosses the identity function an infinite number of times (not visible on the figure).

In Fig. 3c (log-log scale), it is clearer that $\mathcal{V}$ has an infinite number of fixed points, due to regular oscillations (in log-log scale) below and above the identity.

In Fig. 3d, we show that as $\omega$ grows, the correlation map $\mathcal{C}$ becomes closer to the identity function, which means that the correlation between data points tends to propagate perfectly. Note: since an infinite number of stable fixed points are available, we have arbitrarily picked one for each $\omega$, denoted by $v^*$. This choice does not affect the plot of the correlation map $\mathcal{C}$, due to the very specific structure of $\varphi_{\delta,\omega}$.

between $\mathbf{Z}^l$ and $\mathbf{Z}^{l+1}$, which damages propagation or backpropagation. For instance, when $\mathrm{Var}(\mathbf{Z}^l) \to 0$, the network output tends to become deterministic, and when $\mathrm{Var}(\mathbf{Z}^l) \to \infty$, the output tends to forget the operations made by the first layers (i.e., the gradients vanish during backpropagation).

More generally, we denote by $\mathcal{D}^l$ the distribution associated to the pre-activations $\mathbf{Z}^l$, by $\mathcal{T}(\cdot; n_l, \mathrm{P}_l, \phi^l) =: \mathcal{T}_l[\mathrm{P}_l](\cdot)$ the transformation of $\mathcal{D}^l$ performed by layer $l$, that is $\mathcal{D}^{l+1} = \mathcal{T}_l[\mathrm{P}_l](\mathcal{D}^l)$ (where $\mathrm{P}_l$ is the initialization distribution of $(\mathbf{W}^l, \mathbf{B}^l)$ and $\phi^l$ is the activation function at layer $l$), and by $\kappa^l := \chi(\mathcal{D}^l)$ the characteristic of the distribution $\mathcal{D}^l$ we are interested in. Then, according to a heuristic of "information preservation", it is assumed that the sequence $(\kappa^l)_l$ must remain constant, and the initialization distributions $(\mathrm{P}_l)_l$ are built accordingly. In some cases, it is possible to build a map $\tilde{\mathcal{T}}_l[\mathrm{P}_l]$, so that each $\kappa^{l+1}$ can be built out of its predecessor $\kappa^l$, without using all the information we may have on the $(\mathcal{D}^l)_l$.

We summarize this way of building initialization procedures in Figure 4, and we show how it applies to well-known initialization procedures in Table 1.

Table 1 – Examples of $\mathcal{D}^l$, $\phi^l$ and $\chi$ in various setups. Notations: for any vector $\mathbf{x} \in \mathbb{R}^n$, its empirical mean is $\bar{\mathbb{E}}\,\mathbf{x} = \frac{1}{n}\sum_{i=1}^n x_i$ and is empirical variance is $\overline{\mathrm{Var}}\,\mathbf{x} = \frac{1}{n-1}\sum_{i=1}^n (x_i - \bar{\mathbb{E}}\,\mathbf{x})^2$.

| Method | $\mathcal{D}^l$ | $\phi^l$ | $\chi$ | $\tilde{\mathcal{T}}_l[\mathrm{P}_l](\kappa)$ | Assumption |
|---|---|---|---|---|---|
| Glorot & Bengio | distr. of $\mathbf{Z}^l$ | Id | Var | $\sigma_w^2 \kappa^2 + \sigma_b^2$ | – |
| He et al. | distr. of $\mathbf{Z}^l$ | ReLU | Var | $\frac{1}{2}\sigma_w^2 \kappa^2 + \sigma_b^2$ | – |
| Poole, Schoenholz | distr. of $(\mathbf{Z}_a^l, \mathbf{Z}_b^l)$ | $\phi$ | corr | $\mathcal{C}_*(\kappa)$ | $Z_{j;a}^l,\ Z_{j;b}^l$ Gaussian, $v_a^l = v_b^l = v^*$ |
| Ours | distr. of $\mathbf{Z}^l$ | $\phi_\theta$ | Id | $\mathcal{T}_l[\mathrm{P}_l](\kappa)$ | $\bar{\mathbb{E}}\,\mathbf{x}_a = \bar{\mathbb{E}}\,\mathbf{x}_b = 0$, $\overline{\mathrm{Var}}\,\mathbf{x}_a = \overline{\mathrm{Var}}\,\mathbf{x}_b = 1$ |

$$\cdots \xrightarrow{\mathcal{T}_{l-1}[\mathrm{P}_{l-1}]} \mathcal{D}^l \xrightarrow{\mathcal{T}_l[\mathrm{P}_l]} \mathcal{D}^{l+1} \xrightarrow{\mathcal{T}_{l+1}[\mathrm{P}_{l+1}]} \cdots$$
$$\Big\downarrow \chi \qquad\qquad \Big\downarrow \chi$$
$$\cdots \dashrightarrow{\tilde{\mathcal{T}}_{l-1}[\mathrm{P}_{l-1}]} \kappa^l \dashrightarrow{\tilde{\mathcal{T}}_l[\mathrm{P}_l]} \kappa^{l+1} \dashrightarrow{\tilde{\mathcal{T}}_{l+1}[\mathrm{P}_{l+1}]} \cdots$$

Figure 4 – Building process of the initialization distributions $\mathrm{P}_l$ of the parameters $(\mathbf{W}^l, \mathbf{B}^l)$: (i) the pre-activations-related distribution $\mathcal{D}^l$ passes through a map $\mathcal{T}_l[\mathrm{P}_l]$ and becomes $\mathcal{D}^{l+1}$; (ii) some statistical characteristic $\kappa^l$ of $\mathcal{D}^l$ can be computed with a function $\chi$: $\kappa^l = \chi(\mathcal{D}^l)$; (iii) we tune the $(\mathrm{P}_l)_l$ in order to make the sequence $(\kappa^l)_l$ constant.

**Remark 6.** *We can use Figure 4 to build new initialization distributions: first, we choose a statistical property of $\mathbf{Z}^l$, which determines $\mathcal{D}^l$ and $\chi$; then, we build a framework in which*

$\chi(\mathcal{D}^l)$ *can be easily computed for every l (e.g., we choose a specific activation function, or we make simplifying assumptions).*

In the following section, we aim to impose Gaussian pre-activations through a specific activation function $\phi_\theta$ and initialization distribution $P_\theta$. It implies that we would preserve perfectly the distribution $\mathcal{D}^l$ itself: our characteristic is $\chi(\mathcal{D}^l) = \mathcal{D}^l$. That way, *all the statistical properties of $\mathcal{D}^l$ are preserved* during propagation.

## 3   Imposing Gaussian pre-activations

In this section, we propose a family of pairs $(P_\theta, \phi_\theta)$, where $P_\theta$ is the distribution of the weights at initialization, $\phi_\theta$ is the activation function, and $\theta \in (2, \infty)$ is a parameter, such that the pre-activations $Z_j^l$ are $\mathcal{N}(0,1)$ at any layer $l$. Imposing such pre-activations is a way to meet two goals.

First, in Section 2.2, we have shown that the Gaussian hypothesis is not fulfilled in the case of realistic datasets propagated into a simple multilayer perceptron, and we have recalled in the Introduction that the tails of the pre-activations tend to become heavier when information propagates in a neural network. By imposing Gaussian pre-activations, we ensure that the Gaussian hypothesis is true, which reconciles the results provided in the EOC setup (see Eqn. (7) and (8)) and the experiments.[13]

Second, as we recalled in the Introduction, many initialization procedures are based on the preservation of some characteristic of the distribution of the pre-activations (see Table 1 and Figure 4). Usual characteristics are the variance and the correlation between data points. By imposing Gaussian pre-activations, we would ensure that *the whole* distribution is propagated, and not only one of its characteristics.

Besides, we provide a set of constraints, Constraints 1, 2, 3, and 4, that the activation function and the initialization procedure should fulfill in order to maintain Gaussian pre-activations at each layer.

**Summary.**  Formally, we aim to find a family of pairs $(P_\theta, \phi_\theta)$ such that:

$$\left. \begin{array}{ll} Z_j^l \sim \mathcal{N}(0,1) & \text{i.i.d.} \\ W_{ij}^l \sim P_\theta & \text{i.i.d.} \\ B_i^l = 0 & \end{array} \right\} \Rightarrow Z_i^{l+1} := \frac{1}{\sqrt{n_l}} \mathbf{W}_{i\cdot}^l \phi_\theta(\mathbf{Z}^l) + \mathbf{B}^l \sim \mathcal{N}(0,1),$$

where $\mathbf{W}_{i\cdot}^l$ is the $i$-th row of the matrix $\mathbf{W}^l$. In other words, the pre-activations $Z_i^{l+1}$ remain Gaussian for all $l$.

As a result of the present section, we make the following proposition for $(P_\theta, \phi_\theta)$:

---

[13]There exists another way to solve this problem: use propagation equations which would take into account the sequence $(n_l)_l$ of layer widths, that is, adopt a non-asymptotic setup, contrary to the process leading to Eqn. (7) and (8). However, taking into account the whole sequence $(n_l)_l$ would lead to recurrence equations that are far less easy to use than Eqn. (10). Moreover, a precise characterization of the distributions $(\mathcal{D}^l)_l$ of the pre-activations $(Z_1^l)_l$ may be very difficult since they would not be Gaussian anymore.

- $P_\theta$ is the symmetric Weibull distribution $\mathcal{W}(\theta, 1)$, with CDF:

$$F_W(t) = \frac{1}{2} + \frac{1}{2}\operatorname{sgn}(t)\exp\left(-|t|^\theta\right);$$

(12)

- $\phi_\theta$ is computed to ensure that $Z_i^{l+1} := \frac{1}{\sqrt{n_l}}\mathbf{W}_{i\cdot}^l\phi_\theta(\mathbf{Z}^l) + \mathbf{B}^l$ is Gaussian $\mathcal{N}(0, 1)$. In short, the family $(\phi_\theta)_\theta$ spans a range of functions from a tanh-like function (as $\theta \to 2^+$) to the identity function (as $\theta \to \infty$).

In order to obtain this result, we:

1. reduce and decompose the initial problem (Section 3.1);

2. find constraints on the initialization distribution of the parameters to justify our choice $P_\theta = \mathcal{W}(\theta, 1)$ (Section 3.2);

3. compute the distribution $Q_\theta$ of $\phi_\theta(Z_j^l)$ we must choose to ensure Gaussian pre-activations $Z_i^{l+1}$, given an initialization distribution $P_\theta$ (Section 3.3);

4. build $\phi_\theta$ from $Q_\theta$ (Section 3.4).

## 3.1  Decomposing the problem

By combining Equations (1) and (2), the operation performed by each layer is:

$$\mathbf{Z}^{l+1} = \frac{1}{\sqrt{n_l}}\mathbf{W}^l\phi(\mathbf{Z}^l) + \mathbf{B}^l.$$

(13)

In this subsection, we show that finding the distribution of the weights $\mathbf{W}^l$ and the activation function $\phi$ in order to have:

$$\forall l \in [1, L], \forall j \in [1, n_l], \quad Z_j^l \sim \mathcal{N}(0, 1),$$

can be done if we manage to get:

$$Z := W\phi(X) \sim \mathcal{N}(0, 1), \quad \text{with } X \sim \mathcal{N}(0, 1),$$

by tuning the distribution of $W$ and the activation function $\phi$.

$Z_i^{l+1}$ **as a sum of Gaussian random variables.** First, we focus on the operation made by one layer: if each layer transforms Gaussian inputs $Z_j^l$ into pre-activations $Z_i^{l+1}$ that are Gaussian too, then we can ensure that the pre-activations remain Gaussian after each layer. Thus, it is sufficient to solve the problem for one layer. After renaming the variables as $Z \leftarrow Z_i^{l+1}, W_j \leftarrow W_{ij}^l, X_j \leftarrow Z_j^l, B \leftarrow B_i^l, n \leftarrow n_l$, we have:

$$Z = \frac{1}{\sqrt{n}}\sum_{j=1}^{n}W_j\phi(X_j) + B.$$

(14)

In the rest of this subsection, we assume that the $\mathcal{N}(0,1)$ $X_j$ are independent. We discuss the independence hypothesis in Remark 7 and Appendix B. We want to build an activation function $\phi$, and distributions for $(W_j)_j$ and $B$ such that $Z \sim \mathcal{N}(0,1)$.

Second, we narrow our search space. According to Equation (14), $Z$ is the sum of a random variable $B$ and a number $n$ of i.i.d. random variables $W_j \phi(X_j)/\sqrt{n}$. Since $Z$ must be $\mathcal{N}(0,1)$ whatever the value of $n$, it is both convenient and sufficient to check that each summand in the right-hand side of Equation (14) is Gaussian, that is:

$$B \sim \mathcal{N}(0, \sigma_b^2), \quad W_j \phi(X_j) \sim \mathcal{N}(0, 1 - \sigma_b^2),$$

with $\sigma_b \in (0,1)$. In that case, we have $Z \sim \mathcal{N}(0,1)$. For the sake of simplicity, we assume that $B = 0$ with probability 1, so that we just have to ensure that, for all $j$, $W_j \phi(X_j) \sim \mathcal{N}(0,1)$. If one wants to deal with nonzero bias $B \sim \mathcal{N}(0, \sigma_b^2)$, it is sufficient to scale the random variables $W_j \phi(X_j)$ accordingly.

To summarize, we have chosen to build $Z \sim \mathcal{N}(0,1)$ by ensuring that $W_j \phi(X_j) \sim \mathcal{N}(0,1)$. With $B = 0$, this choice is formally imposed by this straightforward proposition.

**Proposition 3.** *Let $(Z_j)_j$ be a sequence of $n$ i.i.d. random variables. Let $Z = \frac{1}{\sqrt{n}} \sum_{j=1}^{n} Z_j$. If $Z$ is $\mathcal{N}(0,1)$, then the distribution of each $Z_j$ is also $\mathcal{N}(0,1)$.*

*Proof.* Let $\psi_Z(x) := \mathbb{E}[e^{iZx}]$ be the characteristic function of the distribution of $Z$. Besides, the $(Z_j)_j$ are i.i.d. and $Z = \frac{1}{\sqrt{n}} \sum_{j=1}^{n} Z_j$, so:

$$\psi_Z(x) = \exp\left(-\frac{x^2}{2}\right) \quad \text{and} \quad \psi_Z(x) = \left[\psi_{\frac{Z_1}{\sqrt{n}}}(x)\right]^n.$$

This proves that $\psi_{Z_1}(x) = e^{-x^2/2}$. So, for all $j$ in $[1, n]$, $Z_j \sim \mathcal{N}(0,1)$. $\qquad\square$

As a result, we obtain the first constraint.

**Constraint 1.** *If, for all $l$, the weights $(W_{ij}^l)_{ij}$ are i.i.d. and independent from the pre-activations $(X_j^l)_j$, which are also supposed to be i.i.d., then we must ensure that:*

$$\forall l, i, j, \quad W_{ij}^l \phi(X_j^l) \sim \mathcal{N}(0,1).$$

**Remark 7.** *The hypothesis of independent inputs $(X_j^l)_j$ truly holds only for the second layer.[14] But, overall, the hypothesis of independent $(X_j^l)_j$ is unrealistic. So, we propose in Appendix B an empirical study of this hypothesis, in order to identify in which cases the dependence between the inputs of one layer damages the Gaussianity of its outputted pre-activations.*

---

[14]The inputs of the first layer are deterministic.

**New formulation of the problem.** We have proven that, to ensure that $Z \sim \mathcal{N}(0, 1)$, it is sufficient to solve the following problem:

$$\text{find P and } \phi \text{ such that:} \quad X \sim \mathcal{N}(0, 1) \text{ and } W \sim \text{P} \Rightarrow W\phi(X) \sim \mathcal{N}(0, 1). \qquad (15)$$

In the following subsections, we build a family $\mathcal{P}$ of initialization distributions (Section 3.2) such that, for any $\text{P} \in \mathcal{P}$, there exists a function $\phi$ such that $(\text{P}, \phi)$ is a solution to (15). We decompose the remaining problem into two parts, by introducing an intermediary random variable $Y = \phi(X)$:

- for a distribution P, deduce Q s.t.: $W \sim \text{P}, Y \sim \text{Q} \quad \Rightarrow \quad WY =: G \sim \mathcal{N}(0, 1)$ (Section 3.3);

- for a distribution Q, find a function $\phi$ s.t.: $X \sim \mathcal{N}(0, 1) \quad \Rightarrow \quad Y = \phi(X) \sim \text{Q}$ (Section 3.4).

### 3.2 Why initializing the weights $W$ according to a symmetric Weibull distribution?

We are looking for a family $\mathcal{P}$ of distributions such that, for any $\text{P} \in \mathcal{P}$, there exists Q such that:

$$W \sim \text{P}, Y \sim \text{Q} \quad \Rightarrow \quad WY =: G \sim \mathcal{N}(0, 1).$$

Therefore, the family $\mathcal{P}$ is subject to several constraints. In this subsection, we present two results indicating that a subset of the family of Weibull distributions is a good choice for $\mathcal{P}$:

1. the density of $W$ at 0 should be 0;

2. $W$ should be a generalized Weibull-tail random variable (see Section 3.2.2 or Vladimirova et al. (2021)) with parameter $\theta \in (2, \infty)$.

In the process, we are able to gather information about the distribution of $|Y|$, namely its density at 0 and the leading power of the log of its survival function at infinity, respectively:

$$f_{|Y|}(0) = \sqrt{\frac{2}{\pi}} \left[ \int_0^\infty \frac{f_{|W|}(t)}{t} \, \mathrm{d}t \right]^{-1},$$

$$\log S_{|Y|}(y) \propto -y^{1/\left(\frac{1}{2} - \frac{1}{\theta}\right)}.$$

As a conclusion of this subsection, we consider that the distribution $\text{P} = \text{P}_\theta$ of $W$ should lie in the following subset of the family of symmetric Weibull distributions (defined at Eqn. (12)):

$$\mathcal{P} := \{\mathcal{W}(\theta, 1) : \theta \in \Theta\},$$
$$\Theta := (2, \infty).$$

### 3.2.1 Behavior near $0$

Since the product $G = WY$ is meant to be distributed according to $\mathcal{N}(0, 1)$, then we must have $f_{|G|}(0) = \sqrt{\frac{2}{\pi}} \in (0, \infty)$, which is impossible for several choices of distributions for $W$.

**Proposition 4** (Density of a product of random variables at 0)**.** *Let $W, Y$ be two independent non-negative random variables and $Z = WY$. Let $f_W, f_Y, f_Z$ be their respective densities. Assuming that $f_Y$ is continuous at 0 with $f_Y(0) > 0$, we have:*

$$if \quad \lim_{w \to 0} \int_w^\infty \frac{f_W(t)}{t} \, dt = \infty, \quad then \quad \lim_{z \to 0} f_Z(z) = \infty.$$

*Moreover, if $f_Y$ is bounded:*

$$if \quad \int_0^\infty \frac{f_W(t)}{t} \, dt < \infty, \quad then \quad f_Z(0) = f_Y(0) \int_0^\infty \frac{f_W(t)}{t} \, dt. \tag{16}$$

The proof can be found in Appendix C.

**Corollary 1.** *If $f_Y$ and $f_W$ are continuous at 0 with $f_Y(0) > 0$ and $f_W(0) > 0$, then:*

$$\lim_{z \to 0} f_Z(z) = \infty.$$

According to Corollary 1, it is impossible to obtain a Gaussian $G$ by multiplying two random variables $W$ and $Y$ whose densities are both continuous and nonzero at 0. So, if we want to manipulate continuous densities, we must have either $f_W(0) = 0$ or $f_Y(0) = 0$.

Let us assume that $f_Y(0) = 0$. We want $Y$ to be the image of $X \sim \mathcal{N}(0, 1)$ through the function $\phi$, where $f_X(0) > 0$. So, in order to obtain $Y$ with a zero density at 0, it is necessary to build a function $\phi$ with $\phi'(0) = \infty$ (see Lemma 2 in Appendix D), which is usually not desirable for an activation function of a neural network for training stability reasons.[15] So, it is preferable to design $W$ such that $f_W(0) = 0$.

---

**Constraint 2.** *To avoid activation functions with a vertical tangent at 0, the density of the initialization distribution of a weight $W_{ij}^l$ must be 0 at 0:*

$$\forall l, i, j, \quad f_{W_{ij}^l}(0) = 0.$$

---

**Remark 8.** *In the common case of neural networks with activation function $\phi = \tanh$ and weights $W$ initialized according to a Gaussian distribution, if we assume that the Gaussian hypothesis is true, then $f_Y(0) > 0$ and $f_Z(0) > 0$. Thus, Corollary 1 applies and the density of $Z$ is infinite at 0.*

---

[15]If $\phi'(0) = \infty$ and $\phi$ is $\mathcal{C}^1$ on $\mathbb{R}^*$, then numerical instabilities may occur during training: if a pre-activation $Z_j^l$ approaches 0 too closely, $\phi'(Z_j^l)$ can explode and damage the training. These instabilities can be handled by gradient clipping (Pascanu et al., 2013).

*If $\phi = \text{Id}$, $Z$ is the product of two independent $\mathcal{N}(0,1)$, whose density is well-known:*

$$f_Z(z) = \frac{K_0(|z|)}{\pi},$$

*where $K_0$ is the modified Bessel function of the second kind, which tends to infinity at $0$, which illustrates Corollary 1.*[16]

Finally, if Constraint 2 holds and we want $f_{|G|}(0) = \sqrt{\frac{2}{\pi}}$, then, according to Equation (16), the following constraint must hold.

> **Constraint 3.** *The density of $Y$ at $0$ must have a specific value depending on the distribution of $W$:*
>
> $$f_{|Y|}(0) = \sqrt{\frac{2}{\pi}} \left[ \int_0^\infty \frac{f_{|W|}(t)}{t} \, \mathrm{d}t \right]^{-1}.$$

### 3.2.2 Behavior of the tail

We use the results of Vladimirova et al. (2021) on the "generalized Weibull-tail distributions" and start by recalling useful definitions and properties.

**Definition 3** (Slowly varying function)**.** *A measurable function $f : (0, \infty) \to (0, \infty)$ is said to be* slowly varying *if:*

$$\forall a > 0, \quad \lim_{x \to \infty} \frac{f(ax)}{f(x)} = 1.$$

**Definition 4** (Generalized Weibull-Tail (GWT) distribution)**.** *A random variable $X$ is called* generalized Weibull-tail *with parameter $\theta > 0$, or $\mathrm{GWT}(\theta)$, if its survival function $S_X$ is bounded in the following way:*

$$\forall x > 0, \quad \exp\left(-x^\theta f_1(x)\right) \leq S_X(x) \leq \exp\left(-x^\theta f_2(x)\right),$$

*where $f_1$ and $f_2$ are slowly-varying functions and $\theta > 0$.*

**Proposition 5** (Vladimirova et al., 2021, Thm. 2.2)**.** *The product of two independent non-negative random variables $|W|$ and $|Y|$ which are respectively $\mathrm{GWT}(\theta_W)$ and $\mathrm{GWT}(\theta_Y)$ is $\mathrm{GWT}(\theta)$, with $\theta$ such that:*

$$\frac{1}{\theta} = \frac{1}{\theta_W} + \frac{1}{\theta_Y}.$$

---

[16]Though, even if each $W_j\phi(X_j)$ has an infinite density at $0$, the density at $0$ of the weighted sum $Z = \frac{1}{\sqrt{n}} \sum_{j=1}^n W_j\phi(X_j) + B$ may be finite. For instance, it occurs when all the $W_j$ and $\phi(X_j)$ are i.i.d. and Gaussian. But in this case, even if $f_Z(0) < \infty$, it is impossible to recover a Gaussian pre-activation (see Prop. 3).

We recall that, in our case, $|G| = |W| \cdot |Y|$ is the absolute value of a Gaussian random variable. So $|G|$ is GWT(2). Thus, if we assume that $|W|$ and $|Y|$ are respectively GWT($\theta_W$) and GWT($\theta_Y$), then we have:

$$\frac{1}{2} = \frac{1}{\theta_W} + \frac{1}{\theta_Y}.$$

Therefore we have the following constraint.

**Constraint 4.** *The weights $W$ are* GWT($\theta$) *with* $\theta \in \Theta = (2, \infty)$.

### 3.2.3 Conclusion

Constraints 2 and 4 indicate that the distribution P of the weights $W$:

(i) should have a density $f_W$ such that $f_W(0) = 0$;

(ii) should be GWT($\theta$) with $\theta \in (2, \infty)$.

A simple choice for P matching these two conditions is: $P = P_\theta = \mathcal{W}(\theta, 1)$ with $\theta \in (2, \infty)$, where $\mathcal{W}(\theta, 1)$ is the symmetric Weibull distribution, defined in Equation (12). Thus, we ensure that $f_W(0) = 0$ and $W$ is generalized Weibull-tail with a parameter $\theta$ easy to control (see remark below).

**Remark 9.** *If $W \sim \mathcal{W}(\theta, 1)$, then $W$ is* GWT($\theta$).

### 3.3 Obtaining the distribution of the activations $Y$

Now that the distribution P of $W$ is supposed to be symmetric Weibull, that is, $P = P_\theta = \mathcal{W}(\theta, 1)$, we are able to look for an *odd*[17] activation function $\phi_\theta$ such that:

$$W \sim \mathcal{W}(\theta, 1), X \sim \mathcal{N}(0, 1) \implies W\phi_\theta(X) \sim \mathcal{N}(0, 1). \tag{17}$$

As a first step, we look for a distribution $Q_\theta$ such that:

$$W \sim P_\theta, Y \sim Q_\theta \implies WY =: G \sim \mathcal{N}(0, 1).$$

In order to "invert" this equation, it is natural to make use of the Mellin transform. A comprehensive and historical work about Fourier and Mellin transforms can be found in Titchmarsh (1937), and a simple application to the computation of the density of the product of two random variables can be found in Epstein (1948).

However, the technique involving the Mellin transform is very difficult to use in this case, both analytically and numerically. Details about the Mellin transform and these difficulties can be found in Appendix E.

---

[17]See Remark 10 for a discussion about the parity of $\phi_\theta$.

**Computation of $f_{|Y|}$: hand-designed parameterized function.** Thus, inspired by the shape of $f_{|Y|}$ computed via the numerical inverse Mellin transform (see Fig. 13, App. E.2), we build an approximation of $f_{|Y|}$ from the family of functions $\{g_{\alpha,\gamma,\lambda_1,\lambda_2} : \alpha, \gamma, \lambda_1, \lambda_2 > 0\}$ with:

$$g_\Lambda(x) := g_{\alpha,\gamma,\lambda_1,\lambda_2}(x) := \gamma\alpha\frac{x^{\alpha-1}}{\lambda_1^\alpha}\exp\left(-\frac{x^\alpha}{\lambda_1^\alpha}\right) + \sqrt{\frac{2}{\pi}}\frac{1}{\Gamma\left(1-\frac{1}{\theta}\right)}\exp\left(-\frac{x^{\theta'}}{\lambda_2^{\theta'}}\right),$$

where $\theta'$ is the conjugate of $\theta$: $\frac{1}{\theta} + \frac{1}{\theta'} = \frac{1}{2}$, and $\Lambda := (\alpha, \gamma, \lambda_1, \lambda_2)$. It is clear that, whatever the parameters, $g_\Lambda(0) = \sqrt{\frac{2}{\pi}}\left[\Gamma\left(1-\frac{1}{\theta}\right)\right]^{-1}$, which is exactly Constraint 3. Moreover, when $\alpha = 0$, $g$ matches also Constraint 4.

Then, we optimize the vector of parameters $\Lambda$ with respect to the following loss:

$$\ell(\Lambda) := \|\hat{F}_\Lambda - F_{|G|}\|_\infty$$
$$\hat{F}_\Lambda(z) := \int_0^\infty F_{|W|}\left(\frac{z}{t}\right) g_\Lambda(t)\, dt \quad \text{(see Eqn. (24)),}$$

where $\hat{F}_\Lambda$ is meant to approximate the CDF of the absolute value of a Gaussian $\mathcal{N}(0,1)$. The integral is computed numerically. For the loss, we have chosen to compute the $\mathcal{L}^\infty$-distance between two CDFs, in order to be consistent with the Kolmogorov–Smirnov test we perform in Section 4.1. Optimization details can be found in Appendix F.

### 3.4 Obtaining the activation function $\phi_\theta$

In the preceding section, we have computed the distribution of $|Y|$. We are restricting ourselves to symmetrical $Y$, whose distribution is denoted by $Q_\theta$. Now, we want to build the activation function $\phi_\theta$, in order to transform a pre-activation $G \sim \mathcal{N}(0,1)$ into an activation $Y = \phi_\theta(G)$ distributed according to $Q_\theta$:

$$\text{if } G \sim \mathcal{N}(0,1), \quad \text{then } \phi_\theta(G) \sim Q_\theta.$$

To compute $\phi_\theta$, we will use the following proposition:

**Proposition 6.** *Let $X$ be a random variable such that $F_X$ is strictly increasing. Let $Q$ be a distribution with a strictly increasing CDF $F_Q$. Then there exists a function $\phi$ such that:*

$$\phi(x) := F_Q^{-1}(F_X(x)) \quad and \quad \phi(X) \sim Q.$$

*Proof.* We want to find $\phi$ such that $F_Q(y) = \mathbb{P}(\phi(X) \le y)$.

Let $\phi(x) := F_Q^{-1}(F_X(x))$, which is a strictly increasing bijection from $\mathbb{R}$ to $\mathbb{R}$. We have:

$$\mathbb{P}(\phi(X) \le y) = \mathbb{P}(X \le \phi^{-1}(y)) = F_X(\phi^{-1}(y)) = F_X(F_X^{-1}(F_Q(y))) = F_Q(y).$$

$\square$

Since $f_G$ and $f_Y$ are strictly positive, $F_G$ and $F_Y$ are strictly increasing and we can use Proposition 6:

$$\phi_\theta(t) := F_Y^{-1}(F_G(t)), \qquad F_Y(t) := \frac{1}{2} + \frac{1}{2}\mathrm{sgn}(t)\int_0^{|t|} f_{|Y|}(y)\, \mathrm{d}y. \qquad (18)$$

**Remark 10.** *The resulting activation functions $\phi_\theta$ are odd, because we have chosen a symmetrical $Y$. Actually, there are several non-symmetrical $Y$ for which $f_{|Y|}$ matches the conditions imposed in the preceding sections, and that lead to derivable activation functions. However, in the family of all possible distributions for $Y$, we identify only two "natural" usable solutions: $Y$ is symmetric or $Y$ is non-negative. We have chosen the first solution.*

### 3.5   Results and limiting cases

We have plotted in Figure 5 the different distributions related to the computation of $\phi_\theta$ and the functions $\phi_\theta$ themselves. The family of the $\phi_\theta$ is a continuum spanning unbounded functions from the tanh-like function $\phi_{2^+}$ and the identity function.

Our construction degenerates into the following two extreme cases at the boundaries of the parameter space $\Theta = (2, \infty)$:

- when $\theta \to \infty$, we have $\mathrm{P}_\theta \xrightarrow{d} \mathcal{R}$ and $\phi_\theta \to \mathrm{Id}$ pointwise;

- when $\theta \to 2$, we have $\mathrm{P}_\theta \xrightarrow{d} \mathcal{W}(2, 1)$ and $\phi_\theta \to \phi_{2^+}$ pointwise;

where $\mathcal{R}$ is the Rademacher distribution ($\xi \sim \mathcal{R} \Leftrightarrow \mathbb{P}(\xi = \pm 1) = \frac{1}{2}$), Id is the identity function, and $\phi_{2^+}$ is a specific increasing function with: $\lim_{\pm\infty} \phi_{2^+} = \pm 1$ and $\phi'_{2^+}(0) = \sqrt{\pi}$.

In the limiting case $\theta \to \infty$, we initialize the weights at $\pm 1$, which corresponds to binary "weight quantization", used in the field of neural networks compression (Pouransari et al., 2020), and we use a linear activation function, commonly used in theoretical analyses of neural networks (Arora et al., 2019). In the limiting case $\theta \to 2$, we recover weights with Gaussian tails, with a tanh-like activation function.

## 4   Experiments

In this section, we test the Gaussian hypothesis with ReLU, tanh and our activation functions $\phi_\theta$, after one layer (Section 4.1) and after several layers (Section 4.2). Then, we plot the Edge of Chaos graphs $(\sigma_b, \sigma_w)$, which are exact with $\phi = \phi_\theta$ (Section 4.3) in the case of finite $n_l$ with independent pre-activations. Finally, in Section 4.4, we show the training trajectories of LeNet-type networks and multilayer perceptrons, when using tanh, ReLU, and various activation functions we have proposed.

In the following subsections, when we use $\phi = \tanh$ or ReLU, we initialize the weights according to a Gaussian at the EOC. This is, for tanh: $\sigma_b^2 = 0.013$, $\sigma_w^2 = 1.46$; and for ReLU: $\sigma_b^2 = 0$, $\sigma_w^2 = 2$. When we use $\phi = \phi_\theta$, we initialize the weights according to $\mathcal{W}(\theta, 1)$.

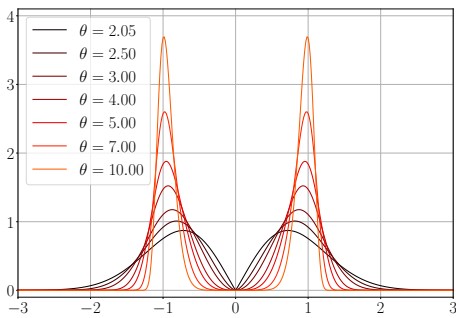

(a) Initialization distribution $P_\theta$ of the weights: symmetric Weibull $\mathcal{W}(\theta, 1)$.

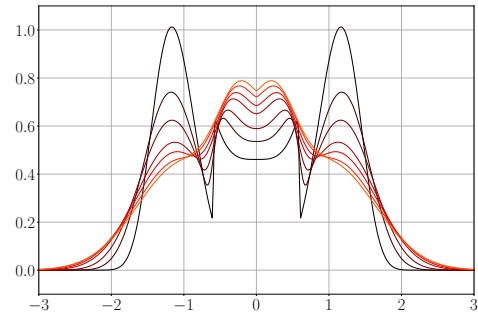

(b) Estimated density $f_Y$ of the distribution $Q_\theta$ of $Y$.

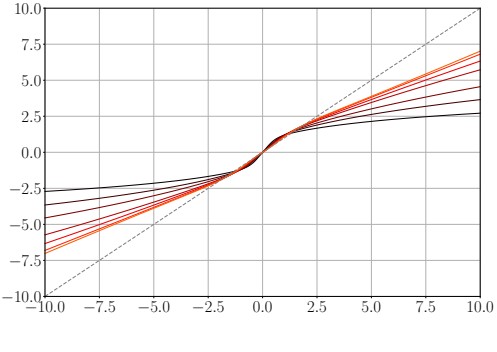

(c) Activation function $\phi_\theta$.

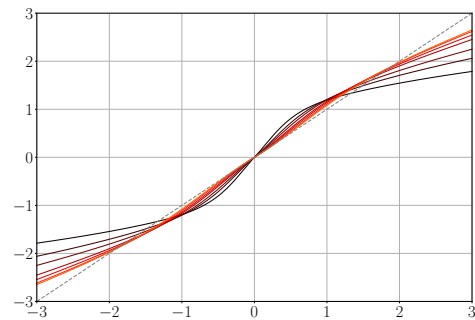

(d) Activation function $\phi_\theta$ around zero.

Figure 5 – How to build a random variable $WY = W\phi_\theta(X) =: G \sim \mathcal{N}(0,1)$, where $X \sim \mathcal{N}(0,1)$? (a) Choose the distribution $P_\theta$ of $W$, then (b) deduce the distribution $Q_\theta$ of $Y$, and finally (c-d) find $\phi_\theta$.

**Notation for the activation functions.** We recall that: $\phi$ denotes an arbitrary activation function; $\varphi_{\delta,\omega}$ denotes the function defined in Definition 1; $\phi_\theta$ denotes the odd function verifying Eqn. (17).

## 4.1 Testing the Gaussian hypothesis: synthetic data, one layer

Above all, we have to check experimentally that we are able to produce Gaussian pre-activations with our family of initialization distributions and activation functions $\{(P_\theta, \phi_\theta) : \theta \in (2, \infty)\}$.

**Framework.** First, we test our setup in the one-layer neural network case with synthetic inputs. More formally, we consider a $\mathcal{N}(0,1)$ *pre-input*[18] $\mathbf{Z} \in \mathbb{R}^n$, which is meant to be first transformed by the activation function $\phi_\theta$ (hence the name "*pre*-input"), then multiplied by

---

[18]Such a pre-input plays the role of the pre-activation outputted by a hypothetical preceding layer.

a matrix of weights $\mathbf{W} \in \mathbb{R}^{1 \times n}$. This one-neuron layer outputs a scalar $Z'$:

$$Z' := \frac{1}{\sqrt{n}} \mathbf{W} \phi(\mathbf{Z}), \quad \text{with } Z_j \sim \mathcal{N}(0,1) \text{ and } W_{1j} \sim \mathrm{P}_\theta \text{ for all } j.$$

We want to check that the distribution $\mathrm{P}'$ of $Z'$ is equal to $\mathcal{N}(0,1)$.

**Experimental results.** For that, we use of the Kolmogorov–Smirnov (KS) test (Kolmogoroff, 1941; Smirnov, 1948) (see Appendix G). We perform the KS test within two setups: with and without preliminary standardization of the sets of samples. With a preliminary standardization, we perform the test on $(\bar{Z}'_1, \cdots, \bar{Z}'_s)$:

$$\bar{Z}'_k = \frac{Z'_k - \bar{\mu}}{\bar{\sigma}}, \quad \bar{\mu} = \frac{1}{s} \sum_{k=1}^{s} Z'_k, \quad \bar{\sigma}^2 = \frac{1}{s-1} \sum_{k=1}^{s} (Z'_k - \bar{\mu})^2.$$

For the sake of simplicity, let us denote by $\hat{F}_{Z'} := F_s$ the empirical CDF of $Z'$, computed with the $s$ data samples $(Z'_1, \cdots, Z'_s)$, and let $\hat{F}_{\bar{Z}'}$ be the empirical CDF of standardized $Z'$, computed with $(\bar{Z}'_1, \cdots, \bar{Z}'_s)$.

We have plotted in Figure 6 the KS statistic of the distribution of the output $Z'$, when using our activation functions $\phi_\theta$, tanh and ReLU. Our sample size is $s = 10^7$. A small KS statistic corresponds to a configuration where $Z'$ is close to being $\mathcal{N}(0,1)$. If a point is above the KS threshold (green line, dotted), then the Gaussian hypothesis is rejected with $p$-value 0.05.

When we perform standardization (Fig. 6b), the neurons using $\phi_\theta$ output always a pre-activation $Z'$ that is closer to $\mathcal{N}(0,1)$ than with ReLU or tanh. But, despite this advantage, the Gaussian hypothesis should be rejected with $\phi_\theta$ when the neuron has a very small number of inputs ($n < 30$).

In Figure 6a, we compare directly the distribution of $Z'$ to $\mathcal{N}(0,1)$. This test is harder than testing the Gaussian hypothesis because the variance of $Z'$ must be equal to 1. We observe that when using $\phi_\theta$, the KS statistic remains above the threshold (while it is still below $10^{-2}$, and even below $6 \cdot 10^{-3}$ for $n \geq 3$). This result is due to the fact that our computation of $\phi_\theta$ is only approximate (see Section 3.4).

**Remark 11.** *Our sample size ($s = 10^7$) is very large, which lowers the threshold of rejection of the Gaussian hypothesis. We have chosen this large $s$ to reduce the noise of the KS statistics. If we had chosen $s = 18000$, a threshold close to $10^{-2}$ would have resulted, which is higher than any of the KS statistics computed with $\phi = \phi_\theta$. One will also note that, in Section 4.2, we use only $s = 10000$ samples to keep a reasonable computational cost.*

**Remark 12.** *Since* tanh *and* ReLU *have not been designed such that $Z' \sim \mathcal{N}(0,1)$, we did not plot the related non-standardized KS statistics in Figure 6a. Anyway, given the standard deviation of $Z'$ when using $\phi = $ tanh or ReLU (see Table 2, Appendix H.3), very large KS statistics are expected in this setup.*

We discuss the limits of the KS test in Appendix G.

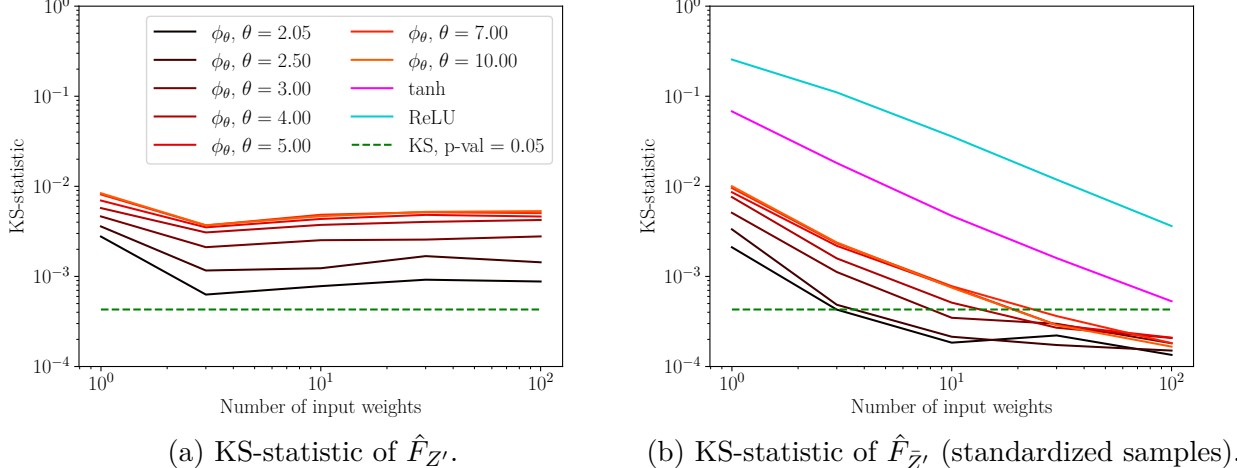

(a) KS-statistic of $\hat{F}_{Z'}$.

(b) KS-statistic of $\hat{F}_{\bar{Z}'}$ (standardized samples).

Figure 6 – Evolution of the KS statistic of the distribution of $Z'$ (Fig. 6a) and the standardized distribution of $Z'$ (Fig. 6b), with a number of inputs $n \in \{1, 3, 10, 30, 100\}$.

## 4.2 Testing the Gaussian hypothesis: CIFAR-10, multilayer perceptron

Now, we test our setup on a multilayer perceptron with CIFAR-10, which is more realistic. We show in Figure 7 how the distribution of the pre-activations propagates in a multilayer perceptron, for different layer widths $n_l \in \{10, 100, 1000\}$.

**Setup.** Let $\mathcal{D}^l$ be the distribution of the pre-activation $Z_1^l$ after layer $l$. For all $l$, let us define $(Z_{1;k}^l)_{k \in [1,s]}$, a sequence of i.i.d. samples drawn from $\mathcal{D}^l$. The plots in Figure 7 show the evolution of the $\mathcal{L}^\infty$ distance $\|\hat{F}_{Z_1^l} - F_G\|_\infty$ between the CDF of $\mathcal{N}(0,1)$, $F_G$, and the empirical CDF of $\mathcal{D}^l$, $\hat{F}_{Z_1^l}$, built with $s = 10000$ samples $(Z_{1;k}^l)_{k \in [1,s]}$.

We have built the plots of Figure 7 with the same input data point.[19] See Appendix H.4 for a comparison of the propagation between different data points.

In Figures 7a and 7c, the propagated data point has been normalized according to the whole training dataset, that is:

**Definition 5** (Input normalization over the whole dataset)**.** *We build the normalized data point $\hat{\mathbf{x}}$:*

$$\hat{x}_{a;ij} := \frac{x_{a;ij} - \mu_i}{\sigma_i}, \quad \mu_i := \frac{1}{p_i d} \sum_{\mathbf{x} \in \mathbb{D}} \sum_{j=1}^{p_i} x_{ij}, \quad \sigma_i^2 := \frac{1}{p_i d - 1} \sum_{\mathbf{x} \in \mathbb{D}} \sum_{j=1}^{p_i} (x_{ij} - \mu_i)^2,$$

*where $x_{a;ij}$ is the $j$-th component of the $i$-th channel of the input image $\mathbf{x}_a$, $p_i$ is the size of the $i$-th channel, and $d$ is the size of the dataset $\mathbb{D}$.*

---

[19]In the PyTorch implementation of the training set of CIFAR-10: data point #47981 (class = plane). This data point has been chosen randomly.

In Figure 7b, the propagated data point has been normalized individually, that is:

**Definition 6** (Individual input normalization). *We build the normalized data point $\hat{\mathbf{x}}$:*

$$\hat{x}_{a;i} := \frac{x_{a;i} - \mu}{\sigma}, \quad \mu := \frac{1}{p}\sum_{i=1}^{p} x_i, \quad \sigma^2 := \frac{1}{p-1}\sum_{i=1}^{p}(x_i - \mu)^2,$$

*where $\hat{\mathbf{x}}$ is the normalized data point, $x_{a;i}$ is the $i$-th component of $\mathbf{x}_a \in \mathbb{R}^p$.*

**Results.** We distinguish two measures of the distance between the distribution of $Z_1^l$ and a Gaussian: in Figures 7a and 7b, we measure the distance between the distribution of $Z_1^l$ and $\mathcal{N}(0,1)$; in Figure 7c, we measure the distance between the *standardized* distribution of $\bar{Z}_1^l$ and $\mathcal{N}(0,1)$. In short, we test in Figure 7c the Gaussian hypothesis, whatever the mean and the variance of the pre-activations.

First, in all cases, ReLU leads to pre-activations that diverge from the Gaussian family. Also, with 10 neurons per layer, tanh does not lead to Gaussian pre-activations.

Second, our proposition of activation functions $\phi = \phi_\theta$ leads to various results, depending on the layer width $n_l$. Above all, with individual input normalization (Fig. 7b), the first pre-activations are very close to $\mathcal{N}(0,1)$, which is what we intended. For $\theta = 2.05$, the curve remains below the KS threshold with $p$-value 0.05, whatever the layer width $n_l$. However, as $\theta$ grows, the related curves tend to drift away from 0, especially when $n_l$ is small.

However, when standardizing the pre-activations (Figure 7c), three of our activation functions ($\theta \in \{2.05, 2.5, 3\}$) remain below the KS threshold in the least favorable case ($n_l = 10$), which is not the case for ReLU and tanh.

**Conclusion.** For all tested widths, combining an activation function $\phi_\theta$ with $\theta$ close to 2 and weights sampled from $\mathcal{W}(\theta, 1)$ leads to pre-activations that are closer to $\mathcal{N}(0,1)$ than with $\phi = \tanh$ or ReLU. However, our proposition is not perfect for all $\theta > 2$: we always observe that the sequence $(\mathcal{D}^l)_l$ drifts away from $\mathcal{N}(0,1)$. But, overall, this drift is moderate with layer widths $n_l \geq 100$, and even disappears when we standardize the pre-activations.

So, in order to keep Gaussian pre-activations along the entire neural network, one should take into account this "drift". We can interpret it as a divergence due to the fact that $\mathcal{D}^l = \mathcal{N}(0,1)$ is not a stable fixed point of the recurrence relation $\mathcal{D}^{l+1} = \mathcal{T}_l[\mathrm{P}_l](\mathcal{D}^l)$ (see Fig. 4 in Section 2.4). So, it is natural that $(\mathcal{D}^l)_l$ drifts away from $\mathcal{N}(0,1)$, and converges to a *stable* fixed point, which seems to be $\mathcal{N}(\mu, \sigma^2)$ with parameters $\mu$ and $\sigma^2$ to determine, at least for $n_l = 1000$.

This search for stable fixed points in the recurrence relation $\mathcal{D}^{l+1} = \mathcal{T}_l[\mathrm{P}_l](\mathcal{D}^l)$ is closely related to the discovery of stable fixed points for the sequence of variances $(v_a^l)_l$ and the sequence of correlations $(c_{ab}^l)_l$ (Poole et al., 2016; Schoenholz et al., 2017), and may be explored further in future works.[20]

---

[20]The fixed points of $\mathcal{D}^{l+1} = \mathcal{T}_l[\mathrm{P}_l](\mathcal{D}^l)$ can also be seen as stationary distributions of the Markov chain $(Z_1^l)_l$, if the layer width $n_l$ and the initialization distribution $\mathrm{P}_l$ are constant.

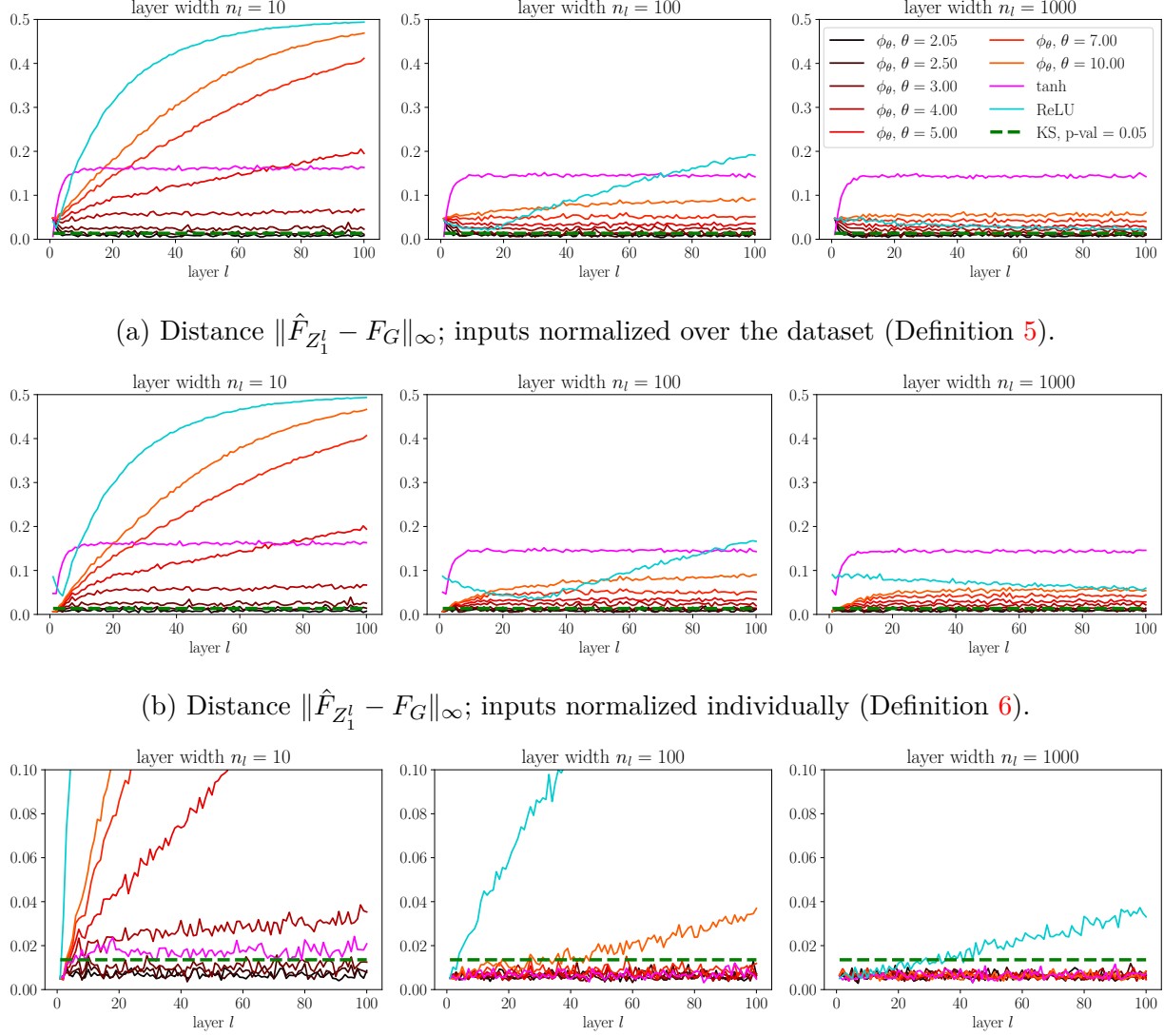

(a) Distance $\|\hat{F}_{Z_1^l} - F_G\|_\infty$; inputs normalized over the dataset (Definition 5).

(b) Distance $\|\hat{F}_{Z_1^l} - F_G\|_\infty$; inputs normalized individually (Definition 6).

(c) Distance $\|\hat{F}_{\bar{Z}_1^l} - F_G\|_\infty$, where $\bar{Z}_1^l = \frac{Z_1^l - \bar{\mu}^l}{\bar{\sigma}^l}$ is the standardized version of $Z_1^l$; inputs normalized over the dataset (Definition 5). This figure has been zoomed around 0.

Figure 7 – Evolution of the distance of $\mathcal{D}^l$ to a Gaussian according to different metrics, during propagation where $l$ varies from 1 to 100. $\bar{Z}_1^l$ is the standardized version of $Z_1^l$; $\bar{\mu}^l$ and $\bar{\sigma}^l$ are respectively the empirical mean and the empirical (corrected) standard deviation of $Z_1^l$. The dotted green line is the KS threshold: any point above it corresponds to a distribution $\mathcal{D}^l$ for which the Gaussian hypothesis should be rejected with $p$-value 0.05. Weight initialization is $\mathcal{W}(\theta, 1)$ when using $\phi = \phi_\theta$ and is Gaussian according to the EOC when using $\phi = \tanh$ or ReLU.

## 4.3  Non-asymptotic Edge of Chaos

In Figure 8, we show the Edge of Chaos graphs for several activation functions: tanh and ReLU on one side, and our family $(\phi_\theta)_\theta$ on the other side. We remind that each graph corresponds to a family of initialization standard deviations $(\sigma_w, \sigma_b)$ such that the sequence of correlations $(c^l_{ab})_l$ converges to 1 at a *sub-exponential* rate (see Section 2.1, Point 2). Such choices ensure that the initial correlation between two inputs changes slowly so that the information contained in these inputs is lost at the slowest possible rate.

Instead of assuming that the pre-activations are Gaussian as an effect of the "infinite-width limit" and the Central Limit Theorem, we claim that, with our activation functions $\phi_\theta$, for any layer widths (including narrow layers and networks with various layer widths), the Edge of Chaos is *non-asymptotic*. Therefore, the corresponding curves in Figure 8 hold for realistic networks.

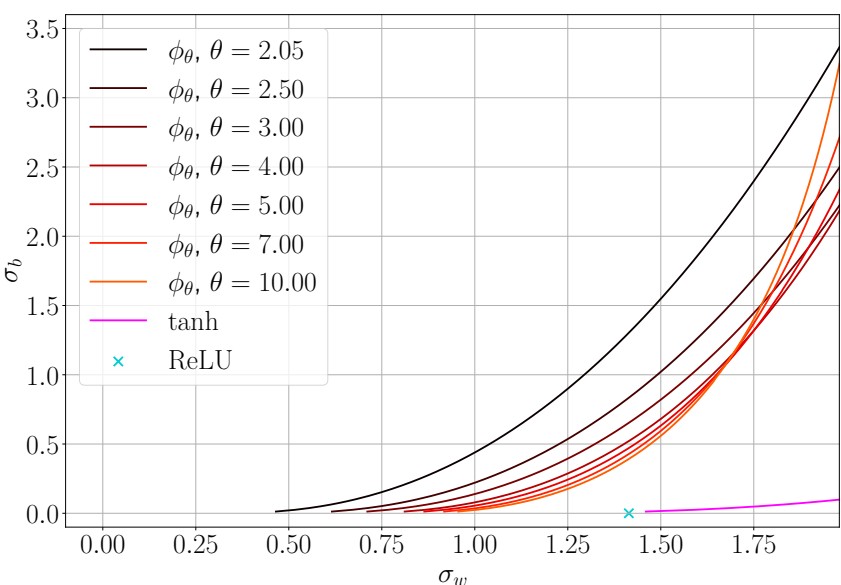

Figure 8 – Edge of Chaos for several activation functions. The ordered phase (resp. chaotic) lies above (resp. below) each EOC curve.

**Remark 13.** *For the* ReLU *activation function, the EOC graph reduces to one point. One can refer to Hayou et al. (2019, Section 3.1) for a complete study of the EOC of "ReLU-like functions", that is, functions that are linear on $\mathbb{R}^+$ and on $\mathbb{R}^-$ with possibly different factors.*

Also, as in Figure 1 (see Section 2.2), we have plotted the propagation of the correlations $(C^l_{pq})_l$ with $\phi = \phi_\theta$ and weights sampled from $\mathcal{W}(\theta, 1)$ in Appendix H.2.

## 4.4 Training experiments

Finally, we compare the performance of a trained neural network when using tanh and ReLU, and our activation functions. Despite the EOC framework (and ours) do not provide any quantitative prediction about the training trajectories, it is necessary to analyze them in order to enrich the theory.

This section starts with a basic check of the training and test performances on a common task: training LeNet on CIFAR-10. Then, we challenge our activation function, along with tanh and ReLU, by training on MNIST a diverse set of multilayer perceptrons, some of them being extreme (narrow and deep).

In the following, we train all the neural networks with the same optimizer, Adam, and the same learning rate $\eta = 0.001$. We use a scheduler and an early stopping mechanism, respectively based on the training loss and the validation loss, the test loss not being used during training. We did not use data augmentation. All the technical details are provided in Appendix H.5.

**LeNet-type networks.** We consider LeNet-type networks (LeCun et al., 1998). They are made of two $(5 \times 5)$-convolutional layers, each of them followed by a 2-stride average pooling, and then three fully-connected layers. We denote by "$6 - 16 - 120 - 84 - 10$" a LeNet neural network with two convolutional layers outputting respectively 6 and 16 channels, and three fully connected layers having respectively 120, 84, and 10 outputs (the final output of size 10 is the output of the network).

We have tested LeNet with several sizes (see Figure 9). In Figure 9a, we have plotted the training loss. Overall, ReLU and tanh perform well, along with some $\phi_\theta$ with small $\theta$ and $\varphi_{\delta,\omega}$, with $(\delta, \omega) = (0.99, 2)$. In Figure 9b, the results in terms of test accuracy are quite different: ReLU and tanh still achieve good accuracy, but the other functions achieving similar results on the training loss seem to be a bit behind.

So, in this standard setup, the functions we are proposing seem to make the neural network trainable and as expressive as with other activation functions, but with some overfitting. This is not surprising, since we are testing long-standing activation functions, ReLU and tanh, which have been selected both for their ability to make the neural network converge quickly with good generalization, against functions we have designed only according to their ability to propagate information. Therefore, according to these plots, taking into account generalization may be the missing piece of our study.

**Multilayer perceptron.** We have trained a family of multilayer perceptrons on MNIST. They have a constant width $n_l \in \{3, 10\}$ and a depth $L \in \{3, 10, 30\}$. So, extreme cases such as a narrow and deep neural network ($n_l = 3$, $L = 30$) have been tested.

Two series of results are presented in Figure 10. The setups of Figure 10a and Figure 10b are identical, except for the initial random seed. In terms of training, the strength of our activation functions $\phi_\theta$ is more visible in the case of narrow neural networks ($n_l = 3$): in

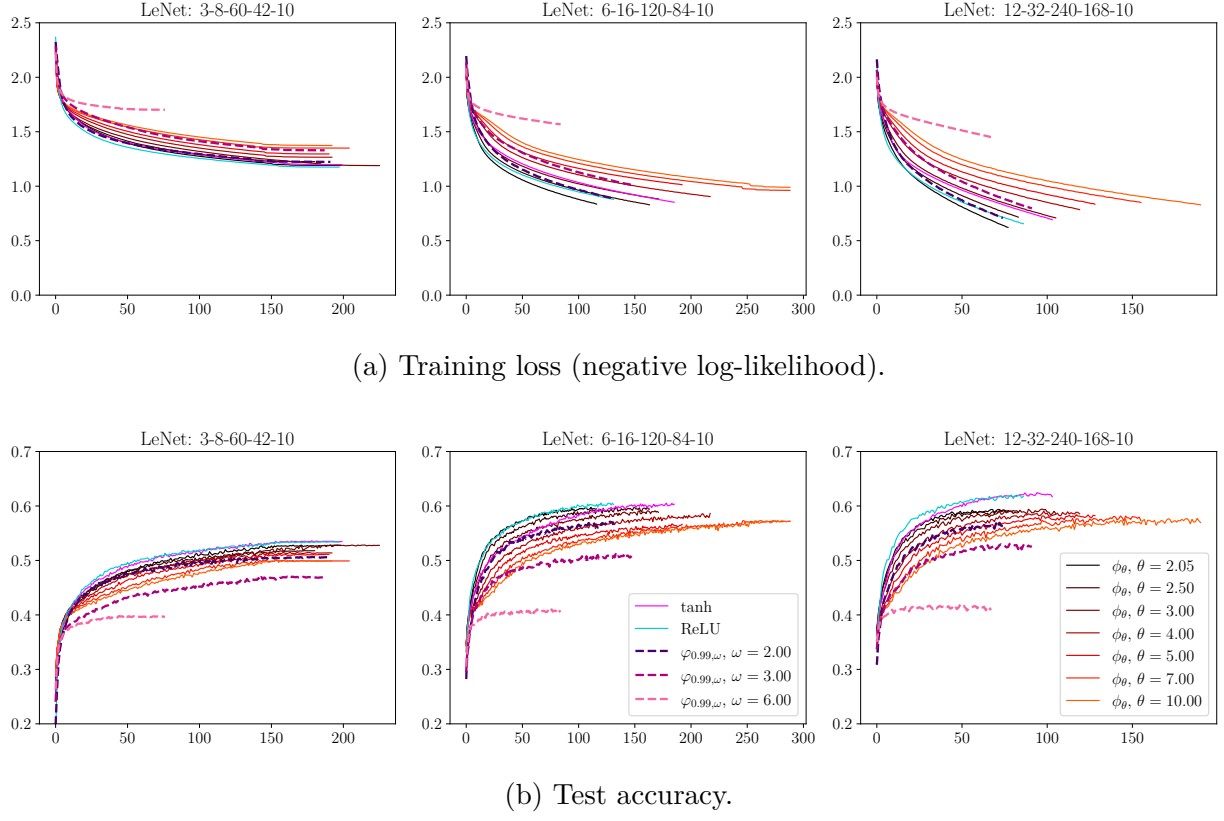

(a) Training loss (negative log-likelihood).

(b) Test accuracy.

Figure 9 – Training curves for LeNet with 3 different numbers of neurons per layer.

general, the loss decreases faster and attains better optima with $\phi = \phi_\theta$ than with $\phi = \tanh$ or ReLU.

We also notice that training a narrow and deep neural network with a ReLU activation function is challenging. This result is consistent with several observations we have made in Section 2.2: in narrow ReLU networks, the sequence of correlations $(C_{pq}^l)_l$ fails to converge to 1 (Fig. 1), and the pre-activations are far from being Gaussian (Fig. 2).

Finally, $\phi_{2.05}$ put aside, the results regarding the activation functions $\phi_\theta$ are consistent between the two runs. This is not the case with tanh and ReLU.

## 5   Discussion

**Generality of Constraints 1 to 4.**   Provided that we want pre-activations that are Gaussian $\mathcal{N}(0, 1)$, and we want the weights of each layer to be i.i.d. at initialization, the set of four constraints that we provide in Section 2 must hold. If one wants to relax these constraints, it becomes unavoidable to break symmetries or to solve harder problems. Let us consider two examples.

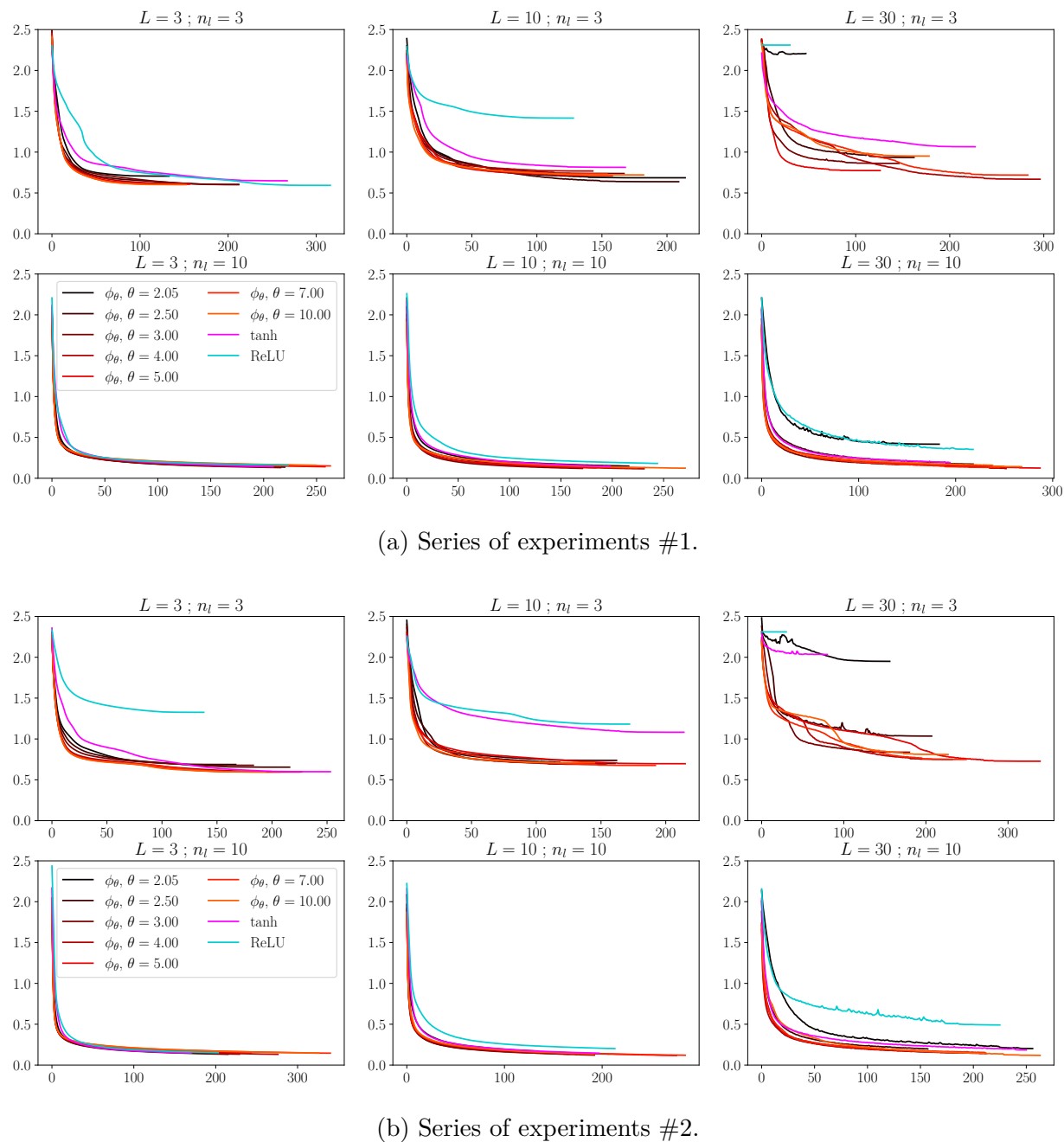

(a) Series of experiments #1.

(b) Series of experiments #2.

Figure 10 – Training loss for a multilayer perceptron, narrow ($n_l \in \{3, 10\}$) and of various depths ($L \in \{3, 10, 30\}$).

Example 1: instead of making all the pre-activations Gaussian, one may want to impose some other distribution. In this case, the problem to solve would be more difficult: we would have to decompose a non-Gaussian random variable $Z$ into a weighted sum $\frac{1}{\sqrt{n}} \sum_{j=1}^{n} W_j \phi(X_j)$. In this case, we cannot use Proposition 3, and we would have to make $W_j \phi(X_j)$ belong to a family of random variables stable by multiplication by a constant and by sum, for arbitrary $n$. If we aim for a non-Gaussian $Z$, this task is much harder. For a study of infinitely wide neural networks going beyond Gaussian pre-activations, see Peluchetti et al. (2020).

Example 2: instead of assuming that all the weights of a given layer are i.i.d., one may want to initialize them with different distributions or to introduce a dependence structure between them. If the goal remains to obtain Gaussian pre-activations, this kind of generalization should be feasible without changing drastically the constraints we are proposing.

**Hypothesis of independent pre-activations.**   In the constraints, we have assumed that, for any layer, its inputs are independent. This assumption is discussed in Remark 7 and in Appendix B. According to the experimental results presented in Appendix B, we can build specific cases where the dependence between inputs breaks the Gaussianity of the outputted pre-activations, including with our pairs $(\phi_\theta, P_\theta)$.

Also, when comparing the distribution of the pre-activations obtained with $(\phi_\theta, P_\theta)$ in the case of independent inputs (Fig. 6b) and in the case of dependent inputs (Fig. 7c and 11), it is probable that the dependence between pre-activations tends to damage the Gaussianity after a certain number of layers, for pairs $(\phi_\theta, P_\theta)$ with large $\theta$.

**Other families of initialization distributions and activation functions.**   Provided the constraints we have derived, we have made one choice to obtain our family of initialization distributions and activation functions: we have decided that the weights should be sampled from a symmetric Weibull distribution $\mathcal{W}(\theta, 1)$. We have made this choice because Weibull distributions meet immediately Constraint 2, and we can modulate their Generalized Weibull Tail parameter easily (see Remark 9).

One may propose another family of initialization distributions, as long as it meets the constraints. However, such a family should be selected wisely: if one chooses a distribution with compact support, or $\mathrm{GWT}(\theta)$ with $\theta \gg 2$, then the related activation function $\phi$ is likely to be almost linear. Intuitively, if we want $W\phi(X)$ to be $\mathcal{N}(0,1)$ with $X \sim \mathcal{N}(0,1)$ and very light-tailed $W$, then $\phi$ must "reproduce" the tail of its input $X$, so $\phi$ must be approximately linear around infinity.

**Preserve a characteristic during propagation or impose stable fixed points?**   In previous works, both ideas have been used: Glorot & Bengio (2010) wanted to preserve the variance of the pre-activations, while Poole et al. (2016); Schoenholz et al. (2017) wanted to impose a specific stable fixed point for the correlation map $\mathcal{C}$. According to the results we have obtained in Section 4.1, it is possible to preserve *approximately* the distribution of the pre-activations when passing through *one* layer. However, according to the results of

Section 4.2, a drift can appear after several layers. So, when testing an initialization setup in the real world, with numerical errors and approximations, it is necessary to check the stable fixed points of the monitored characteristic. However, this is not easy to do in practice: without the Gaussian hypothesis, finding the possible limits of $(c_{ab}^l)_l$ can be difficult.

**Gaussian pre-activations and Neural Tangent Kernels (Jacot et al., 2018).** With our pair of activation functions and initialization procedure, we have been able to provide non-asymptotic Edge of Chaos, removing the infinite-width assumption. Since this infinite-width assumption is also fundamental in works on the Neural Tangent Kernels (NTKs), would it be possible to obtain the same kind of results within the NTK setup? We believe it is not. On one side, the infinite-width limit is used to end up with an NTK, which is *constant* during training, in order to provide exact equations of evolution of the trained neural network. On the other side, within our setup, we only ensure Gaussian pre-activations, and it does not imply that the NTK would be constant during training. Nevertheless, with Gaussian pre-activations, we might expect an improvement in the convergence rate of the neural network towards a stacked Gaussian process, as the widths of the layers tend to infinity. So, the NTK regime would be easier to attain.

**Taking into account the generalization performance.** In the EOC framework and ours, a common principle could be discussed and improved. The generalization performance is not taken into account at any step of the reasoning. As we have seen in the training experiments, the main difficulty encountered within our setup is the overfitting: as the training loss decreases without obstacle, the test loss is, at the end, worse than with ReLU and tanh activation functions. To improve these generalization results, one may integrate into the framework a separation between the training set and a validation set.

**Precise characterization of the pre-activations distributions $\mathcal{D}^l$.** Finally, finding a precise and usable characterization of the $\mathcal{D}^l$ remains an unsolved problem. In the EOC line of work, the problem has been simplified by using the Gaussian hypothesis. But, as shown in the present work, such a simplification is too rough and leads to inconsistent results. Nevertheless, our approach has its own drawbacks. Namely, we can ensure Gaussian pre-activations only when using specific initialization distributions and activation functions. So, we still miss a characterization of the distributions $\mathcal{D}^l$ which would apply to widely-used networks without harsh approximations, and be easy to use to achieve practical goals, such as finding an optimal initialization scheme. From this perspective, we hope that the problem representation of Section 2.4 will result in fruitful future research.

**Is it desirable to have Gaussian pre-activations?** We have provided several results that should help to answer this tough question, but it remains difficult to answer it definitively. One shall note a paradox regarding the ReLU activation. On one side, when ReLU is used in narrow and deep perceptrons, the pre-activations are far from being Gaussian, the sequence of correlations does not converge to 1, and training is difficult and unstable. On the other side, in LeNet-type networks with ReLU, training is easy, and the resulting networks

generalize well. Besides, our activation functions $\phi_\theta$ perform quite differently depending on the setup: with $\phi_{2.05}$, LeNet can achieve good training losses, but the training of narrow and deep perceptrons may fail. We observe opposite results with $\phi_{10.00}$. Therefore, the strongest answer we can give is: with Gaussian pre-activations at initialization, a neural network is likely (but not sure) to be trainable, but it is impossible to predict its ability to generalize.

**Is the "Gaussian pre-activations hypothesis" a myth or a reality?**   We have shown that, when using the ReLU activation function in several practical cases, it is largely a myth. But, when using tanh, the results depend on the neural network width: with a sufficient, but still reasonable, number of neurons per layer, it becomes a reality. In order to ensure that this reality remains tangible for any number of neurons per layer, we have established a set of constraints that the design of the neural networks must fulfill, and we have proposed a set of solutions fulfilling them. As a result, several of these solutions have all the prerequisite to become strong foundations of this Gaussian hypothesis, making it real in all tested cases.

## Acknowledgements

We would like to thank an anonymous reviewer for pointing out an error in a previous version of the paper, and proposing an example which inspired Example 1 (see App. B).

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

# A    Activation function with infinite number of stable fixed points for $\mathcal{V}$

## A.1    Proof that $\mathcal{V}$ admits an infinite number of fixed points when using $\phi = \varphi_{\delta,\omega}$

**Proposition 2.** *For any $\delta \in (0,1]$ and $\omega > 0$, let us pose the activation function $\phi = \varphi_{\delta,\omega}$. We consider the sequence $(v^l)_l$ defined by:*

$$\forall l \geq 0, \quad v^{l+1} = \sigma_w^2 \int \varphi_{\delta,\omega}\left(\sqrt{v^l}z\right)^2 \mathcal{D}z + \sigma_b^2, \tag{19}$$
$$v^0 \in \mathbb{R}_*^+.$$

*Then there exists $\sigma_w > 0$, $\sigma_b \geq 0$, and a strictly increasing sequence of stable fixed points $(v_k^*)_{k \in \mathbb{Z}}$ of the recurrence Equation (19).*

*Proof.* Let us define:

$$\tilde{\mathcal{V}}(v) := \frac{1}{v}\mathcal{V}(v|\sigma_w = 1, \sigma_b = 0),$$
$$\text{so we have:} \quad \mathcal{V}(v|\sigma_w, \sigma_b) = \sigma_w^2 v \tilde{\mathcal{V}}(v) + \sigma_b^2. \tag{20}$$

In the following, we use a simplified notation: $\mathcal{V}(v) = \mathcal{V}(v|\sigma_w, \sigma_b)$.

Our goal is to find a sequence $(v_k^*)_{k \in \mathbb{Z}}$, $\sigma_w$ and $\sigma_b$ such that:

$$\forall k \in \mathbb{Z}, \quad \mathcal{V}(v_k^*) = v_k^* \quad \text{and} \quad \mathcal{V}'(v_k^*) \in (-1, 1),$$

which would ensure that all $v_k^*$ are stable fixed points of $\mathcal{V}$. In order to understand how to build the sequence $(v_k^*)_{k \in \mathbb{Z}}$, let us consider $v > 0$. Let $\sigma_b^2 = 0$ and $\sigma_w^2 = 1/\tilde{\mathcal{V}}(v)$. So we have:

$$\mathcal{V}(v) = v.$$

So, any $v > 0$ can possibly be a fixed point if we tune $\sigma_w$ accordingly. We just have to find a $v > 0$ such that:

$$\mathcal{V}'(v) \in (-1, 1).$$

with:

$$
\begin{aligned}
\mathcal{V}'(v) &= \frac{1}{\tilde{\mathcal{V}}(v)} \left( \tilde{\mathcal{V}}(v) + v\tilde{\mathcal{V}}'(v) \right) \\
&= 1 + \tilde{\mathcal{V}}'_{\mathrm{e}}(\ln(v)),
\end{aligned}
\tag{21}
$$

where $\tilde{\mathcal{V}}_{\mathrm{e}} : r \mapsto \ln(\tilde{\mathcal{V}}(\exp(r)))$. So, knowing that $\tilde{\mathcal{V}}_{\mathrm{e}}$ is $\mathcal{C}^1$ and periodic, it is sufficient to prove that it is not constant to ensure that we can extract one $v_0^*$ such that $\mathcal{V}'(v_0^*) \in (-1, 1)$. Then, by periodicity of $\tilde{\mathcal{V}}_{\mathrm{e}}$, we can build a sequence of stable fixed points $(v_k^*)_{k \in \mathbb{Z}}$.

We have:

$$
\begin{aligned}
\tilde{\mathcal{V}}(v) &= \frac{1}{v} \int_{-\infty}^{\infty} \varphi_{\delta,\omega}(\sqrt{v}z)^2 \, \mathcal{D}z \\
&= \int_{-\infty}^{\infty} z^2 \exp\left( 2\frac{\delta}{\omega} \sin(\omega \ln|\sqrt{v}z|) \right) \mathcal{D}z \\
&= 2 \int_0^{\infty} z^2 \exp\left( 2\frac{\delta}{\omega} \sin(\omega \ln(\sqrt{v}z)) \right) \mathcal{D}z \\
&= 2 \int_0^{\infty} z^2 \exp\left( 2\frac{\delta}{\omega} \sin\left( \frac{\omega}{2} \ln v + \omega \ln z \right) \right) \mathcal{D}z.
\end{aligned}
$$

**Lemma 1.** $\tilde{\mathcal{V}}$ *is not constant.*

*Proof.*

$$
\begin{aligned}
\tilde{\mathcal{V}}(v) &= \frac{1}{\sqrt{2\pi}} \frac{2}{v^{3/2}} \int_0^{\infty} z^2 \exp\left( 2\frac{\delta}{\omega} \sin(\omega \ln z) \right) \exp\left( -\frac{z^2}{2v} \right) \mathrm{d}z \\
&= \frac{1}{\sqrt{2\pi}} \frac{2}{v^{3/2}} \int_0^{\infty} \frac{z}{2\sqrt{z}} \exp\left( 2\frac{\delta}{\omega} \sin(\omega \ln \sqrt{z}) \right) \exp\left( -\frac{z}{2v} \right) \mathrm{d}z \\
&= \frac{1}{\sqrt{2\pi}} v^{-3/2} \mathcal{L}\left[ \varphi_{\delta,\omega}(\sqrt{z}) \right] \left( \frac{1}{2v} \right),
\end{aligned}
$$

where $\mathcal{L}$ is the Laplace transform.

Let us suppose that $\tilde{\mathcal{V}}$ is constant: $\forall v > 0, \tilde{\mathcal{V}}(v) = c$. So, if we pose $v \leftarrow \frac{1}{2v}$, we have:

$$\forall v > 0, \quad c = \frac{2}{\sqrt{\pi}} v^{3/2} \mathcal{L}\left[ \varphi_{\delta,\omega}(\sqrt{z}) \right](v),$$

that is:

$$\mathcal{L}\left[ \varphi_{\delta,\omega}(\sqrt{z}) \right](v) = \frac{c\sqrt{\pi}}{2} v^{-3/2}.$$

Since the function $z \mapsto \varphi_{\delta,\omega}(\sqrt{z})$ is continuous on $\mathbb{R}^+$, then, almost everywhere (see Thm. 22.2, Billingsley, 1995):

$$\varphi_{\delta,\omega}(\sqrt{z}) = \frac{c\sqrt{\pi}}{2} \mathcal{L}^{-1}[v^{-3/2}](z) = c\sqrt{z},$$

which is impossible for $\delta \in (0, 1]$. Hence the result. $\qquad\square$

The function $\tilde{\mathcal{V}}_e : r \mapsto \ln(\tilde{\mathcal{V}}(\exp(r)))$ is continuous and $\frac{4\pi}{\omega}$-periodic:

$$\tilde{\mathcal{V}}_e(r) = \ln\left[2 \int_0^\infty z^2 \exp\left(2\frac{\delta}{\omega} \sin\left(\frac{\omega}{2}r + \omega \ln z\right)\right) \mathcal{D}z\right],$$

so $\tilde{\mathcal{V}}_e$ is lower and upper bounded and reach its bounds (and, by Lemma 1, these bounds are different). We define:

$$\tilde{\mathcal{V}}_e^+ = \max \tilde{\mathcal{V}}_e \qquad\qquad r_0^+ = \inf\left\{r > 0 : \tilde{\mathcal{V}}_e(r) = \tilde{\mathcal{V}}_e^+\right\},$$
$$\tilde{\mathcal{V}}_e^- = \min \tilde{\mathcal{V}}_e \qquad\qquad r_0^- = \inf\left\{r > r_0^+ : \tilde{\mathcal{V}}_e(r) = \tilde{\mathcal{V}}_e^-\right\}.$$

By continuity, $\tilde{\mathcal{V}}_e(r_0^+) = \tilde{\mathcal{V}}_e^+$ and $\tilde{\mathcal{V}}_e(r_0^-) = \tilde{\mathcal{V}}_e^-$. Since $\tilde{\mathcal{V}}^+ > \tilde{\mathcal{V}}^-$ and $\tilde{\mathcal{V}}_e$ is $\mathcal{C}^1$, then there exists $r_0^* \in (r_0^+, r_0^-)$ such that $\tilde{\mathcal{V}}_e'(r_0^*) \in (-2, 0)$.

Since $\tilde{\mathcal{V}}_e$ is $\frac{4\pi}{\omega}$-periodic, we can define a sequence $(r_k^*)_{k\in\mathbb{Z}}$ such that:

$$r_k^* := r_0^* + \frac{4k\pi}{\omega}$$
$$\tilde{\mathcal{V}}_e(r_k^*) = \tilde{\mathcal{V}}_e(r_0^*) =: \tilde{\mathcal{V}}_e^0$$
$$\tilde{\mathcal{V}}_e'(r_k^*) = \tilde{\mathcal{V}}_e'(r_0^*) \in (-2, 0).$$

So, by using Eqn. (20) and Eqn. (21) with $\sigma_b^2 = 0$ and $\sigma_w^2 = 1/\exp(\tilde{\mathcal{V}}_e^0)$:

$$v_k^* := \exp(r_k),$$
$$\mathcal{V}(v_k^*) = \frac{1}{\exp(\tilde{\mathcal{V}}_e^0)} v_k^* \tilde{\mathcal{V}}(v_k^*) = v_k^*$$
$$\mathcal{V}'(v_k^*) = 1 + \tilde{\mathcal{V}}_e'(r_0^*) \in (-1, 1).$$

Thus, $(v_k^*)_{k\in\mathbb{Z}}$ is a sequence of stable fixed points of $\mathcal{V}(\cdot|\sigma_w, \sigma_b)$ for well-chosen $\sigma_w$ and $\sigma_b$. $\quad\square$

## A.2 Practical computation of $\sigma_w^2$

We propose a practical method to ensure that $\mathcal{V}(\cdot|\sigma_w, \sigma_b = 0)$ admits an infinite number of stable fixed points when using activation function $\phi = \varphi_{\delta,\omega}$.

In order to achieve this goal, we build $\sigma_w^2 = \sigma_\omega^2$ in the following way:

$$\mathcal{V}_{\text{low}} := 2 \int_0^\infty z^2 \exp\left(-2\frac{\delta}{\omega}\sin(\omega\ln(z))\right)\mathcal{D}z$$

$$\mathcal{V}_{\text{upp}} := 2 \int_0^\infty z^2 \exp\left(2\frac{\delta}{\omega}\sin(\omega\ln(z))\right)\mathcal{D}z,$$

$$\sigma_\omega^2 := \left[\frac{\mathcal{V}_{\text{low}} + \mathcal{V}_{\text{upp}}}{2}\right]^{-1}.$$

In practice, for $\delta = 0.99$, we obtain $\sigma_\omega$ for $\omega \in \{2,3,6\}$:

$$\sigma_2 \approx 0.879, \quad \sigma_3 \approx 0.945, \quad \sigma_6 \approx 0.987.$$

# B  Discussion about the independence of the pre-activations

In Proposition 3 and Constraint 1, we assume that, for any layer $l$, its inputs $(Z_j^l)_j$ are independent. In general, this is not true:

**Example 1.** *Let $X \sim \mathcal{N}(0,1)$ be some random input of a two-layer neural network. We perform the following operation:*

$$Z = \frac{1}{\sqrt{2}}\left[W_1^2\phi(W_1^1 X) + W_2^2\phi(W_2^1 X)\right],$$

*where $(W_1^1, W_2^1, W_1^2, W_2^2)$ be i.i.d. random variables samples from some distribution $\mathrm{P}$, and $\phi$ is some activation function.*

*Let $\mathrm{P} = \mathcal{R}$, the Rademacher distribution, i.e., if $W \sim \mathcal{R}$, then $W = \pm 1$ with probability $1/2$. Let $\phi = \mathrm{Id}$. Then we have: $Y_1 := W_1^1 X \sim \mathcal{N}(0,1)$ and $Y_2 := W_2^1 X \sim \mathcal{N}(0,1)$. But they are not independent:*

$$W_1^2 W_1^1 X + W_2^2 W_2^1 X = (W_1' + W_2')X,$$

*where $W_1^2 W_1^1$ and $W_2^2 W_2^1$ are two independent Rademacher random variables. So, $Z = 0$ with probability $1/2$. So, $Z$ is not Gaussian.*

In this example, we build a non-Gaussian random variables with a minimal neural network, in which we construct two dependent random variables. So, we should pay attention to this phenomenon when propagating the pre-activations in a neural network.

One should note that the structure of dependence of $W_1^2 W_1^1 X$ and $W_2^2 W_2^1 X$ does not involve their correlation (which is zero), and yet breaks the Gaussianity of their sum. So, in order to obtain a theoretical result about the distribution of $Z$, we should study finer aspects of the dependence structure.

But, on the practical side, we want to answer the question: to which extent does the relation of dependence between the pre-activations $(Z_j^l)_j$ affect the Gaussianity of $(Z_i^{l+1})_i$?

In order to answer this question, we propose a series of experiments on a two-layer neural network. We consider a vector of inputs $\mathbf{X} \in \mathbb{R}^{n_0}$, where the $(X_j)_{1 \leq j \leq n_0}$ are $\mathcal{N}(0,1)$ and i.i.d. The scalar outputted by the network is:

$$Z = \frac{1}{\sqrt{n_1}} \mathbf{W}^2 \phi \left( \frac{1}{\sqrt{n_0}} \mathbf{W}^1 \mathbf{X} \right),$$

where the weights $\mathbf{W}^1 \in \mathbb{R}^{n_1 \times n_0}$ and $\mathbf{W}^2 \in \mathbb{R}^{1 \times n_1}$ are i.i.d.

We test this setup with different initialization distributions P and activation functions $\phi$:

- usual ones: $\phi = $ tanh or ReLU, $P = \mathcal{N}(0, \sigma_w^2)$, where $\sigma_w^2$ is such that the pair $(\sigma_w^2, \sigma_b^2 = 0.01)$ lies at the EOC;

- ours: $\phi = \phi_\theta$, $P = P_\theta = \mathcal{W}(\theta, 1)$.

We study three cases:

- unfavorable case: $n_0 = 1$, various $n_1 \in [1, 10]$; the intermediary features $\mathbf{W}^1 \mathbf{X}$ are weakly "mixed", so it is credible that they lead to an output that is far from being Gaussian;

- favorable case: $n_1 = 2$, various $n_0 \in [1, 10]$;[21] the intermediary features $\mathbf{W}^1 \mathbf{X}$ are "mixed" with an increasing rate as $n_0$ increases;

- same-width case: $n_0 = n_1$.

According to Figure 11:

- unfavorable case (1st graph, $n_0 = 1$): $Z$ is far from being Gaussian, regardless of the activation function. One should note that tanh and $(P_\theta, \phi_\theta)$ with small $\theta$ lead a distribution of $Z$ that is closer to the Gaussian than with ReLU and $(P_\theta, \phi_\theta)$ with large $\theta$;

- favorable case (2nd graph, $n_1 = 2$): $Z$ is closer to be Gaussian with out setup $(P_\theta, \phi_\theta)$ than with $\phi = $ tanh or ReLU, especially with larger $n_1$;

- same-width case (3rd graph, $n = n_0 = n_1$): the larger the width $n$, the closer $Z$ is to being Gaussian. For a fixed $n$, the distribution of $Z$ is closer to a Gaussian with smaller $\theta$. When we use $(P_\theta, \phi_\theta)$, we are close to the performance of tanh or better.

---

[21]The case $n_1 = 1$ is trivial: there is no sum of dependent random variables in the second layer.

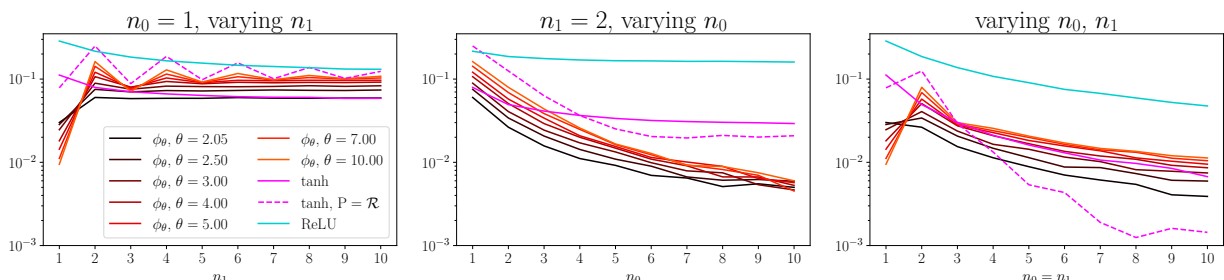

KS-statistic for various layer sizes in a 2-layer NN

Figure 11 – Evolution of the distance of the standardized distribution of $Z$ to the $\mathcal{N}(0,1)$ according to the Kolmogorov-Smirnov statistic. Weight initialization is $\mathcal{W}(\theta,1)$ when using $\phi = \phi_\theta$ and is Gaussian according to the EOC when using $\phi = \tanh$ or ReLU. For each point, we have computed the KS-statistic over $200\,000$ samples of $(\mathbf{X}, \mathbf{W}^1, \mathbf{W}^2)$.

The case $n_0 = n_1$ is the most realistic one: usually, the sizes of the layers of a neural network are of the same order of magnitude. In this case, our setup is better than or equivalent to the one with tanh.

**Remark 14** (Mixing of inputs and interference phenomenon)**.** *We see in Figure 11 that, in every graph, setups with small $\theta$ lead to better results than the ones with large $\theta$. This observation could be explained by Example 1. In this example, the inputs are weakly "mixed": since $X_0$ is multiplied by a Rademacher random variable, it is possible to partially reconstruct $X_0$ after the first layer, and then build destructive interference (leading to an output $Z = 0$ half of the time).*

*So, if we want to avoid this "interference" phenomenon, we should use initialization distributions and activation functions such that every layer "mixes" strongly the inputs. Notably, initialization distributions should be far from being a combination of Dirac distributions. Typically, $\mathrm{P}_{\theta=10}$ is close to the Rademacher distribution (see Fig. 5a).*

**Remark 15** (Case $\phi = \tanh$ and $\mathrm{P} = \mathcal{R}$)**.** *According to Remark 14, the relatively good performance of* tanh *should be explained by the choice of a Gaussian initialization $\mathrm{P}$. However, the case $\phi = \tanh$ and $\mathrm{P} = \mathcal{R}$, shown in Fig. 11 (dotted magenta line), contradicts partially this explanation. In this case, choosing a Rademacher initialization of the weights should lead to worse results than with a Gaussian initialization.*

*It is certainly true in the first graph ($n_0 = 1$), but it is obviously false in the third graph ($n = n_0 = n_1$) for larger n. Actually, the setup $\phi = \tanh$ and $\mathrm{P} = \mathcal{R}$ is better than all the others in this graph.*

So, the contradiction between Remarks 14 and 15 indicates that the dependence between the inputs remains to be investigated in depth. Specifically, we did not take into account the shape of the activation function in Remark 14. For instance, tanh-like functions make the network "forget" information about the input, which may improve the "mixing" process, and thus lead to better results

# C   Constraints on the product of two random variables

**Proposition 4** (Density of a product of random variables at 0). *Let $W, Y$ be two independent non-negative random variables and $Z = WY$. Let $f_W, f_Y, f_Z$ be their respective density. Assuming that $f_Y$ is continuous at 0 with $f_Y(0) > 0$, we have:*

$$\text{if} \quad \lim_{w \to 0} \int_w^\infty \frac{f_W(t)}{t} \, \mathrm{d}t = \infty, \quad \text{then} \quad \lim_{z \to 0} f_Z(z) = \infty. \tag{22}$$

*Moreover, if $f_Y$ is bounded:*

$$\text{if} \quad \int_0^\infty \frac{f_W(t)}{t} \, \mathrm{d}t < \infty, \quad \text{then} \quad f_Z(0) = f_Y(0) \int_0^\infty \frac{f_W(t)}{t} \, \mathrm{d}t. \tag{23}$$

*Proof.* Let $z, z_0 > 0$:

$$
\begin{aligned}
f_Z(z) &= \int_0^\infty f_Y(t) \frac{1}{t} f_W \left( \frac{z}{t} \right) \mathrm{d}t \\
&\geq \int_0^{z_0} f_Y(t) \frac{1}{t} f_W \left( \frac{z}{t} \right) \mathrm{d}t \\
&\geq \inf_{[0, z_0]} f_Y \cdot \int_0^{z_0} \frac{1}{t} f_W \left( \frac{z}{t} \right) \mathrm{d}t \\
&\geq \inf_{[0, z_0]} f_Y \cdot \int_{z/z_0}^\infty \frac{f_W(t)}{t} \, \mathrm{d}t.
\end{aligned}
$$

Let us take $z_0 = \sqrt{z}$. We have:

$$f_Z(z) \geq \inf_{[0, \sqrt{z}]} f_Y \cdot \int_{\sqrt{z}}^\infty \frac{f_W(t)}{t} \, \mathrm{d}t.$$

Then we take the limit $z \to 0$, hence:

- if $\int_0^\infty \frac{f_W(t)}{t} \, \mathrm{d}t = \infty$, then: $\lim_{z \to 0} f_Z(z) = \infty$, which achieves (22);

- if $\int_0^\infty \frac{f_W(t)}{t} \, \mathrm{d}t < \infty$, then: $f_Z(0) \geq f_Y(0) \int_0^\infty \frac{f_W(t)}{t} \, \mathrm{d}t$, which achieves one half of (23);

Let us prove the second half of (23). Let $z, z_0 > 0$:

$$
\begin{aligned}
f_Z(z) &= \int_0^\infty f_Y(t) \frac{1}{t} f_W \left( \frac{z}{t} \right) \mathrm{d}t \\
&= \int_0^{z_0} f_Y(t) \frac{1}{t} f_W \left( \frac{z}{t} \right) \mathrm{d}t + \int_{z_0}^\infty f_Y(t) \frac{1}{t} f_W \left( \frac{z}{t} \right) \mathrm{d}t \\
&\leq \sup_{[0, z_0]} f_Y \cdot \int_{z/z_0}^\infty \frac{f_W(t)}{t} \, \mathrm{d}t + \int_1^\infty f_Y(z_0 t) \frac{1}{t} f_W \left( \frac{z}{z_0 t} \right) \mathrm{d}t
\end{aligned}
$$

Let $z_0 = \sqrt{z}$. We have:

$$f_Z(z) \leq \sup_{[0,\sqrt{z}]} f_Y \cdot \int_{\sqrt{z}}^{\infty} \frac{f_W(t)}{t} \, \mathrm{d}t + \int_1^{\infty} f_Y\left(\sqrt{z}t\right) \frac{1}{t} f_W\left(\frac{\sqrt{z}}{t}\right) \mathrm{d}t,$$

where:

$$\int_1^{\infty} f_Y\left(\sqrt{z}t\right) \frac{1}{t} f_W\left(\frac{\sqrt{z}}{t}\right) \mathrm{d}t \leq \|f_Y\|_{\infty} \int_1^{\infty} \frac{1}{t} f_W\left(\frac{\sqrt{z}}{t}\right) \mathrm{d}t$$

$$\leq \|f_Y\|_{\infty} \int_0^{\sqrt{z}} \frac{f_W(t)}{t} \, \mathrm{d}t.$$

According to the hypotheses, we have, as $z \to 0$:

$$\sup_{[0,\sqrt{z}]} f_Y \to f_Y(0)$$

$$\int_{\sqrt{z}}^{\infty} \frac{f_W(t)}{t} \, \mathrm{d}t \to \int_0^{\infty} \frac{f_W(t)}{t} \, \mathrm{d}t$$

$$\int_0^{\sqrt{z}} \frac{f_W(t)}{t} \, \mathrm{d}t \to 0,$$

hence the result. $\qquad\qquad\qquad\qquad\qquad\qquad\qquad\qquad\qquad\qquad\qquad\qquad\square$

## D    Activation functions with vertical tangent at $0$

In the following lemma, we show that if we want the activation $Y$ to have a density that is $0$ at 0, then the activation function $\phi$ should have a vertical tangent at 0. $G$ plays the role of pre-activation.

**Lemma 2.** *Let $\phi$ be a function transforming a Gaussian random variable $G \sim \mathcal{N}(0,1)$ into a symmetrical random variable $Y$ with a density $f_Y$ such that $f_Y(0) = 0$. That is, $Y = \phi(G)$. Then $\phi$ has a vertical tangent at 0.*

*Proof.* We have:

$$\phi(x) = F_Y^{-1}(F_G(x)),$$

where $F_G$ and $F_Y$ are the respective CDFs of $G$ and $Y$.

Thus:

$$\phi'(x) = F_G'(x) \frac{1}{F_Y'(F_Y^{-1}(F_G(x)))}$$

Therefore:

$$\phi'(0) = F'_G(0) \frac{1}{F'_Y(F_Y^{-1}(F_G(0)))}$$

$$= \frac{1}{\sqrt{2\pi}} \frac{1}{F'_Y(F_Y^{-1}(1/2))}$$

$$= \frac{1}{\sqrt{2\pi}} \frac{1}{F'_Y(0)}$$

$$= \infty.$$

$\square$

# E    The Mellin transform

## E.1    Generalities

We assume that $G = WY \sim \mathcal{N}(0,1)$. Let us consider the random variables $|W|$, $|Y|$ and $|G| = |W| \cdot |Y|$. Let $f_{|W|}$, $f_{|Y|}$ and $f_{|G|}$ be their densities. Under integrability conditions, we can express the density $f_{|G|}$ of the product $|G| = |W||Y|$ with the product-convolution operator $\dot{*}$:

$$f_{|G|}(z) = (f_{|W|} \dot{*} f_{|Y|})(z), \quad \text{where} \quad (f_{|W|} \dot{*} f_{|Y|})(z) = \int_0^\infty f_{|W|}\left(\frac{z}{t}\right) f_{|Y|}(t) \frac{1}{t} \, dt.$$

We can also express the CDF of $|G|$ this way:

$$F_{|G|}(z) = \int_0^\infty F_{|W|}\left(\frac{z}{t}\right) f_{|Y|}(t) \, dt. \tag{24}$$

Then, we can use the following property of the Mellin transform $\mathcal{M}$:

$$\mathcal{M} f_{|G|} = (\mathcal{M} f_{|W|}) \cdot (\mathcal{M} f_{|Y|}), \quad \text{where} \quad (\mathcal{M} f)(t) = \int_0^\infty x^{t-1} f(x) \, dx.$$

In short, $\mathcal{M}$ transforms a product-convolution into a product in the same manner as the Fourier transform $\mathcal{F}$ transforms a convolution into a product. We have then:

$$f_{|Y|}(y) := \mathcal{M}^{-1}\left[\frac{\mathcal{M} f_{|G|}}{\mathcal{M} f_{|W|}}\right](y).$$

Then, by symmetry, we can obtain $f_Y$ from $f_{|Y|}$. However, while $\mathcal{M} f_{|G|}$ and $\mathcal{M} f_{|W|}$ are easy to compute, the inverse Mellin transform $\mathcal{M}^{-1}$ seems to be analytically untractable in this case:

$$(\mathcal{M} f_{|G|})(s) = \frac{2^{\frac{s}{2}-\frac{1}{2}} \Gamma(\frac{s}{2})}{\sqrt{\pi}}, \qquad (\mathcal{M} f_{|W|})(s) = \Gamma\left(\frac{s-1}{\theta} + 1\right),$$

$$\text{so:} \quad \frac{(\mathcal{M} f_{|G|})(s)}{(\mathcal{M} f_{|W|})(s)} = \frac{1}{\sqrt{\pi}} \frac{2^{\frac{s}{2}-\frac{1}{2}} \Gamma(\frac{s}{2})}{\Gamma\left(\frac{s-1}{\theta} + 1\right)}.$$

## E.2 Numerical inversion of the Mellin transform

**Computation of $f_{|Y|}$ by numerical inverse Mellin transform.** The Mellin transform of a function can be inverted by using Laguerre polynomials. Specifically, we use the method proposed by Theocaris & Chrysakis (1977) and slightly accelerated by the numerical procedure of Gabutti & Sacripante (1991):

$$(\mathcal{M}^{-1}f)(z) = e^{-\frac{z}{2}} \sum_{k=0}^{\infty} c_{k+1} L_k\left(\frac{z}{2}\right), \quad \text{with } c_k := \sum_{n=1}^{k} \binom{k-1}{n-1}(-1)^{n-1}\frac{f(n)}{2^n \Gamma(n)}, \qquad (25)$$

where the $(L_k)_k$ are the Laguerre polynomials (see Section 7.41, Gradshteyn & Ryzhik, 2014).

**Experiments.** A common way of computing the inverse Mellin transform consists of using Equation (25). Specifically, the sequence $(c_k)_k$ must be computed.

In order to compute the density $f_Y$ of $Q_\theta$ with $\theta = 2.05$, we have computed numerically $(c_k)_k$ for $k \in [1, 500]$. The results are plotted in Figure 12. We have tested three methods to compute the $c_k$:

- floating-point operations using 64 bits floats;

- floating-point operations using 128 bits floats;

- SymPy: make the whole computation using SymPy, a Python library of symbolic computation (very slow).

In all three cases, instabilities appear before the sequence $(c_k)_k$ has fully converged to 0. Moreover, the oscillations of $(c_k)_k$ around 0 have an increasing wavelength, which indicates that we may have to go far beyond $k = 500$ to get enough coefficients $(c_k)_k$ to reconstruct the wanted inverse Mellin transform.

In Figure 13, we have plotted two estimations of $f_{|Y|}$: the density obtained directly by using $(c_k)_{k \in [1,300]}$, and the density obtained by using a sequence $(c_k)_{k \in [1,20000]}$ where the values $(c_k)_{k \in [301,20000]}$ have been extrapolated from $(c_k)_{k \in [1,300]}$.[22]

The resulting estimations of the density $f_{|Y|}$ take negative values and seem to be noised. So, more work is needed to obtain smooth and proper densities, especially if we want them to meet Constraints 3 and 4 (density at 0 and decay rate at $\infty$).

**Conclusion.** We observe that this computation of the inverse Mellin transform has several intrinsic problems:

- the computation of the $c_k$ coefficients involves a sum of terms with alternating signs, which become larger (in absolute value) as $k$ grows and which are supposed to compensate such that $c_k \to 0$ as $k \to \infty$. Such a numerical computation, involving both large and small terms, makes the resulting $c_k$ very unstable as $k$ grows;

---

[22]The extrapolation has been performed by modeling the graph of $(c_k)_k$ as the product of a decreasing function and a cosine with decreasing frequency.

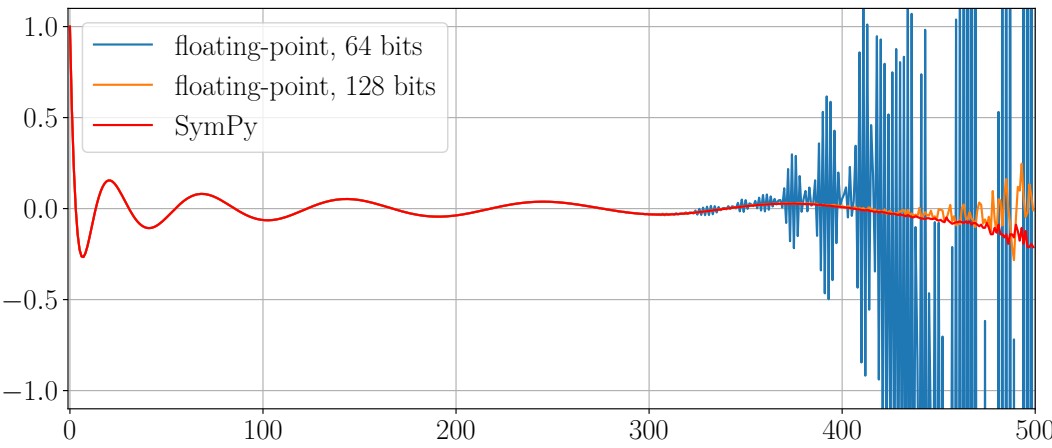

Figure 12 – Evolution of various numerical computations of $c_k$ as $k$ grows.

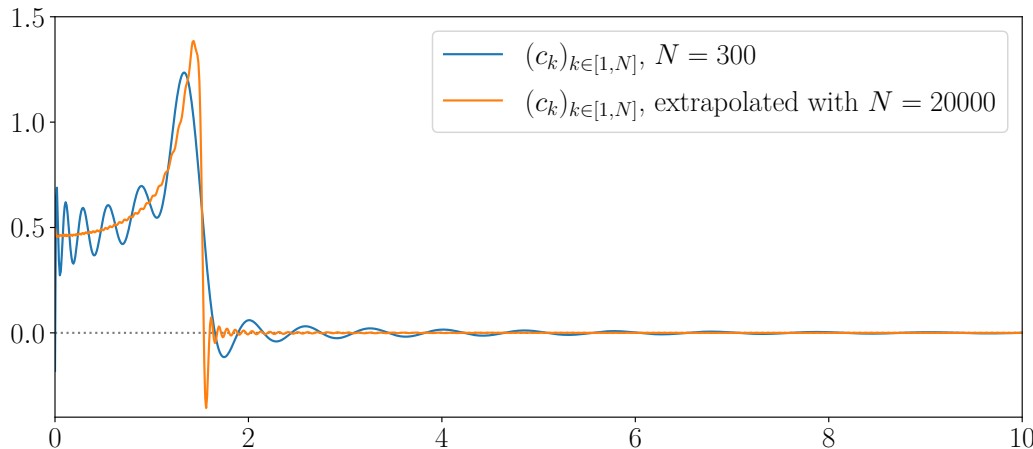

Figure 13 – Numerical inverse Mellin transform with two different computations of $(c_k)_k$: direct computation of $(c_k)_{k \in [1,N]}$ with $N = 300$; extrapolation of $(c_k)_{k \in [301, 20000]}$ from $(c_k)_{k \in [1,300]}$.

- when $\theta \approx 2$, the sequence $(c_k)_k$ tends extremely slowly to 0;

- if we approximate $\mathcal{M}^{-1}f$ with the finite sum of the first $K$ terms of the series in Equation (25), we cannot guarantee the non-negativeness of the resulting function, which is meant to be a density.

So, this method is unpractical to compute the density of a distribution in our case.

# F    Obtaining $f_{|Y|}$: experimental details

We optimize the vector of parameters $\Lambda$ with respect to the following loss:

$$\ell(\Lambda) := \|\hat{F}_\Lambda - F_{|G|}\|_\infty$$

$$\hat{F}_\Lambda(z) := \int_0^\infty F_{|W|}\left(\frac{z}{t}\right) g_\Lambda(t)\,\mathrm{d}t.$$

**Dataset.**    We build the dataset $\mathcal{Z}$ of size $d$:

$$\mathcal{Z} = \left\{0, z_{\max}\frac{1}{d-1}, z_{\max}\frac{2}{d-1}, \cdots, z_{\max}\right\}.$$

In our setup, $d = 200$ and $z_{\max} = 5$.

**Computing the loss.**    For each $z$ in $\mathcal{Z}$, we compute numerically $\hat{F}_\Lambda(z)$. Then, we are able to compute $\ell(\Lambda)$. We keep track of the computational graph with PyTorch, in order to backpropagate the gradient and train the parameters $\Lambda$ by gradient descent.

**Initialization of the parameters.**    We initialize $\Lambda = (\alpha, \gamma, \lambda_1, \lambda_2)$ in the following way:

$$\alpha = 3, \quad \gamma = 1, \quad \lambda_1 = 1, \quad \lambda_2 = 1.$$

**Optimizer.**    We use the Adam optimizer (Kingma & Ba, 2015) with the parameters:

- learning rate: 0.001;

- $\beta_1 = 0.9$;

- $\beta_2 = 0.999$;

- weight decay: 0.

We train $\Lambda$ for 100 epochs.

**Learning rate scheduler.**    We use a learning rate scheduler based on the reduction of the training loss. If the training loss does not decrease at least by a factor 0.01 for 20 epochs, then the learning rate is multiplied by a factor $1/\sqrt[3]{10}$. After a modification of the learning rate, we wait at least 20 epochs before any modification.

**Scheduler for $\theta'$.**    We recall that the definition of $g_\Lambda$ involves $\theta'$, defined by: $\frac{1}{\theta} + \frac{1}{\theta'} = \frac{1}{2}$. It is not a parameter to train. Empirically, we found that the following schedule improves the optimization process:

- from epoch 0 to epoch 49, $\theta'$ increases linearly from 2 to its theoretical value $(\frac{1}{2} - \frac{1}{\theta})^{-1}$;

- at the beginning of epoch 50, we reinitialize the optimizer and the learning rate scheduler;

- we finish the training normally, with $\theta' = (\frac{1}{2} - \frac{1}{\theta})^{-1}$.

# G  The Kolmogorov–Smirnov test

**Description.** We describe here the Kolmogorov–Smirnov (KS) test (Kolmogoroff, 1941; Smirnov, 1948). Given a sequence $(Z'_1, \cdots, Z'_s)$ of $s$ i.i.d. random variables sampled from $\mathrm{P}'$:

1. we build the empirical CDF $F_s$ of this sample:

$$F_s(z) = \frac{1}{s} \sum_{k=1}^{s} \mathbb{1}_{Z'_k \leq z};$$

2. we compare $F_s$ to the CDF $F_G$ of $G \sim \mathcal{N}(0, 1)$ by using the $\mathcal{L}^\infty$ norm:

$$D_s = \|F_s - F_G\|_\infty,$$

where $D_s$ is the "KS statistic";

3. under the null hypothesis, i.e. $\mathrm{P}' = \mathcal{N}(0, 1)$, we have:

$$\sqrt{s} D_s \xrightarrow{d} K,$$

where $K$ is the Kolmogorov distribution (Smirnov, 1948). We denote by $(K_\alpha)_\alpha$ the quantiles of $K$:

$$\mathbb{P}(K \leq K_\alpha) = 1 - \alpha, \quad \text{for all } \alpha \in [0, 1];$$

4. finally, we reject the null hypothesis at level $\alpha$ if:

$$\sqrt{s} D_s \leq K_\alpha.$$

**Limitations.** In the KS test presented above, the null hypothesis $\mathbb{H}_0$ is $\mathrm{P}' = \mathcal{N}(0, 1)$, which is exactly what we intend to demonstrate when using our family $\{(\mathrm{P}_\theta, \phi_\theta) : \theta \in (2, \infty)\}$, while the alternative hypothesis $\mathbb{H}_1$ is $\mathrm{P}' \neq \mathcal{N}(0, 1)$. With this test design, we face a problem: it is impossible to claim that $\mathbb{H}_0$ holds. More precisely, we have two possible outcomes: either $\mathbb{H}_0$ is rejected, or it is not. Since $\mathbb{H}_1$ is the complementary of $\mathbb{H}_0$, a reject of $\mathbb{H}_0$ means an accept of $\mathbb{H}_1$. But the converse does not hold: if $\mathbb{H}_0$ is not rejected, then it is impossible to conclude that $\mathbb{H}_0$ is true: the sample size $s$ may simply be too small, or the KS test may be inadequate for our case.

So, to be able to conclude that the pre-activations are Gaussian, we should have built an alternative test, where the null hypothesis $\mathbb{H}'_0$ is some *misfit* between P′ and $\mathcal{N}(0,1)$. Or, at least, we should have computed the power of the current KS test.

However, our experimental results are sufficient to *compare* the quality of fit between P′ and $\mathcal{N}(0,1)$ in the tested setups (see Figures 6 and 7). It remains clear that, with $\{(P_\theta, \phi_\theta) : \theta \in (2, \infty)\}$, the Gaussian pre-activations hypothesis is far more likely to hold than in setups involving tanh, ReLU and Gaussian weights.

## H  Experiments

### H.1  Propagation of the correlations with MNIST

Within the setup of Section 2.2, we have plotted in Figure 14 the correlations propagated in a multilayer perceptron with inputs sampled for MNIST (10 samples per class, that is, 100 samples in total). The layers of the perceptron have $n_l = 10$ neurons each, and the activation function is $\phi = \text{ReLU}$.

We observe that, in this narrow NN case ($n_l = 10$), some irregularities appear as information propagates into the network: some classes seem to (de)correlate in an inconsistent way with the others. Specifically, while most of the classes tend to correlate exactly ($C^* \approx 1$), the third class (corresponding to the digit "2") tends to a correlation $C^* \approx 0.7 \neq 1$. This result is contradictory to the EOC theory.

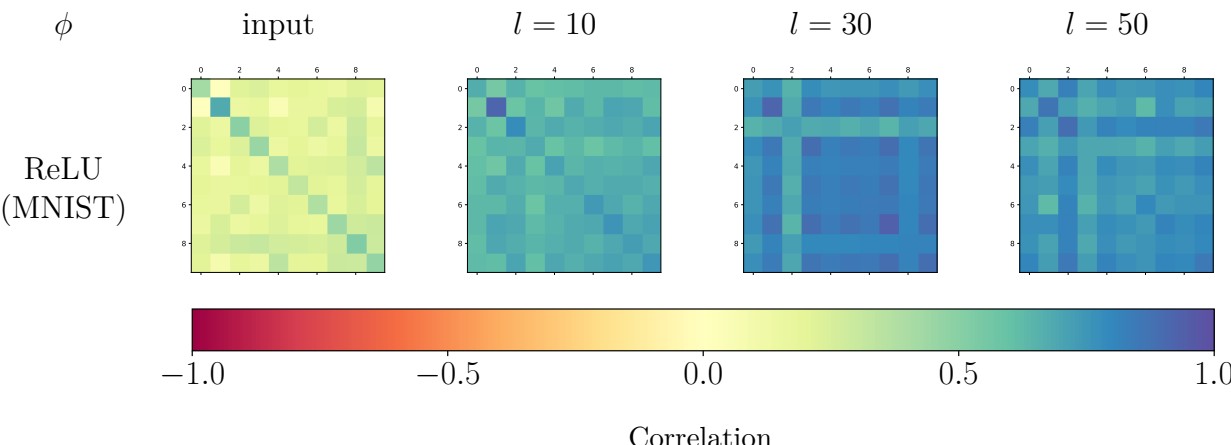

Figure 14 – Propagation of correlations $c^l_{ab}$ in a multilayer perceptron with activation function $\phi = \text{ReLU}$ and inputs sampled from the MNIST dataset. The neural network is initialized at the EOC. Each plot displays a $10 \times 10$ matrix $C^l_{pq}$ whose entries are the average correlation between the pre-activations propagated by samples from classes $p, q \in \{0, \cdots, 9\}$, at the input and right after layers $l \in \{10, 30, 50\}$. See also Figure 1 in Section 2.2 for results on CIFAR-10.

## H.2  Propagation of the correlations with $\phi = \phi_\theta$

Within the setup of Section 2.2, we have plotted in Figure 15 the correlations propagated in a multilayer perceptron with $\phi = \phi_\theta$. The weights have been sampled from $\mathcal{W}(\theta, 1)$.

In this setup, the results are consistent, and are consistent with the results for $\phi = \tanh$ (see Figure 1): the sequence $(C_{pq}^l)_l$ converges to 1, which was not the case for $\phi = \mathrm{ReLU}$ and $n_l = 10$.

## H.3  Variance of the pre-activation when using $\phi = \phi_\theta$

As observed in Figure 6a, the product $W\phi_\theta(X)$ is above the KS threshold, corresponding to $s = 10^7$ samples and a $p$-value of 0.05. This is not necessarily the case in Figure 6b, where the samples are standardized. So, we suspect that the variance of $W\phi_\theta(X)$ is not exactly 1.

Therefore, we have reported in Table 2 the empirical standard deviation of the product $W\phi(X)$ (i.e., $Z'$ with $n = 1$), computed with $s = 10^7$ samples, where $X \sim \mathcal{N}(0, 1)$, $W \sim \mathcal{W}(\theta, 1)$ if $\phi = \phi_\theta$ and $W \sim \mathcal{N}(0, 1)$ if $\phi = \tanh$ or ReLU. We observe that the standard deviation of $Z'$, which is expected to be 1 with $\phi = \phi_\theta$, is actually a bit different. This would largely explain the differences between Figure 6a and Figure 6b.

Table 2 – Empirical standard deviation of $Z'$ with $n = 1$.

| $\theta$ | 2.05 | 2.5 | 3 | $\phi_\theta$ 4 | 5 | 7 | 10 | tanh | ReLU |
|---|---|---|---|---|---|---|---|---|---|
| $\bar{\sigma}$ | 1.003 | 0.994 | 0.989 | 0.984 | 0.981 | 0.979 | 0.978 | 0.628 | 0.707 |

## H.4  Propagation of the pre-activations

In this subsection, we show how the choice of the input data points affect the propagation of the pre-activations.

Let $\mathcal{D}^l$ be the distribution of the pre-activation $Z_1^l$ after layer $l$. In Figure 16, we have plotted the $\mathcal{L}^\infty$ distance between the CDFs of $\mathcal{D}^l$ and $\mathcal{N}(0, 1)$ for various input points, sampled from different classes to improve diversity between them. We have chosen the following setup: CIFAR-10 inputs, multilayer perceptron with 100 layers and 100 neurons per layer, Gaussian initialization at the EOC when using the activation function $\phi = \tanh$ or ReLU, and symmetric Weibull initialization $\mathcal{W}(\theta, 1)$ when using $\phi = \phi_\theta$.

This setup has been selected to illustrate clearly the variability of $\mathcal{D}^l$ when using various data points. This variability can be observed with 10 or 1000 neurons per layer, though it is less striking.

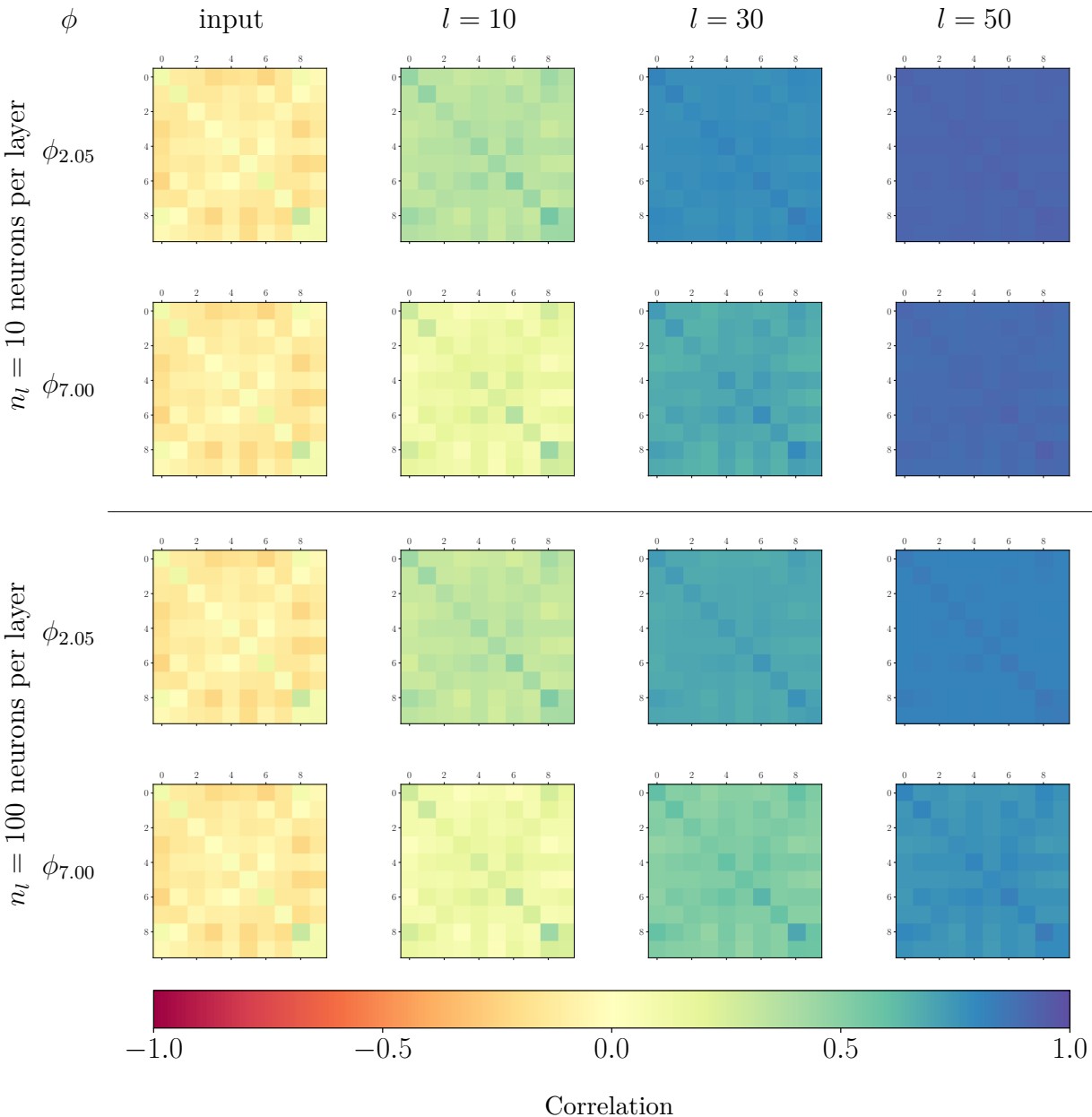

Figure 15 – Propagation of correlations $c_{ab}^l$ in a multilayer perceptron with activation function $\phi_\theta$ with $\theta \in \{2.05, 7.00\}$ and inputs sampled from the CIFAR-10 dataset. The weights are sampled from $\mathcal{W}(\theta, 1)$ and the biases are zero. Each plot displays a $10 \times 10$ matrix $C_{pq}^l$ whose entries are the average correlation between the pre-activations propagated by samples from classes $p, q \in \{0, \cdots, 9\}$, at the input and right after layers $l \in \{10, 30, 50\}$.

**With input normalization over the whole dataset.** In this setup, we perform the usual normalization over the whole dataset:

$$\hat{x}_{a;ij} := \frac{x_{a;ij} - \mu_i}{\sigma_i},$$

$$\mu_i := \frac{1}{Np_i} \sum_{\mathbf{x} \in \mathbb{D}} \sum_{j=1}^{p_i} x_{ij}$$

$$\sigma_i^2 := \frac{1}{Np_i - 1} \sum_{\mathbf{x} \in \mathbb{D}} \sum_{j=1}^{p_i} (x_{ij} - \mu_i)^2,$$

where $\hat{\mathbf{x}}$ is the normalized data point, $x_{a;ij}$ is the $j$-th component of the $i$-th channel of the input image $\mathbf{x}_a$, $p_i$ is the size of the $i$-th channel, and $N$ is the size of the dataset $\mathbb{D}$.

We observe that, when using the activation function $\phi = \tanh$, the shape of the curves is the same for all data points. This is not the case for ReLU: for the input point "bird", the sequence $(\mathcal{D}^l)_l$ drifts away from $\mathcal{N}(0, 1)$ since the beginning, while for the input "car", $(\mathcal{D}^l)_l$ first becomes closer to $\mathcal{N}(0, 1)$, then drifts away. It is also the case for $\phi_\theta$ with $\theta = 10$: for the input "deer", the distance remains high, while it starts low and increases very slowly (input "dog"), or even decreases in the first place (input "truck").

But, in general, when the curves have converged after 100 layers, it seems that the limit is the same whatever the starting data point, and depends only on the choice of the activation function.

**With individual input normalization.** In this setup, we perform the individual normalization of the inputs. Without loss of generality, we define it for inputs that are order-1 tensors as follows:

$$\hat{x}_{a;i} := \frac{x_{a;i} - \mu}{\sigma},$$

$$\mu := \frac{1}{p} \sum_{i=1}^{p} x_i$$

$$\sigma^2 := \frac{1}{p-1} \sum_{i=1}^{p} (x_i - \mu)^2,$$

where $\hat{\mathbf{x}}$ is the normalized data point, $x_{a;i}$ is the $i$-th component of $\mathbf{x}_a \in \mathbb{R}^p$.

According to Figure 17, the curves seem to be approximately identical for each data point, contrary to the setup with normalization over the whole dataset. And, naturally, the distance between $\mathcal{D}^1$ and $\mathcal{N}(0, 1)$ is close to 0 when using the activation functions $\phi_\theta$, which confirms that our method to build Gaussian pre-activations is efficient at least in the beginning.

**Conclusion.** In the specific case of CIFAR-10 images, it seems that the trajectory of the sequence $(\mathcal{D}^l)_l$ depends on the inputs, but only to the extent that they have different norms. As it can be seen in Figure 17, images from different classes which have been normalized individually lead to the same trajectories of distribution of pre-activations.

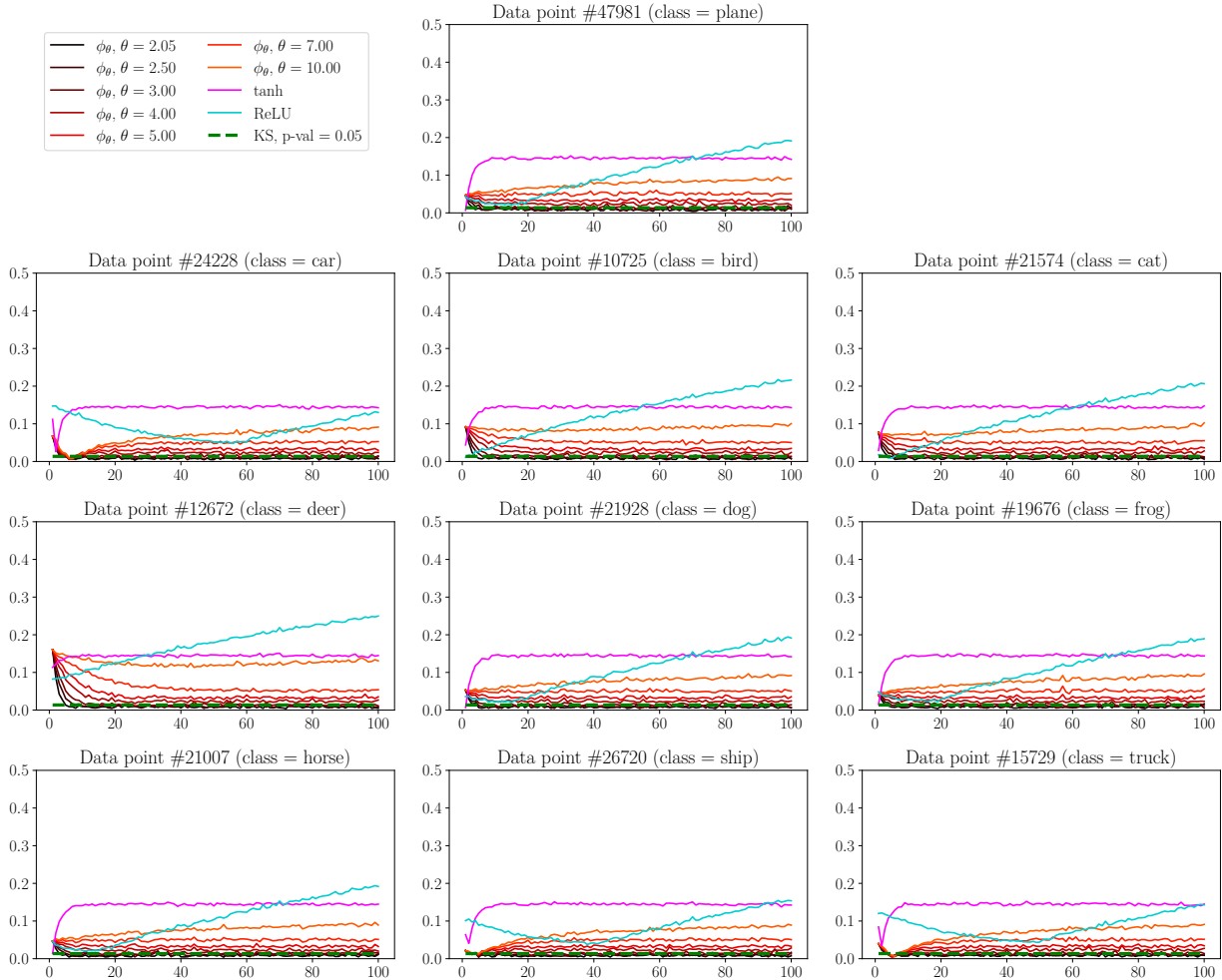

Figure 16 – Setup with data points **normalized over the whole dataset**.
Propagation of the distribution $\mathcal{D}^l$ of the pre-activations across the layers $l \in [1, 100]$ when inputting various data points of CIFAR-10. Each curve represents the $\mathcal{L}^\infty$ distance between the CDF of $\mathcal{D}^l$ and the CDF of the Gaussian $\mathcal{N}(0, 1)$. The dashed green line is the threshold of rejection of the Kolmogorov–Smirnov test with $p$-value 0.05: if a distribution $\mathcal{D}^l$ is represented by a point above this threshold, then the hypothesis "$\mathcal{D}^l = \mathcal{N}(0, 1)$" is rejected with $p$-value 0.05.

## H.5  Experimental details of the training procedure

**Training, validation, and test sets.**  For MNIST and CIFAR-10, we split randomly the initial training set into two sets: the training set, which will be actually used to train the neural network, and the validation set, which will be used to stop training when the network begins to overfit.

The sizes of the different sets are as follows:

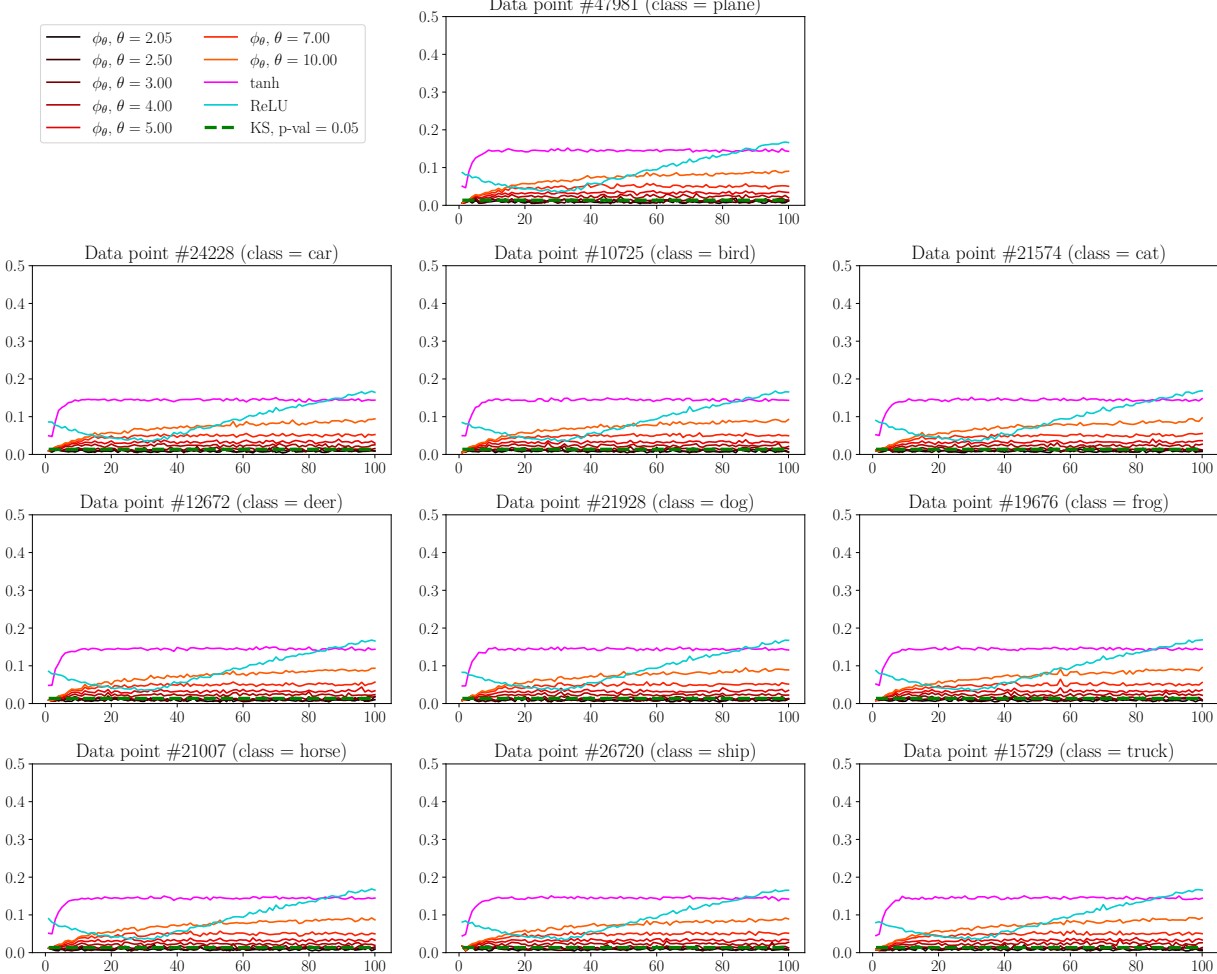

Figure 17 – Setup with data points **normalized individually**.
The rest of the setup is identical to the one used in Figure 16.

- MNIST: 50000 training samples; 10000 validation samples; 10000 test samples;

- CIFAR-10: 42000 training samples, 8000 validation samples; 10000 test samples.

The training sets are split into mini-batches with 200 samples each. No data augmentation is performed.

**Loss.**  Given a classification task with $P$ classes, let $\mathbf{z}^L \in \mathbb{R}^P$ be the pre-activation outputted by the last layer of the neural network. First, we perform a softmax operation:

$$y_p := \text{softmax}(z_p^L) = \frac{\exp(z_p^L)}{\sum_{p'=1}^{P} \exp(z_{p'}^L)}.$$

where the $(y_p)_p$ are the components of $\mathbf{y} \in \mathbb{R}^P$ and $(z_p^L)_p$ are the components of $\mathbf{z}^L$. Then, we compute the negative log-likelihood loss. For a target class $p \in \{1, \cdots, P\}$, we pose:

$$\ell(\mathbf{y}, p) := -\log(y_p).$$

**Optimizer.**   We use the Adam optimizer (Kingma & Ba, 2015) with the parameters:

- learning rate: 0.001;

- $\beta_1 = 0.9$;

- $\beta_2 = 0.999$;

- weight decay: 0.

**Learning rate scheduler.**   We use a learning rate scheduler based on the reduction of the training loss. If the training loss does not decrease at least by a factor 0.01 for 10 epochs, then the learning rate is multiplied by a factor $1/\sqrt{10}$.

**Early stopping.**   We add an early stopping rule based on the reduction of the validation loss. If the validation loss does not decrease at least by a factor of 0.001 for 30 epochs, then we stop training.

