# OpenReview forum: "Gaussian Pre-Activations in Neural Networks: Myth or Reality?"
_TMLR — Rejected by TMLR_

### Review · Reviewer_D3ec · 2023-02-05

**Summary Of Contributions:**

The paper considers the task of designing an initialization scheme, together with a suitable activation function, such that the pre-activation at initialization is Gaussian for any width and depth. The main contribution of the paper is the proposal of a class of initialization schemes with symmetric Weibull weights, derived from a set of constraints to address this task.

**Audience:**

Yes

**Claims And Evidence:**

No

**Requested Changes:**

A fix to Proposition 4 would be important.

**Strengths And Weaknesses:**

The paper addresses a relevant problem that has been lacking a definite answer, and this problem roots from a large literature behind the Edge of Chaos (EOC) phenomenon. In particular, recent findings point to the fact that pre-activations in practice tend to be heavy-tailed than Gaussian, which poses a serious challenge to supporters of EOC literature. The paper aims to avoid the two main obstacles: the width which typically is assumed to be very large and the depth which typically is assumed to be comparably smaller in the EOC literature. As such, a resolution to this problem, if correct, would be a very good contribution.

Some of the results in the paper, in particular those in Section 2, are actually well known in the community. The paper Poole et el 2016 is known to have several loose claims, including the one claim addressed in this paper. So Section 2 does not convey new information, though it serves as a good context for the main contribution in Section 3 where the paper derives its initialization and activation function design.

The main flaw I can see in the paper is actually in Section 3, in particular, Proposition 4. The proof of this proposition claims that the sum of random variables whose marginal distributions are Gaussian must be Gaussian. This is not necessarily true.

Now let me give a counter argument to the statement of Proposition 4. According to this proposition, if $W_1$, $W_2$, $Y_1$ and $Y_2$ are such that $W_1$, $W_2$ and $(Y_1, Y_2)$ are mutually independent ($Y_1$ and $Y_2$ can be dependent), $W_1Y_1$ and $W_2Y_2$ are $N(0,1)$, then $Z = W_1Y_1 + W_2Y_2$ is Gaussian. Now Figure 5 suggests that I can take $W_1$ and $W_2$ to be Rademacher ($\theta=\infty$),  and $Y_1$ and $Y_2$ assume some continuous distribution so that  $W_1Y_1$ and $W_2Y_2$ are Gaussian $N(0,1)$. Since the proposition allows arbitrary dependency between $Y_1$ and $Y_2$, let me take $Y_1=Y_2=Y$. So now the sum is $Z = (W_1 + W_2)Y$. Note that $W_1+W_2=0$ with probability $1/2$. So with probability at least $1/2$, $Z=0$. So $Z$ cannot be a Gaussian with non-zero variance.

This proposition is actually a central step in the entire argument: it is the one step where the paper bypasses the complication due to width and reduces the problem to a much simpler problem given in (15). Indeed treating the dependency among the neurons (exemplified by $Y_1$ and $Y_2$ in my counter argument) at a finite width is the most difficult part. I suspect this is one large cause for the disappointing results in Figures 6 and 7, which fail to show that the design in the paper can consistently maintain Gaussianity.

---

> ### Author Response · Authors · 2023-03-03
> **Answer for Reviewer D3ec**
>
> We thank Reviewer D3ec for checking carefully our formal statements and proposing a realistic counter-example.
> We propose a general answer in the section *Independence of the pre-activations and "Proposition 4"* of our main answer, and we propose in Appendix B a series of experiments exploring the problem of dependent pre-activations.

---

### Review · Reviewer_QEpf · 2023-02-20

**Summary Of Contributions:**

The paper is concerned with what they call the "Gaussian hypothesis", i.e., the assumption that the inputs to each layer of a neural network are approximately Gaussian. This assumption is found in many theoretical works, especially when moving to the infinite-width limit where it is justified by central limit theorems.

In this paper the authors take a different approach, and they study how to impose this constraint in the finite-width regime. The bulk of the paper is devoted to deriving a family of pairs of (initialization distribution, activation function) which approximately preserve this property. Some representative members of this family are then tested on some simple benchmarks, including MNIST and CIFAR-10. The experiments show that, while the Gaussian properties are much better than using ReLUs or tanh, the empirical results tend to be poorer.

**Audience:**

Yes

**Broader Impact Concerns:**

This is a theoretical paper proposing a construction for ensuring a property at initialization. No broader impact is necessary.

**Claims And Evidence:**

Yes

**Requested Changes:**

I have a few minor comments that might (or not) be integrated into a final version.

1) I suggest that the authors formalize the "Gaussian hypothesis" at some point in the text explicitly, possibly even in Section 1. Variants of this statement are provided several times in the text (p6, p7, p8), and it is unclear if they always refer to the same hypothesis.

2) "have already converged to the same limit" (p7). I assume this is converged for $l \rightarrow \infty$? This is also stated later on p12.

3) On the section "Bayesian prior and initialization distribution": I feel this paragraph is weak. In particular, the relation between initialization and Bayesian NNs is stronger when using Monte Carlo techniques, since in that case the prior is how we sample values for the network, and how to choose a good prior is a well-investigated task (e.g., [TRMF22]). For VI, the prior acts more as a regularization term, while the task of initializing the network is similar to the one faced in non-Bayesian training. This is also been partly investigated in its own literature [RMF19]. I would suggest rewriting this paragraph in light of these considerations or maybe removing it (is it connected somehow to the rest of the paper?).

4) "sample the parameters according to the EOC": can you clarify how? If I understand correctly, you followed procedures shown in footnote 7? Is this the same procedure mentioned on p26?

5) The Kolmogorov–Smirnov test: this is used in Section 2 but described in Section 4.1. A reference could be made in Section 2, or maybe Section 4.1 could be moved to the appendix and referenced in both sections.

6) On the Section "Computation of $f_{|Y|}$ by numerical inverse Mellin transform." Since this is not used, maybe it can be moved in the appendix to simplify reading.

7) A table with common notation can be added (e.g., I kept getting confused about the meaning of $\phi$ and its variants through the text).

Smaller typographic changes:
P1: Upper-case letters $\mathbf{W}$, should W be bold?
P2: and should -> and how should
P8: with datasets --> with dataset
P25: Our construction generates into the following two extreme cases --> degenerates?

[KUM17] https://arxiv.org/pdf/1706.02515.pdf
[RMF19] https://proceedings.mlr.press/v97/rossi19a.html
[STT20] https://arxiv.org/abs/2004.09506
[TRMF22] https://jmlr.org/papers/volume23/20-1340/20-1340.pdf

**Strengths And Weaknesses:**

I think this is a very good paper that should be published in TMLR. It is a novel take on an old problem (weight initialization in neural networks). While the results are not particularly good, the derivation is interesting and the authors make use of sophisticated mathematical tools that might be of broader interest to the community. I also think they made a strong job in framing their method into a broader framework for selecting the initialization strategy (e.g., Table 1). The paper is generally well written and, despite the complexity of some derivations, it can be followed easily.

The main weaknesses I can identify can be solved easily by small additions or content rewriting, which is why they are listed in the next section. Apart from them, I list here some generic comments for consideration, mostly concerning related works.

1) There are examples of initialization methods that do not rely on preservation of some statistics (e.g., [STT20]). Maybe these could be mentioned in the text.

2) The paper only considers initialization for networks having ReLUs and tanh activation functions. Many other proposals have been made in the literature, sometimes along similar lines (e.g., SELU [KUM17] preserves the mean/std mapping for many layers). These may be added to the related works.

3) The paper claims to "construct a family of pairs of activation functions and initialization distributions that ensure that the pre-activations remain Gaussian throughout the network’s depth, even in narrow neural networks". But if I understand the paper correctly, this is only approximately true because of the computation of the activation function. This is never mentioned explicitly until much later in the paper.

4) Using the results of the paper appears complex, especially due to the derivation of $\phi_\theta$. Are you planning to release some usable code to this end?

---

> ### Author Response · Authors · 2023-03-03
> **Answer for Reviewer QEpf (1/2)**
>
> We thank Reviewer QEpf for the detailed review and the propositions of improvements.
>
> ## Initialization methods that do not rely on the preservation of some statistics (Rev. QEpf)
>
> *Revisiting Weight Initialization of Deep Neural Networks*, Skorski et al., 2021.
>
> We thank the reviewer for indicating this type of initialization criterion. This paper proposes an init method based on the norm of the Hessian of a NN at init.
> It is true that, apparently, this method is not based on the preservation of some statistic.
> However, the *practical* integration of the method involves an approximation of the Hessian (see Theorem 7, "Hessian factorized into Jacobians"). Specifically, the authors recommend to check that the tensors $\frac{\partial z^{(L)}}{\partial z^{(l)}}$ and $\frac{\partial z^{(l)}}{\partial z^{(0)}}$ are of norm $1$.
> So, the preservation of a statistic is involved in this way:
>
>  * the (back-)propagated distributions are the distributions of $z^{(l)}$ and of $\frac{\partial z^{(L)}}{\partial z^{(l)}}$;
>  * from these distributions, it is possible to deduce the norms mentioned above.
>
> But we agree with Reviewer QEpf that this procedure does not fit perfectly the formalism we propose in Section 2.4 and Figure 4, since we do not take into account backpropagation.
>
>
> ## SELU activation functions (Rev. QEpf)
>
> *Self-Normalizing Neural Networks*, Klambauer et al., 2017.
>
> We thank the reviewer for the suggestion. Indeed, the idea proposed in this paper is closely related to the Edge of Chaos line of works: instead of looking for the stable fixed points of the variance $\mathcal{V}$ of the pre-activations for a given activation function (as in the EOC), the authors design the scaling of their activation function to obtain a given fixed point for $\mathcal{V}$.
>
> So, we have included two sentences about this paper in our introduction.
>
>
> ## Initialization distribution and Bayesian prior (Rev. 7n9i and QEpf)
>
> We agree with Reviewers 7n9i and QEpf: the influence of a Bayesian prior can be interpreted as regularization. For instance, the standard loss in variational inference (which is an approximate Bayesian method) is [Graves2011]:
>
> $$L(\beta) = - \frac{1}{N} \mathbb{E}\_{\theta \sim \beta} \sum\_{i=1}^N \log p(x_i, y_i | \theta) + \frac{1}{N} \mathrm{KL}(\beta || \alpha),$$
>
> where $N$ is the size of the dataset $\{(x_i, y_i)\}$, $p$ is the likelihood, $\beta$ is the candidate (variational) posterior and $\alpha$ is the prior. It is then clear that the second term, which only depends on $\beta$, $\alpha$, and $N$, corresponds to the influence of the prior, and is a regularization term at the same time.
>
> In the paragraph "Bayesian prior and initialization distribution" of the introduction, we only claim that a NN sampled from the Bayesian prior should be trainable, exactly as a NN at initialization. This (informal) claim is not contradictory with the fact that the prior plays *also* a regularizing role. A sentence about this regularizing role can possibly be added, if the reviewers want to.
>
> For the current OpenReview discussion, we add the following argument: let us consider the following situation:
>
> 1. we perform Bayesian "online learning" with a sequence of subsets $(D_1, D_2, \cdots)$ of a given dataset;
> 2. we compute the Bayesian posterior $p(\theta | D_1)$ with the first subset $D_1$;
> 3. if we want to get $p(\theta | D_n, \cdots, D_1)$, we just have to compute $p(D_n | \theta)$ and multiply it by $p(\theta | D_{n-1}, \cdots, D_1)$ (up to a normalizing constant).
>
> We apply this setup to NNs, and we approximate each Bayesian posterior $p(\theta | D_n, \cdots, D_1)$ with the variational posterior $\beta_n$ (see [Blier2018], Sec. 3.4). So, each $\beta_n$ is built by training the NN on subset $D_n$ with $\beta_{n-1}$ as a prior distribution. And, from an optimization point of view, it is natural to optimize $\beta_n$ with $\beta_{n-1}$ as a starting point. So, if we adapt the reasoning to the first step, $\beta_1$ should be optimized with the prior $\alpha$ as a starting point.
>
> We admit that this example is not a formal proof that the prior *should* be interpreted as a "by default" initialization distribution (and that is not our claim in the introduction). But it indicates that this interpretation is useful in some circumstances.
>
> Note: this interpretation is not contradictory with the papers mentioned by the Reviewer QEpf: [Tran2022] is about finding a prior matching a condition related to the function represented by a NN; [Rossi2017] proposes a data-dependent initialization (which is out of the scope of our paper).
>
>
> [Blier2018] *The Description Length of Deep Learning Models*, Blier et al., 2018.
>
> [Graves2011] *Practical variational inference for neural networks*, Graves, 2011.
>
> [Rossi2017] *Good Initializations of Variational Bayes for Deep Models*, Rossi, 2017.
>
> [Tran2022] *All You Need is a Good Functional Prior for Bayesian Deep Learning*, Tran et al., 2022.

---

> ### Author Response · Authors · 2023-03-03
> **Answer for Reviewer QEpf (2/2)**
>
> ## Length of the paper (Rev. 7n9i and QEpf)
>
> We understand the concerns of Reviewers 7n9i and QEpf about the length of the paper. We have put the sections about the Kolmogorov-Smirnov test and the Mellin transform in the appendix.
>
> We have made the choice of writing a long paper for two reasons:
>
> 1. highlight the research process itself, so that each step of the process can be discussed (and is not arbitrarily put);
> 2. include remarks, footnotes, and discussions that could be valuable for a reader, but that are not part of our contributions.
>
> Specifically, it is impossible to avoid a discussion about the Mellin transform, even if we did not use it formally. First, we have used it informally to build the family of functions $g_{\Lambda_{}}$ (see Section 3.3). Second, it is the most natural way to "invert" the equation "$W Y \sim \mathcal{N}(0, 1)$", where the distribution of $W$ is known and the distribution of $Y$ is the unknown.

---

> > ### Comment · Reviewer_QEpf · 2023-03-03
> > **Response to the authors**
> >
> > I thank the authors for the detailed answer. I still believe some parts of the paper could be removed without impacting readability, but I agree this is a stylistic choice. I am happy to fully endorse the paper for publication.

---

### Review · Reviewer_7n9i · 2023-02-20

**Summary Of Contributions:**

**Summary:** The paper performs a thorough study of why previously proposed weight initialization schemes fail to stably lead to Gaussian pre-activations in deep neural networks (under RELU and tanh activation). This is important since several popular initialization schemes were designed with this consideration in mind. Based on the theoretical analysis of the failures, the paper identifies four conditions that need to hold for Gaussian pre-activations to propagate through many layers. Finally, the paper uses these insights to introduce a design scheme for pairs of parameterized activation functions and weight-initialization distributions that should lead to more well-behaved propagation of Gaussians. The scheme is non-trivial, and relies on some numerical approximations. Empirically, the scheme leads to better propagation of Gaussian pre-activations deeper into networks than RELU and tanh networks with standard initializations. This translates into networks that are well-trainable (low training loss) but seem to generalize worse (test loss/error) than standard schemes, limiting the practical impact of the work.

**Main contributions:**
 * Review of the theoretical reasoning behind common weight initialization schemes (Glorot, He, edge-of-chaos).
 * Theoretical and empirical analysis showing the shortcomings of the previous work.
 * Identification of theoretical conditions to allow for optimal propagation of Gaussian pre-activations.
 * Design of novel initialization scheme (which must be paired with custom activation function) based on these conditions and empirical evaluation with some promising results in terms of trainability, but negative results in terms of performance of the trained networks.

**Verdict:**
Overall I think the work is ready for publication in TMLR. Though I have not carefully checked all derivations and proofs, the main results look sound to me. Experiments are conducted well and support the criticism of previous methods. They also support the validity of the main theoretical claims in the paper, but unfortunately do not lead to better generalization performance (which is the criterion that would convince practitioners to adopt a novel method). The reason for this is most likely that the novel method requires an approximation which introduces a certain deviation from the strict theory. Putting all of this together, I think the paper adds an interesting and thorough analysis to the discussion on initialization schemes that ensure Gaussian pre-activations. The novel method in itself is not fully convincing yet and might need another iteration for being taken seriously by practitioners. The bigger question that the paper raises is whether the goal of achieving optimal Gaussian propagation is necessary, since the empirical results show that this is not the decisive factor for generalization performance. If the answer to that question is negative, then it would be unclear whether an improved method based on the current theory would lead to significant practical gains.

**Audience:**

Yes

**Broader Impact Concerns:**

No concerns.

**Claims And Evidence:**

Yes

**Requested Changes:**

I do not have any major comments that imply strictly required changes. Below are two related questions and some very minor comments.

 1. If the ultimate goal for well-trainable networks is to perfectly preserve information in the forward-pass very deep into the network, then would it not make sense to use, e.g., architectures with skip connections or residual processing (where layers by default perform essentially an identity operation) or invertible neural networks? In these architectures conservation of information is not only ensured at initialization but potentially also throughout training. It would be nice to add at least a short discussion, but if the authors think this is beyond the scope of the paper I'd also be happy with an informal answer via OpenReview.

2. (Somewhat related to 1.) to me the empirical results of the paper raise the question of whether Standard-Normal pre-activations should really be the goal to aim for. This is discussed in a paragraph towards the end of the paper, but it would be nice to see a clearer stance. Do the authors think that further improvements to the approximation to the scheme proposed in the paper should ultimately lead to good generalization? Is the goal of information-preservation correct, but aiming for Gaussian pre-activations too strong (which cannot capture higher-order moments)? Given the new knowledge from the paper, what are next possible steps to answer these questions? (I understand and respect that the authors might not want to speculate, particularly in the paper - but I'd be curious to hear their opinion)

 3. [Minor comment] P3 - Bayesian prior: Another important aspect of the prior in Bayesian NNs is to regularize during training, e.g., by inducing sparsity (log-uniform prior); see Bayesian interpretations of Dropout.

 4. [Minor comment] P2 typo: “and should the distribution of the”

 5. [Minor comment] P33: Figure 10a is referenced twice - one reference should be to Figure 10b.

**Strengths And Weaknesses:**

**Pro:**
  * Very detailed theoretical analysis.
  * Novel method derived from sound theoretical arguments.
  * Theoretical arguments are empirically verified (for both, flaws in argumentation of previous methods, and performance of the novel method).
  * Well written paper.

**Con:**
 *  By far the biggest weakness is that despite the thorough theoretical analysis and theory-guided design of the method, the practically most relevant question ("Do these insights lead to a new method that leads to neural networks that train more reliably to lower test error?") currently has a negative answer. This limits the potential practical impact and puts the focus of the paper on performing an analysis rather than proposing a superior new method. There is not too much that can be done here (often a novel idea does not immediately pan out into improved empirical results), but it is currently the main limiting factor in terms of practical impact to me (which does not play a role for TMLR publication).
 * Analysis and derivation fairly long-winded, leading to a very long paper. I personally would have preferred the main results, novel method and empirical analysis presented on roughly 10-12 pages with the remaining material pushed into the appendix (but this is partly a matter of taste).

---

> ### Author Response · Authors · 2023-03-03
> **Answer for Reviewer 7n9i**
>
> We thank Reviewer 7n9i for the insightful remarks and propositions of discussions.
>
> ## Information preservation, skip connections, and invertible NNs (Rev. 7n9i)
>
> The main benefit of using skip connections is indeed to improve the trainability of the NN, by preserving information of the input data throughout the NN (which is usually very deep).
> But we have decided to exclude these NNs from our study, since information is preserved in a very simple way.
> We focused on NNs without skip connections, because information preservation is non-trivial in these cases.
>
> The case of invertible NNs is trickier. It is true that information is preserved, but not in a trivial way: in general, the *invertible* layers are not identity functions (while skip connections are).
> So, this kind of networks is probably worth studying. But we do not know where to start, nor what to achieve.
>
>
> ## Initialization distribution and Bayesian prior (Rev. 7n9i and QEpf)
>
> We agree with Reviewers 7n9i and QEpf: the influence of a Bayesian prior can be interpreted as regularization. For instance, the standard loss in variational inference (which is an approximate Bayesian method) is [Graves2011]:
>
> $$L(\beta) = - \frac{1}{N} \mathbb{E}\_{\theta \sim \beta} \sum\_{i = 1}^N \log p(x\_i, y\_i | \theta) + \frac{1}{N} \mathrm{KL}(\beta || \alpha),$$
>
> where $N$ is the size of the dataset $\{(x_i, y_i)\}$, $p$ is the likelihood, $\beta$ is the candidate (variational) posterior and $\alpha$ is the prior. It is then clear that the second term, which only depends on $\beta$, $\alpha$, and $N$, corresponds to the influence of the prior, and is a regularization term at the same time.
>
> In the paragraph "Bayesian prior and initialization distribution" of the introduction, we only claim that a NN sampled from the Bayesian prior should be trainable, exactly as a NN at initialization. This (informal) claim is not contradictory with the fact that the prior plays *also* a regularizing role. A sentence about this regularizing role can possibly be added, if the reviewers want to.
>
> For the current OpenReview discussion, we add the following argument: let us consider the following situation:
>
> 1. we perform Bayesian "online learning" with a sequence of subsets $(D_1, D_2, \cdots)$ of a given dataset;
> 2. we compute the Bayesian posterior $p(\theta | D_1)$ with the first subset $D_1$;
> 3. if we want to get $p(\theta | D_n, \cdots, D_1)$, we just have to compute $p(D_n | \theta)$ and multiply it by $p(\theta | D_{n-1}, \cdots, D_1)$ (up to a normalizing constant).
>
> We apply this setup to NNs, and we approximate each Bayesian posterior $p(\theta | D_n, \cdots, D_1)$ with the variational posterior $\beta_n$ (see [Blier2018], Sec. 3.4). So, each $\beta_n$ is built by training the NN on subset $D_n$ with $\beta_{n-1}$ as a prior distribution. And, from an optimization point of view, it is natural to optimize $\beta_n$ with $\beta_{n-1}$ as a starting point. So, if we adapt the reasoning to the first step, $\beta_1$ should be optimized with the prior $\alpha$ as a starting point.
>
> We admit that this example is not a formal proof that the prior *should* be interpreted as a "by default" initialization distribution (and that is not our claim in the introduction). But it indicates that this interpretation is useful in some circumstances.
>
> Note: this interpretation is not contradictory with the papers mentioned by the Reviewer QEpf: [Tran2022] is about finding a prior matching a condition related to the function represented by a NN; [Rossi2017] proposes a data-dependent initialization (which is out of the scope of our paper).
>
>
> [Blier2018] *The Description Length of Deep Learning Models*, Blier et al., 2018.
>
> [Graves2011] *Practical variational inference for neural networks*, Graves, 2011.
>
> [Rossi2017] *Good Initializations of Variational Bayes for Deep Models*, Rossi, 2017.
>
> [Tran2022] *All You Need is a Good Functional Prior for Bayesian Deep Learning*, Tran et al., 2022.
>
>
> ## Length of the paper (Rev. 7n9i and QEpf)
>
> We understand the concerns of Reviewers 7n9i and QEpf about the length of the paper. We have put the sections about the Kolmogorov-Smirnov test and the Mellin transform in the appendix.
>
> We have made the choice of writing a long paper for two reasons:
>
> 1. highlight the research process itself, so that each step of the process can be discussed (and is not arbitrarily put);
> 2. include remarks, footnotes, and discussions that could be valuable for a reader, but that are not part of our contributions.
>
> Specifically, it is impossible to avoid a discussion about the Mellin transform, even if we did not use it formally. First, we have used it informally to build the family of functions $g_{\Lambda_{}}$ (see Section 3.3). Second, it is the most natural way to "invert" the equation "$W Y \sim \mathcal{N}(0, 1)$", where the distribution of $W$ is known and the distribution of $Y$ is the unknown.

---

> > ### Comment · Reviewer_7n9i · 2023-03-05
> > **Thank you for the detailed response.**
> >
> > Since none of the points raised by me were critical, there is not much further discussion needed. Just to clarify:
> >   * I did not mean to ask for additional experiments with networks with skip connections or invertible layers - I was just curious on a very high level whether the authors think that in the future it is worth pursuing preservation of Gaussianity, or solving the problem of information conservation with the architecture itself rather than the initialization (i.e. skip connections, invertible layers, etc.).
> >   * I don't disagree with the authors' argument regarding interpreting the prior as a "by default" initialization - but the prior also has its regularization effect throughout training (i.e. it "keeps pulling the weight-distribution towards the prior"). Practicioners typically emphasize the latter (but again, there was no wrong claim in the orig. manuscript).
> >   * Finally, I think that the choice to report (parts of) the research process rather than just the outcomes can be justified in this case and might be helpful to readers who want to employ the same broad methodology but not necessarily the precise same algorithm. Whether one prefers a crisp paper with a long appendix or a detailed "multi-step narrative" is ultimeately a matter of taste and there are pros and cons both ways. So no need to make any further changes, and I want to thank the authors for trimming the paper a bit.
> >
> > To the best of my knowledge all open issues have been addressed, and there is no further discusison needed for the points I raised, but I'd still be curious to hear what D3ec thinks about the updated manuscript that corrects the error in Proposition 4.

---

### Author Response · Authors · 2023-02-10
**Correction of Proposition 4**

We thank Reviewer D3ec for spotting an error in Proposition 4 and proposing a counter-example. So, we have updated our paper accordingly. Proposition 4 has been replaced by Remark 7 and Appendix B. Also, a discussion about the independence of the pre-activations has been added to the conclusion. All the additions are highlighted in red in the updated version of the manuscript.

---

### Author Response · Authors · 2023-03-03
**Main answer (1/2)**

We would like to thank all the reviewers for their careful reading, suggestions, and propositions of discussions.

We organize our answer as follows: in the present "official comment", we discuss the major points raised by the reviewers; in the individual answers, we discuss the remaining points (which are possibly shared by several reviewers).

## Availability of the code, numerical computation of $\phi_{\theta}$ and numerical errors (Rev. QEpf)

We have made available for the reviewers the code involving the numerical computation of $\phi_{\theta}$, so the whole process can be replicated, checked, and discussed.

In short, the process can be divided into three steps:

1. approximation of the density $f_Y$: this step is explained in Section 3.3. We report that the $\mathcal{L}^{\infty}$ loss that we used in Sec. 3.3, is lower than $10^{-2}$ for all tested $\theta$;
2. interpolation of the theoretical $\phi_{\theta}$: given the approximation of $f_Y$, we are able to compute numerically the function $\phi_{\theta}$ at any point (see Section 3.4). We do this for $400$ points uniformly distributed in the interval $[0, 10]$;
3. best fit of the interpolation: we fit the interpolation (or "graph") proposed in step 2 with a parameterized function $\hat{\phi}_{\theta_{}}$. This parameterized function is coded in `activation.ActivationFunction`. We train the parameters of $\hat{\phi}\_{\theta}$ in order to minimize a modified $\mathcal{L}^2$ distance between $\hat{\phi}\_{\theta}$ and the interpolation (see the code for a detailed explanation). The final root mean square error is always smaller than $2 \cdot 10^{-2}$.

**Flexibility of the code.**
The code can be easily adapted to various purposes.
Specifically, it can be used to find a function $\phi$ such that:
$$W \sim \mathrm{P}, X \sim \mathrm{R} \Rightarrow W \phi(X) \sim \mathcal{T},$$
where $\mathrm{P}$, $\mathrm{R}$ and $\mathcal{T}$ could be selected by the user.

## Is it desirable to have $\mathcal{N}(0, 1)$ pre-activations? (Rev. 7n9i)

We agree with Reviewer 7n9i that it is a question that remains to be answered.
We also agree that there is a broader question to answer: should we aim for information-preservation? Or: is it desirable to keep constant a given property during propagation?

We think that the method we have developed can be generalized, and used to test hypotheses about the ideal distribution of the pre-activations at initialization.
To this aim, we could perform a series of experiments where:

1. we fix a target distribution $\mathcal{D}_l^*$ for each layer $l$;
2. we want to check whether having $Z^l_j \sim \mathcal{D}_l^*$ for each $l$ at initialization: a) accelerates the training process; b) improves the final test loss;
3. for each $l$, we find an initialization distribution $\mathrm{Q}_l$ of the weights and an activation function $\phi_l$ such that: $\frac{1}{\sqrt{n\_l}}\sum\_{j = 1}^{n\_l} W^l\_j \phi\_l(Z^l\_j) \sim \mathcal{D}\_{l + 1}^*$, when $Z^l_j \sim \mathcal{D}_l^*$;
4. we make an empirical conclusion about our initial choices for $(\mathcal{D}_l^*)_l$.

We emphasize that such a series of experiments is **hypothetical** at that point. First, the influence of the dependence between pre-activations (pointed out by Rev. D3ec and explored in Appendix B in the 2nd version of the paper), is not solved yet and is not even well understood. Second, aiming for a non-Gaussian pre-activation is a theoretical challenge, since Proposition 3 (and Constraint 1) cannot be used or adapted.

Eventually, we think that further exploration in that direction should be inspired by empirical results with good-performing NNs.
For instance, we could test heavy-tailed target distributions $\mathcal{D}_l^*$ with various Weibull tail parameters.

Note: [Fortuin2021] showed that priors with heavy tails (e.g., Laplace tails) fit better the empirical distribution of the weights after training than a Gaussian distribution. Such heavy-tailed weights are not compatible with Gaussian pre-activations, but they play a role *during training*, and not at init.

[Fortuin2021] *Bayesian Neural Network Priors Revisited*, Fortuin and Garriga-Alonso et al., 2021.


## Non-odd activation functions

We have added Remark 10 to indicate that we have made a choice when computing $\phi_{\theta}$:
we wanted them to be odd.
In fact, it is possible to obtain Gaussian pre-activations with non-odd activation functions.
Our choice of odd activation functions was based on the fact that they are one of the two "natural" choices we had, along with positive ones.

---

### Author Response · Authors · 2023-03-03
**Main answer (2/2)**

## Independence of the pre-activations and "Proposition 4" (Rev. D3ec)

We thank again Reviewer D3ec for spotting an error in the former "Proposition 4" (v1 of the paper).
As mentioned in our first "official comment", we have replaced it with Remark 7 and Appendix B. We have also loosened our claim of having an "exact" Edge of Chaos (see the introduction of Sec. 4): actually, our result is "non-asymptotic with independent pre-activations".

Notably, we discuss the influence of the dependence of the preactivations $(Z^l_j)_j$ on the Gaussianity of the $(Z^{l+1}_i)_i$. Given the experimental results of Appendix B, the dependence plays a role, especially in NNs with a small width (small number of neurons $n_l$ per layer).
The importance of the dependence can be measured empirically (see Fig. 11).

Ideally, an approximate alternative to Proposition 4 could be found as follows, with an adapted measure $m(\mathbf{Z}^l)$ of the dependence between the pre-activations $(Z^l_j)_j$. That is, a dependence measure with the following property:
if $m(\mathbf{Z}^l)$ is in a specific range of values, then:

1. we can ensure that the $(Z^{l+1}_i)_i$ are close to $\mathcal{N}(0, 1)$ (e.g., the KS-statistic between $Z^{l+1}_i$ and $\mathcal{N}(0, 1)$ is below a given threshold);
2. we can ensure that $m(\mathbf{Z}^{l + 1})$ is in a specific range of values.

With this **hypothetical** measure, we could take into account the dependence structure in such a way that we can control it over the propagation in the entire NN. We already know that the correlation between the pre-activations cannot be
used for that purpose (because it is equal to zero). So, there is still room for improvement here, but with an approximate solution.

**Additional note (exploratory):** in order to obtain a theoretical result about the Gaussianity of:

$$Z^{l+1} = \frac{1}{\sqrt{n}} \sum_{j = 1}^{n_l} W^l_j \phi(Z^l_j),$$

with dependent $(Z^l_j)_j$, a "dependent" version of the Berry-Esseen theorem may be used. According to [Blum1958], it would be sufficient to prove that the random variables $(W^l_j \phi(Z^l_j))_j$ are exchangeable (they are), uncorrelated (they are), and verify $m(Z_j^l, Z_i^l) = 0$ for any $i \neq j$, with the measure of dependence $m(\cdot, \cdot)$ defined by:

$$m(Z\_j^l, Z\_i^l) := \mathbb{E} \left[ (W^l\_j \phi(Z^l\_j))^2 (W^l\_i \phi(Z^l\_i))^2  \right] - (\mathbb{E} \left[ (W^l\_1 \phi(Z^l\_1))^2 \right])^2.$$

But in general $m(Z_j^l, Z_i^l) \neq 0$. Typically, independence of $(Z_j^l, Z_i^l)$ implies $m(Z_j^l, Z_i^l) = 0$ (but the reciprocal is false), so $m$ is an indicator of independence (like $\mathrm{corr}$).

We can compute $m$ in the two-layer NN case proposed in Example 1 (App. B). The output is: $Z := \frac{1}{\sqrt{2}} (Z_1 + Z_2)$, with $Z_1 := W^{(2)}_1 \phi(W_1^{(1)} X)$ and $Z_2 := W^{(2)}_2 \phi(W_2^{(1)} X)$, where $X$ is some random variable. We denote by $\sigma_1^2$ and $\sigma_2^2$ the respective variances of the weights $W^{(1)}_i$ and $W^{(2)}_i$. We have:

* with $\phi = \mathrm{Id}$: $m(W_1^{(1)} X, W_2^{(1)} X) = \sigma_1^4 \sigma_2^4 (\mathbb{E}[X^4] - \mathbb{E}[X^2]^2)$, which is zero iff $X^2$ is a constant; if $X \sim \mathcal{N}(0, 1)$, then $m = 2 \sigma_1^4 \sigma_2^4$; so, this configuration is unfavorable;
* with $\phi = \mathrm{sgn}$: $m(W_1^{(1)} X, W_2^{(1)} X) = 0$; favorable config (but this activation function is unusable).

We may guess that such a measure could be helpful to understand this dependence phenomenon, explain the results of Appendix B, and evaluate the quality of activation functions (in terms of dependence-breaking). According to App. B and the back-of-the-envelope computation proposed here, it seems that $\mathrm{sgn}$- or $\tanh$-like activation functions could be better suited than $\mathrm{Id}$-like ones for obtaining Gaussian pre-activations accommodating for units dependence.

[Blum1958] *Central Limit Theorems for Interchangeable Processes*, Blum et al., 1958.

---

> ### Comment · Reviewer_7n9i · 2023-03-05
> **Thank you for the detailed response**
>
> Please see my detailed response below my review - I currently believe all open issues have been addressed and that the paper is ready for publication, but I would like to hear D3ec's updated opinion regarding Proposition 4 for a final decision.

---

> ### Comment · Reviewer_D3ec · 2023-03-05
> **thanks**
>
> Many thanks for the detailed response that goes beyond the materials in the manuscript.
>
> I think the actual difficulty of the problem that the paper intended to solve was greatly underestimated. I can see that one can try to bake in different notions of "approximate Gaussianity" (which I think should appear everywhere in the manuscript, as it is now incorrect to say that the scheme maintains Gaussianity). I also find "non-asymptotic" EOC is a vague term. The EOC curve that the paper presents follows the same EOC formula that was derived in the infinite-width limit, and since we know that the Gaussianity is only approximate for both known initialization schemes and proposed schemes, I think calling the EOC curves here as non-asymptotic but still exact does not quite reflect its actuality; I would have imagined a different EOC formula where the width and the depth are both factored in, as conventional for the term "non-asymptotic" in mathematics or elsewhere.
>
> The main dilemma is this. The whole analysis and the proposal rely on the previously Proposition 4 and the now assumption that the neurons are independent, which are not true. Any logical deduction from a wrong statement should be warned against. Suppose that the paper tries to claim "approximate Gaussianity" by some measure. That is, the central claim might then be changed from "constructing a scheme that maintains Gaussianity" to something like "constructing a scheme that approximates Gaussianity better than well-known schemes". Now one can see that the new claim is quite distant from the materials of the paper. Of course the paper would then have to follow with a lot of discussions about the different measures to test Gaussianity, the effect of width and depth, the effect of architectures/activation functions, how things vary with $\sigma_w$ and $\sigma_b$, etc, in comparison with known initialization distributions. All of these numerical tests have to be carried out unfortunately since the one mathematical claim is already wrong. I think one can eventually find one approximate Gaussianity measure that fits the picture. But for practitioners who already know that just increasing the width can help with Gaussianity for **any** initialization distribution, would they chase this path?
>
> In short, while I think the paper would have to change its central claim and do a lot more work to support it, I also struggle to find a good reason to believe the new pursuit would be of sizable interest.

---

> > ### Author Response · Authors · 2023-03-06
> > **Answer**
> >
> > We want to thank Reviewer D3ec for sharing some remaining concerns with our contributions.
> >
> > In summary, we agree with Reviewer D3ec that our claims on Gaussianity and exact EOC should be changed, but we think that the role of dependence is overstated in Reviewer D3ec's answer. We would like to defend that point in this answer. We will include in the Introduction a caveat about the condition of application of our result, acknowledging that it requires an independence assumption. Edits will also be made throughout the text regarding statements of exact Gaussianity or non-asymptotic EOC.
> >
> > ### About the role of dependence
> > First, we would like to recall that our result remains *theoretically exact* in the finite-width case with i.i.d. pre-activations (the exactness may be discussed in practice due to numerical approximations, which are unavoidable).
> > And, even without taking dependence into account, we show that our framework generally yields preactivations whose KS distance to the Gaussian is lower than networks with ReLU or tanh activation function, see Figure 6 and Figure 7. This holds true for many configurations of widths and depths.
> >
> > Second, when using a Central Limit Theorem on a sequence $(X\_n)\_n$ of *dependent* r.v., the limit $n \rightarrow \infty$ is involved in two ways: (i) the marginal distributions of the $X\_n$ have to be averaged; (ii) the specific dependence between the $(X\_n)\_n$ has to be averaged.
> > We claim to have made an advance in solving problem (i), and to have proposed a discussion about the importance of problem (ii) (thanks to the remark of Rev. D3ec, see App. B).
> >
> > Third, existing results are disappointing regarding these *two* points.
> > Moreover, even the asymptotic results taking into account dependency are weak in some way.
> > For instance, the famous result of [Matthews2018] is only valid if we perform the proof sequentially as the layer index $l$ grows.
> > From an i.i.d. point of view, this assumption is unnecessary: it is sufficient to make the size $n\_l$ of layer $l$ tend to infinity to make the pre-activations $Z^{l+1}\_i$ tend to a Gaussian distribution.
> > This result remains valid even if the other layer sizes $n\_l$ are finite.
> > From a dependent point of view, such recurrence on $l$ becomes necessary, in order to have a dependency relation between the $Z^{l}\_j$ which is manageable when using common versions of dependent CLTs.
> >
> > ### About the possibility for practitioners to chase our path
> > Our paper has the merit of proposing a completely novel approach that provides a new angle of research to study the Gaussianity of preactivations. We foresee multiple possible extensions, in particular thanks to the proposed package, concerning other activation functions (including positive ones), and families of initialization distributions other than the Weibull one.
> >
> > ### About the interest of the paper
> > In addition to the previous point, we believe that our other contributions also provide new insights on the question of Gaussian preactivations, including our experimental illustrations on realistic datasets CIFAR-10 and MNIST (Section 2.2), the EOC counter-examples of Section 2.3, and the formalization of Section 2.4.
> >
> >
> > [Matthews2018] *Gaussian process behaviour in wide deep neural networks*, Matthews et al., 2018.

---

> > > ### Comment · Reviewer_D3ec · 2023-03-06
> > > **reply**
> > >
> > > Thanks for the response. My main point is the following:
> > >
> > > > First, we would like to recall that our result remains theoretically exact in the finite-width case with i.i.d. pre-activations
> > >
> > > the basis is on the **wrong** assumption of i.i.d. pre-activations. The theory breaks down from this starting point. I'm unsure if the theory has insights if it is based on a wrong hypothesis.

---

### Decision · Action_Editors · 2023-04-30

**Recommendation:** Reject

**Comment:**

In light of the above discussion of claims and evidence, I conclude that the article addresses an interesting topic and contributes a valuable discussion, but that it will need a major revision ensuring that the claims are correct and supported by accurate convincing and clear evidence before it can be considered for publication at TMLR. Hence I must reject the article at this time, but I encourage a resubmission upon addressing the concerns voiced by the reviewers.


**Audience:**

The topic of the work is of interest to TMLR's audience.

**Claims And Evidence:**

The submission studies feature propagation in neural networks and the question for which activation functions and initialisations the pre activations can be ensured to be Gaussian across the layers of a network. It observes that finite width networks usually do not have Gaussian pre activations, takes a critical view at previous works, and proposes necessary conditions to make the pre activations Gaussian. At the end of the discussion period, the reviewers recommend Reject and Leaning Accept (twice).

The main concern is an error that one of the reviewers found, which, in their words: leads to an unfortunate situation where theory is based on a wrong hypothesis and as such the derivations cannot be interpreted in a good way. They identified two possible ways to make amendments: to try to test the iid pre activation hypothesis or to try to test Gaussianity, both of which come with challenges and will necessitate many more numerical tests. The central claims would need to be changed, since known initialisations  achieve approximate Gaussianity. In either case, this would necessitate a substantial overhaul of the paper.

Two other reviewers indicated in their final recommendation that they could accept the work provided the authors state and discuss the iid assumptions and tone down the claims regarding exact Gaussianity and EOC prominently throughout the paper. They indicate they are not too optimistic that the current work lays a strong foundation for practitioners to follow, but that it has its merits in summarising issues with previous work and illustrating the difficulty of a sensible idea. They conclude that, in light of the limitations and valid criticism of other reviewers they are not ready to champion the submission, but that they believe that the authors would be able to revise in regard to the criterion of accurate, convincing and clear evidence. The third reviewer, leaning accept, agrees that the article should clarify its current limitations and reduce some of its claims.